# Rate or Fate? RLV$^\varepsilon$R: Reinforcement Learning with Verifiable Noisy Rewards

**Ali Rad** [1] **Khashayar Filom** [1] **Darioush Keivan** [1] **Peyman Mohajerin Esfahani** [2] **Ehsan Kamalinejad** [1]

## Abstract

Reinforcement learning with verifiable rewards (RLVR) trains a policy by verifying sampled completions and reinforcing higher-scoring outputs, but practical verifiers (e.g., incomplete unit tests or noisy judges) are prone to false positives and false negatives. We ask when such noise merely slows learning and when it reverses it. Modeling GRPO-style RLVR as a bandit over recurring *reasoning modes*, we derive mean-field replicator-style (natural-selection) flow on the probability simplex. The dynamics decouples into within-correct-mode competition and a one-dimensional evolution for the mass on incorrect modes, whose drift is determined solely by Youden's index $J = \text{TPR} - \text{FPR}$. This yields a sharp phase transition: when $J > 0$, the incorrect mass is driven toward extinction (learning); when $J = 0$, the process is neutral; and when $J < 0$, incorrect modes amplify until they dominate (anti-learning and collapse). In the learning regime $J > 0$, noise primarily rescales convergence time ("rate, not fate"). Experiments on verifiable programming tasks under synthetic noise reproduce the predicted $J = 0$ boundary, and additional PPO experiments on mathematical reasoning show the same qualitative transition, suggesting that the phenomenon extends beyond GRPO and programming benchmarks. Beyond noise, the framework offers a general lens for analyzing RLVR stability, convergence, and algorithmic interventions.

**Code:** https://cognichip.github.io/Noisy-RL.

## 1. Introduction

Reinforcement learning (RL) is increasingly used to steer large language models (LLMs). In domains where correct-

ness is programmatically checkable–e.g., unit tests for code or exact answers for some math problems–*reinforcement learning with verifiable rewards* (RLVR) offers a compelling alternative to learning a reward model. Recent group-normalized policy-gradient variants, such as Group Relative Policy Optimization (GRPO), can be sample-efficient and avoid training an explicit critic (Shao et al., 2024; Ahmadian et al., 2024; Wen et al., 2025; Su et al., 2025).

In practice, however, "verifiable" rewards are often *sloppy*. Test suites are incomplete, checkers can be flaky, and LLM-as-a-judge signals can be biased. Because policy-gradient methods repeatedly amplify the feedback they receive, even mild verification bias can compound across updates, leading to mode collapse, reward hacking, or systematic performance degradation (Chen et al., 2025; Cai et al., 2025). This paper asks: *When does noisy verification in RLVR merely slow training, and when does it flip the learning direction?*

**A single scalar threshold.** We identify a simple verifier-quality statistic that governs the *direction* of learning under GRPO-style group normalization. Under a binary noisy-verifier model with false positives and false negatives, the update direction is controlled by *Youden's index* $J = \text{TPR} - \text{FPR}$ (Youden, 1950). Intuitively, $J$ measures whether the verifier is positively correlated with true correctness ($J > 0$), uninformative ($J = 0$), or perversely anti-correlated ($J < 0$).

**Contributions.** The contributions of this study are summarized as follows:

- **Noisy-verifier formalization.** We formalize **RLV$^\varepsilon$R**: RLVR with *verifiable but noisy* binary rewards, parameterized by $(\text{TPR}, \text{FPR})$ and summarized by $J = \text{TPR} - \text{FPR}$; cf. Section 3.

- **Mean-field dynamics and phase line.** Viewing each prompt as a small bandit over reasoning modes, we derive **mean-field ODEs** (cf. Proposition 5.1), for correct ("good") and incorrect ("bad") probability masses. In the simplest, binary, case of one ultimate good and one ultimate bad solution, the incorrect ("bad") probability masses, $p(t)$, obey the ODE

$$\dot{p} = -J\, p^2 (1-p)^2\, \psi(p)$$

[1]Cognichip AI [2]University of Toronto. Correspondence to: Ali Rad <ali@cognichip.ai>.

*Proceedings of the 43rd International Conference on Machine Learning*, Seoul, South Korea. PMLR 306, 2026. Copyright 2026 by the author(s).

where $\psi : [0, 1] \to \mathbb{R}_{\geq 0}$ is strictly positive on $(0, 1)$. This establishes a *phase transition at $J = 0$*: $J > 0$ yields learning, $J = 0$ neutral drift, and $J < 0$ anti-learning (see Theorem 4.2) as well as the dependence of the **rate** of change of bad arm probability on $J$.

- **Geometry and extensions.** We connect GRPO to replicator/natural-gradient flow on the probability simplex (see Section 5), obtain multi-mode decoupling dynamics, and analyze how convergence rates, learnability, and KL regularization depend on verifier noise (see Section 5.3).

- **Empirical validation.** On Python code generation with synthetic verifier noise, we reproduce the $J = 0$ boundary (see Figure 1) and show that, for $J > 0$, noise primarily affects *rate, not fate*.

**Why this is RLVR-specific.** The use of $(\mathrm{TPR}, \mathrm{FPR})$ is not meant to introduce a new noise model by itself; rather, it isolates how this familiar noise channel interacts with group-normalized policy-gradient updates in RLVR. The resulting ODE tracks the full mode-probability vector, not only aggregate accuracy, and therefore exposes effects that are specific to RLVR post-training dynamics: within-good diversity collapse, spreading among bad modes, prompt learnability, and the role of KL anchoring.

**Roadmap.** After reviewing relevant literature in Section 2, we set up the bandit/mode abstraction of LLMs and the noisy-verifier model in Section 3. We present our mean-field model, and its dynamical consequences, including phase transition, first in the binary case in Section 4, and then more generally in Section 5. Extensions of these results appear in Subsection 5.3. Finally, Section 6 evaluates our results empirically.

**Conflict of Interest Disclosure.** The authors declare no financial conflicts of interest related to this work.

## 2. Background & Related Work

**RLVR and group-normalized policy gradients.** Recent breakthroughs in the reasoning capabilities of large language models (LLMs) through Reinforcement Learning (RL)–particularly Reinforcement Learning with Verifiable Rewards (RLVR) and group-normalized algorithms such as Group Relative Policy Optimization (GRPO) (Wen et al., 2025; Su et al., 2025)–have greatly expanded the frontier of model intelligence (Shao et al., 2024). Group-normalized approaches in RL, like GRPO, eliminate the need for an explicit reward model (a critic or a PRM, (Schulman et al., 2017; Lightman et al., 2024)) in verifiable domains such as mathematics and code generation, and demonstrate that even a few rollouts per prompt are often sufficient to approximate the advantage of each generated sequence (Wen et al.,

2025; Su et al., 2025).

**LLM as a Multi-Armed Bandit.** When the reward is evaluated at the completion level rather than per token, it is often more appropriate to treat the entire output sequence as a single decision made by the policy. This viewpoint naturally suggests a bandit-style abstraction, where each sequence corresponds to one action (or "arm") and the learning signal is attached to that action as a whole (Kreutzer et al., 2017; Nguyen et al., 2017; Dang et al., 2025). Earlier sequence-level policy gradient methods–such as RLOO and related REINFORCE variants (Ahmadian et al., 2024)–implicitly operated in this regime, while contemporary approaches like GRPO (Shao et al., 2024) make this perspective explicit by defining advantages and updates directly over full generations.

**Noisy feedback, judges, and reward misspecification.** The effects of noisy or biased supervision are classical in learning theory and RL. In LLM alignment, imperfect verifiers and judge models can induce reward hacking or collapse, motivating both empirical and theoretical analyses of noisy-verifier regimes (Chen et al., 2025; Cai et al., 2025). Our focus is on isolating a *directional* condition (a "phase line") for GRPO under binary noisy verification.

**Noisy Reward for Coding Tasks.** Noisy verification is especially acute in RLVR for coding, where unit tests are inherently incomplete and many distinct implementations can be semantically correct, unlike short-answer math problems (e.g., AIME) or fixed-answer benchmarks (e.g., MMLU; (Hendrycks et al., 2021)). As models are pushed to harder programming tasks, test coverage and fidelity inevitably degrade, and pass/fail signals can become weakly correlated with true functional correctness, sometimes approaching chance-level reliability. We therefore focus our experiments on Python programming tasks, where imperfect verification is the norm and the induced noise regime is both realistic and practically important.

**Replicator dynamics and natural gradients.** Our mean-field GRPO dynamics recover a replicator skeleton with a drift term controlled by the verifier alignment. Replicator dynamics appear in evolutionary game theory as "natural selection" flows. On the probability simplex, they coincide with natural-gradient flows under the Shahshahani/Fisher geometry (Shahshahani, 1979; Amari et al., 2019).

## 3. Problem Setup

We study per-prompt RLVR updates with binary rewards and model the effect of verification noise.

**From sequences to modes (bandit abstraction).** Fix a prompt $x$. Repeated sampling from an LLM at non-zero

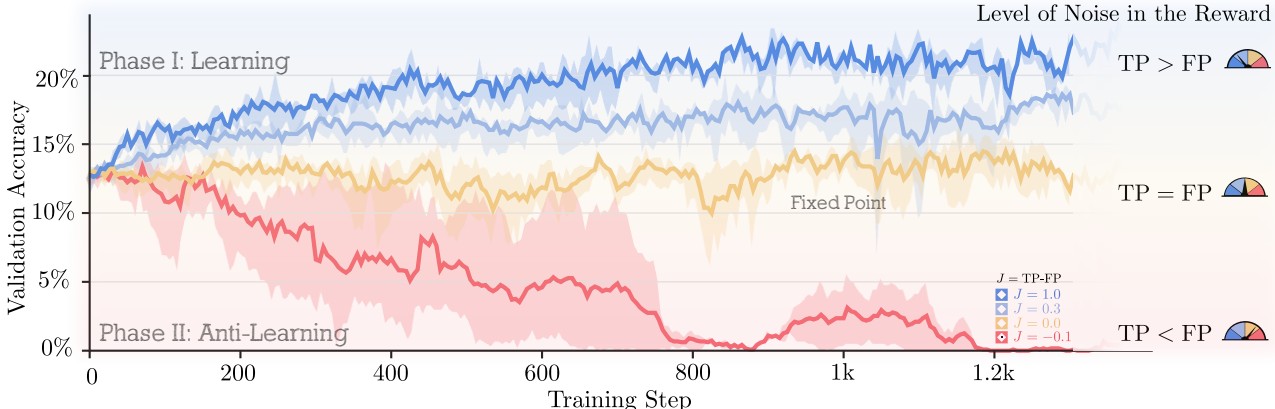

*Figure 1.* **Phase transition under a sloppy verifier.** Validation pass@1 during GRPO training under synthetic reward noise, parameterized by $J = \mathrm{TPR} - \mathrm{FPR}$. Training uses Qwen2.5-3B fine-tuned with GRPO on filtered OpenR1 code-generation prompts; validation uses the noise-free unit-test checker. Curves are averaged over five seeds, and shaded bands visualize across-seed variability. The transition at $J = 0$ matches the theory: $J > 0$ improves, $J \approx 0$ stagnates, and $J < 0$ degrades.

temperature yields many sequences that often cluster into a small number of recurring *reasoning modes* (solution templates). We model these modes as arms of a categorical policy $\pi_\theta(\cdot \mid x)$ with logits $\theta$ (Appendix B gives details). Let $\mathcal{H} = \mathcal{H}^+ \cup \mathcal{H}^-$ denote good (correct) and bad (incorrect) mode sets with $|\mathcal{H}^+| := K$ and $|\mathcal{H}^-| := M$.

**Probability simplex.** We shall treat a $d$-dimensional probability vector as a point of the simplex

$$\Delta^d := \left\{ \mathbf{x} \in \mathbb{R}^{d+1}_{\geq 0} : \mathbf{1}^\top \mathbf{x} = 1 \right\}.$$

Therefore, the mode-probability vector can be written as $(\mathbf{p}^+, \mathbf{p}^-) \in \Delta^{K+M-1}$ where $\mathbf{p}^+ \in \mathbb{R}^K_{\geq 0}$ (good) and $\mathbf{p}^- \in \mathbb{R}^M_{\geq 0}$ (bad). Define the total *good mass* $s := \mathbf{1}^\top \mathbf{p}^+$ and total *bad mass* $p := \mathbf{1}^\top \mathbf{p}^- = 1 - s$, and the within-class compositions (when $s, p > 0$)

$$\mathbf{y} := \mathbf{p}^+/s \in \Delta^{K-1}, \quad \mathbf{z} := \mathbf{p}^-/p \in \Delta^{M-1}$$

To summarize, the scalar $p$ captures "how often we are wrong" while vectors $\mathbf{y}, \mathbf{z}$ model inner good/bad dynamics.

**Noisy verifier.** Let $z \in \{0, 1\}$ denote the *true* correctness and let $r \in \{0, 1\}$ be the observed reward produced by a noisy verifier with

$$\mathrm{TPR} := \Pr(r = 1 \mid z = 1), \quad \mathrm{FPR} := \Pr(r = 1 \mid z = 0);$$
$$J := \mathrm{TPR} - \mathrm{FPR} \in [-1, 1].$$

Equivalently, with false-negative/positive rates $\delta_{\mathrm{FN}} := 1 - \mathrm{TPR}$ and $\delta_{\mathrm{FP}} := \mathrm{FPR}$, hence $J = 1 - \delta_{\mathrm{FN}} - \delta_{\mathrm{FP}}$.

**Scope of the verifier-noise assumption.** In the main analysis we assume mode-independent, block-symmetric verifier

noise: conditional on true correctness, all good modes share the same TPR and all bad modes share the same FPR. This assumption is intended to isolate the coarse alignment between the verifier and correctness. Mode-dependent verifiers would replace the scalar gap $J$ by arm-specific gaps and may introduce additional within-block selection effects; we view that setting as a natural extension.

**GRPO-style group normalization.** For each prompt, GRPO samples $G$ rollouts $y_g \sim \pi_\theta(\cdot \mid x)$, assigns rewards $r_g = r(x, y_g)$, and forms normalized advantages

$$\widehat{A}_g = \frac{r_g - \bar{r}}{\sigma_r + \epsilon}, \qquad \bar{r} = \frac{1}{G}\sum_{g=1}^{G} r_g, \quad \sigma_r^2 = \frac{1}{G}\sum_{g=1}^{G}(r_g - \bar{r})^2.$$

A basic policy-gradient step (abstracting away PPO clipping (Schulman et al., 2017) and KL penalties, which are analyzed in Appendix G and Appendix H) is REINFORCE-based (Williams, 1992)

$$\Delta\theta = \eta \frac{1}{G}\sum_{g=1}^{G} \widehat{A}_g \nabla_\theta \log \pi_\theta(y_g \mid x) \qquad (1)$$

where $\eta$ is the learning rate. (We reiterate that in our bandit model $\theta$ should not be thought of as the language model's parameters, but as logits assigned to various arms (solutions); cf. Appendix B.)

**Mean-field approximation.** A key insight is to study the update induced by Equation (1) under a mean-field approximation that treats each prompt independently and tracks how probability mass shifts between modes. Here "mean-field" means that the empirical group mean and variance in

Equation (1) are replaced by their population counterparts for the current prompt. This is exact in the large-group limit and gives the leading-order drift for finite $G$; finite-group sampling noise can affect the variance and sharpness of the trajectory, but the expected drift direction remains governed by the sign of the verifier-correctness gap under the block-symmetric model.

We start with the simple case of binary reward in the next section.

# 4. The Case of Binary Arms

It is easiest to first state our results in the case of binary arms: one good meta-mode vs. one bad meta-mode. Let

$$f(\text{good}) := \mathbb{E}[\widehat{A} \mid \text{good}], \qquad f(\text{bad}) := \mathbb{E}[\widehat{A} \mid \text{bad}],$$

denote conditional mean normalized advantages ("fitness"). In this setup, $p$ becomes the probability of generating the bad solution; and the parameter is a logit $z$ where $p = \sigma(z) = 1/(1 + e^{-z}) \equiv \pi(\text{Bad})$.

## 4.1. Replicator skeleton for group-normalized updates

Differentiating the logistic function $\sigma$ yields

$$\nabla_z \log \pi(\text{bad}) = 1 - p, \quad \nabla_z \log \pi(\text{good}) = -p.$$

Therefore, $\Delta z = \eta \, \mathbb{E}[\widehat{A} \, \nabla_z \log \pi(a)]$ simplifies as $p(1 - p)\big(f(\text{bad}) - f(\text{good})\big)$. Passing to continuous time, given $\dot{p} = \frac{dp}{dz} \dot{z} = p(1 - p) \dot{z}$, we arrive at the *replicator skeleton* for the bad mass:

$$\dot{p} = -\eta \, [p(1-p)]^2 \big(f(\text{good}) - f(\text{bad})\big). \qquad (2)$$

This is a special case of the replicator dynamics (8), which also appears in Section 5 where we generalize the results of this section to the multi-mode/arm setting.

## 4.2. Youden's index appears

Youden's index, $J$, controls the "fitness gap" in (2). To elaborate, let $q(p) = \mathbb{E}[r]$ denote the probability that the noisy verifier outputs 1 given current bad mass $p$. Under the noise model,

$$q(p) = (1 - \delta_{\text{FN}}) - J \, p. \qquad (3)$$

In the mean-field approximation, group normalization divides by $\sigma(p) = \sqrt{q(p)(1 - q(p))}$ (Appendix C).

**Proposition 4.1** (Youden-controlled fitness gap). *In the binary good/bad setting with Bernoulli reward $r \sim$ Bernoulli($q(p)$), the fitness gap satisfies*

$$f(\text{good}) - f(\text{bad}) = \frac{J}{\sigma(p)}. \qquad (4)$$

Combining Equations (2) and (4) yields

$$\dot{p}(t) = -\eta \, \frac{J}{\sigma(p(t))} \, [p(t)(1 - p(t))]^2. \qquad (5)$$

## 4.3. Phase transition at $J = 0$

Because $\sigma(p) > 0$ for interior states $p \in (0, 1)$ (other than the extreme case of $\{\delta_{\text{FN}}, \delta_{\text{FP}}\} = \{0, 1\}$), the right-hand side of Equation (5) can be written as $-J \, p^2 (1 - p)^2 \, \psi(p)$ where $\psi$ is a function strictly positive on $(0, 1)$. This implies the drift has sign $-J$ everywhere away from the boundaries:

**Theorem 4.2** (Sign-of-$J$ phase transition). *Assume $p(0) \in (0, 1)$ and consider the mean-field GRPO dynamics Equation (5).*

- *If $J > 0$, then $p(t)$ decreases monotonically and converges to $0$ (learning succeeds).*

- *If $J = 0$, then $\dot{p}(t) = 0$ to first order and $p(t)$ exhibits neutral drift (no directional learning signal).*

- *If $J < 0$, then $p(t)$ increases monotonically and converges to $1$ (anti-learning).*

*Moreover, $p = 0$ and $p = 1$ are absorbing: if the base policy assigns zero mass to good modes for a prompt ($p(0) = 1$), RLVR cannot bootstrap learning on that prompt.*

Next, we have the following corollary on the rate of convergence for which we refer the reader to Appendix A.

**Corollary 4.3** (Polynomial convergence rates (binary mean-field)). *Under Equation (5) with $J > 0$ and $\delta_{\text{FP}} < 1$, the bad mass decays polynomially. If $\delta_{\text{FN}} > 0$ (nondegenerate verifier noise), then $p(t) = \mathcal{O}(t^{-1})$. If $\delta_{\text{FN}} = 0$ (variance-degenerate at the good vertex), then $p(t) = \mathcal{O}(t^{-2})$.*

## 4.4. Rate, not fate

Theorem 4.2 determines the *direction* of learning via $\text{sign}(J)$. For $J > 0$, the *speed* depends on the factor $|J|/\sigma(p)$: smaller $|J|$ slows updates, but trajectories remain in the same basin ($p \to 0$). A convenient way to separate "rate" from "fate" is to change time: let $\tau$ satisfy $d\tau = \eta |J| \, dt / \sigma(p(t))$. Then Equation (5) becomes

$$\frac{dp}{d\tau} = -\text{sign}(J) \, [p(1-p)]^2, \qquad (6)$$

making explicit that verifier noise primarily rescales the clock. Figure 3 visualizes the mean-field solution by plotting $\text{accuracy} = 1 - p(t)$ over time for a sweep of $J = \text{TPR} - \text{FPR}$ and three representative initial conditions $p(0)$. Consistent with Theorem 4.2, the boundary at $J = 0$ separates improvement from degradation, while the transient timescale grows as $J \to 0^+$.

# 5. Beyond Binary: Dynamics on the Simplex

In practice, real-world prompts have multiple good and bad modes. Going beyond the binary arm/mode setup, here we present generalizations of results from the previous section. As described in Section 3, we treat the probability vector coming from our multi-arm bandit as a point of a simplex.

## 5.1. GRPO dynamics modeled by a system of ODEs

The following system generalizes ODE (5), and exhibits a compact mean-field description of the GRPO dynamics.

**Proposition 5.1** (Decoupled multi-mode mean-field ODEs). *Let $\mathbf{p} = ((1-p)\mathbf{y},\ p\mathbf{z})$ with $\mathbf{y} \in \Delta^{K-1}$ and $\mathbf{z} \in \Delta^{M-1}$. Ignoring $\mathcal{O}(\eta^2)$ terms, the mean-field GRPO flow decomposes as*

$$\dot{\mathbf{y}} = +\kappa(p)\,\mathbf{y} \odot \big(\mathbf{y} - \|\mathbf{y}\|_2^2\,\mathbf{1}\big), \tag{7a}$$

$$\dot{\mathbf{z}} = -\kappa(p)\,\mathbf{z} \odot \big(\mathbf{z} - \|\mathbf{z}\|_2^2\,\mathbf{1}\big), \tag{7b}$$

$$\dot{p} = -\eta\,\frac{J}{\sigma(p)}[p(1-p)]^2\,\big(\|\mathbf{y}\|_2^2 + \|\mathbf{z}\|_2^2\big), \tag{7c}$$

*where $\kappa(p) := \eta\,\frac{J}{\sigma(p)}\,p(1-p)$, and $\odot$ denotes the Hadamard product (elementwise product).*

Notice that Youden's index, $J$, is still present, hence phase transition at $J = 0$ as in the binary case. Moreover, going beyond the REINFORCE estimator (1), we point out that the result above (stated as the first order in $\eta$), is valid in the presence of importance sampling and PPO clipping. See Appendix D for a full derivation of the system above in the case of REINFORCE, and Appendix G for modifications coming from PPO sampling and clipping.

Before ending the section, we point out that, as in Section 4, there is a natural time rescaling $\frac{d\tau}{dt} := \kappa(p)$ that completely separates (7a) and (7b), turning them into independent systems on $\Delta^{K-1}$ and $\Delta^{M-1}$ respectively.

*Remark* 5.2. A diligent reader may notice that the last line of system (7) does not reduce to equation (5) when $n = 2$: In the binary case, probability vectors $\mathbf{y}$ and $\mathbf{z}$ become one-dimensional, hence $\|\mathbf{y}\|_2^2 + \|\mathbf{z}\|_2^2 = 2$. This is due to the fact that we modeled the bandit probabilities in the binary (good, bad) case by the logistic function $p = 1/(1 + e^{-z})$ which is slightly different from what the softmax parametrization–on which (7) is based–yields; the bad mass probability based on softmax becomes $p = \frac{e^{\theta_2}}{e^{\theta_1} + e^{\theta_2}}$, hence the logit $z$ will correspond to $\theta_2 - \theta_1$.

## 5.2. Replicator-style flow: GRPO as natural selection

The system (7) is an example of the *replicator dynamics*

$$\dot{p}_i(t) = p_i(t)\Big(f_i\big(\mathbf{p}(t)\big) - \bar{f}\big(\mathbf{p}(t)\big)\Big),\ \ \bar{f}(\mathbf{p}) = \sum_j p_j\,f_j(\mathbf{p}). \tag{8}$$

Here, $f$ is a *fitness* function, $\mathbf{p} = (p_i)_i$ a probability vector, and each type (or strategy) $i$ is rewarded/penalized according to how its fitness compares with the population average.

Geometrically, modeled by (7), GRPO induces a replicator-style flow on the probability simplex. Indeed, when $J > 0$, the mass equation Equation (7c) drives $p \to 0$ (learning), but the shape equations show additional structure: (i) Equation (7a) tends to *concentrate* $\mathbf{y}$ (diversity collapse among good modes), while (ii) Equation (7b) tends to *spread* $\mathbf{z}$ toward uniformity on bad modes. The geometry factor $\|\mathbf{y}\|_2^2 + \|\mathbf{z}\|_2^2 \in [\frac{1}{K} + \frac{1}{M}, 2]$ therefore modulates speed, but cannot change the sign-of-$J$ phase line.

Finally, we point out that the geometry of optimization on the simplex is governed by the softmax Jacobian, $\mathfrak{J}(\mathbf{p}) = \mathrm{Diag}(\mathbf{p}) - \mathbf{p}\mathbf{p}^\top$, which projects updates to the tangent space and corresponds to the inverse Shahshahani/Fisher metric (Shahshahani, 1979; Amari et al., 2019); cf Figure 4. GRPO updates for good arms become *natural-gradient ascent* under this metric scaled by $\kappa(p)$: (7a) may be written as

$$\dot{\mathbf{y}} = \kappa(p)\,\mathrm{grad}_{\mathrm{Shah}}\Phi(\mathbf{y}), \qquad \Phi(\mathbf{y}) := \tfrac{1}{2}\|\mathbf{y}\|_2^2 = \tfrac{1}{2}\sum_{i=1}^K y_i^2.$$

See Appendix I for the details.

## 5.3. Extensions: Learnability, Variance, and KL Regularization

This section collects additional consequences of the mean-field ODE view that help explaining empirically observed training behaviors.

**Learnability peaks at intermediate difficulty.** A natural notion of *per-prompt learnability* is the instantaneous drift magnitude in the bad-mass ODE:

$$\mathcal{L}(p) := \big|\dot{p}\big| \propto \frac{|J|}{\sigma(p)}[p(1-p)]^2 \cdot \big(\|\mathbf{y}\|_2^2 + \|\mathbf{z}\|_2^2\big).$$

In the noiseless regime ($J = 1$, $\sigma(p) = \sqrt{p(1-p)}$), this scales as $\mathcal{L}(p) \propto [p(1-p)]^{3/2}$ and is maximized at $p = 1/2$. Thus, GRPO steps are most effective on *medium-difficulty* prompts where the model is roughly 50–50 between good and bad modes. Under asymmetric noise, the maximizer shifts to an interior $p^\dagger \in (0, 1)$, but it remains intermediate (Appendix E). This aligns with empirical observations that intermediate-difficulty questions are most learnable (Bae et al., 2025; Foster et al., 2025).

**Reward variance and asymmetric noise.** The normalization $\sigma(p) = \sqrt{q(p)(1 - q(p))}$ plays two roles: it stabilizes group advantages, but it also controls convergence rates. If the reward variance is nonzero at the attracting vertex (e.g., $\delta_{\mathrm{FN}} > 0$ when $J > 0$), the bad mass decays as $t^{-1}$; if

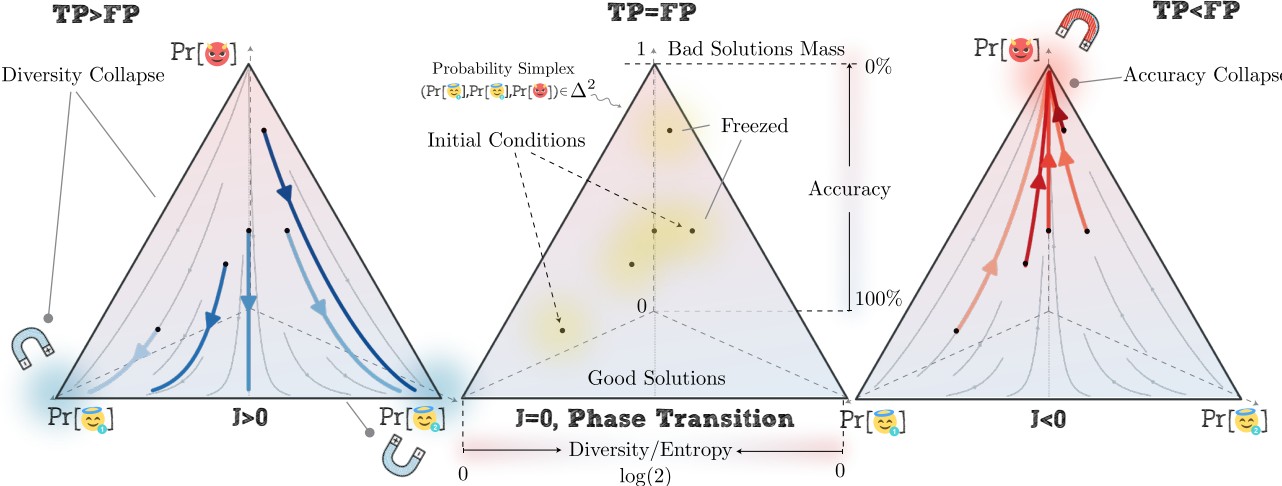

*Figure 2.* **Within-good competition (illustration).** A three-mode mean-field example (two good, one bad). When $J > 0$, bad mass shrinks, but probability within the good set can still concentrate on a single dominant good mode.

the variance vanishes (the clean case $\delta_{\text{FN}} = 0$), the decay accelerates to $t^{-2}$ (Corollary 4.3). Moreover, $\sigma(p)$ depends differently on false positives and false negatives, which can make high-FPR regimes practically more damaging at fixed $J$ (cf. Table 1).

**Continuous or graded rewards.** Although the exposition uses binary rewards for clarity, the same calculation extends to graded verifiers such as LLM judges. Let $r \in [0, 1]$, define $\mu_g := \mathbb{E}[r \mid \text{good}]$, $\mu_b := \mathbb{E}[r \mid \text{bad}]$, and let $\Gamma := \mu_g - \mu_b$ be the class-conditional mean gap. With bad mass $p$ and within-class variances $v_g, v_b$, the population variance is

$$\sigma^2(p) = (1 - p)v_g + pv_b + p(1 - p)\Gamma^2,$$

and z-score normalization gives

$$\mathbb{E}[\tilde{r} \mid \text{good}] - \mathbb{E}[\tilde{r} \mid \text{bad}] = \frac{\Gamma}{\sigma(p)}.$$

Thus the binary phase line $J = 0$ is the special case of a more general mean-gap condition $\Gamma = 0$: a graded verifier helps when its average score is higher on correct modes than on incorrect modes, and anti-learns when the gap is reversed.

**KL regularization smooths the phase transition.** More recent works on GRPO do not include a KL penalty term, arguing that such a term is more suitable for RLHF applications rather than RLVR (Yu et al., 2025; Liu et al., 2025). Nevertheless, earlier works on GRPO (Shao et al., 2024; Mroueh, 2025) contain a KL penalty term. Here, for the sake of completion, we present a generalization of our results to the case of GRPO dynamics with KL penalty. We base our

treatment here on the replicator dynamics which is closely connected to natural gradient on the probability simplex; see Appendix I. In the binary reduction, a forward-KL penalty toward reference bad mass $p_{\text{ref}}$ introduces a restoring drift

$$\dot{p}\big|_{\text{KL}} = -\beta\, p(1 - p)\Big( \text{logit}(p) - \text{logit}(p_{\text{ref}}) \Big),$$

which yields the regularized ODE:

$$\begin{aligned}
\dot{p} = &-\eta\, \frac{J}{\sigma(p)}\, [p(1 - p)]^2\, C(y, z) \\
&- \eta\beta\, p(1 - p)\big( \text{logit}(p) - \text{logit}(p_{\text{ref}}) \big).
\end{aligned} \tag{9}$$

where $C(y, z) := \|\mathbf{y}\|_2^2 + \|\mathbf{z}\|_2^2 \in [\frac{1}{K} + \frac{1}{M}, 2]$. For any $\beta > 0$, the dynamics admit a unique stable interior fixed point $p^\star \in (0, 1)$: for $J > 0$ it satisfies $p^\star < p_{\text{ref}}$, for $J = 0$ it satisfies $p^\star = p_{\text{ref}}$, and for $J < 0$ it satisfies $p^\star > p_{\text{ref}}$. Thus, KL anchoring converts boundary collapse into a controlled interior equilibrium. We refer the reader to Appendix H for more details.

# 6. Experiments

We test whether practical GRPO training exhibits the predicted $J = 0$ phase transition and whether $J > 0$ noise mainly slows convergence.

## 6.1. Setup

**Task.** We use Python code generation with programmatic verification via unit tests. The dataset is a filtered subset of OpenR1 (Hugging Face, 2025) with $N_{\text{train}} = 10{,}239$ prompts and $N_{\text{val}} = 594$ prompts.

**Model and training.** We fine-tune Qwen2.5-3B using

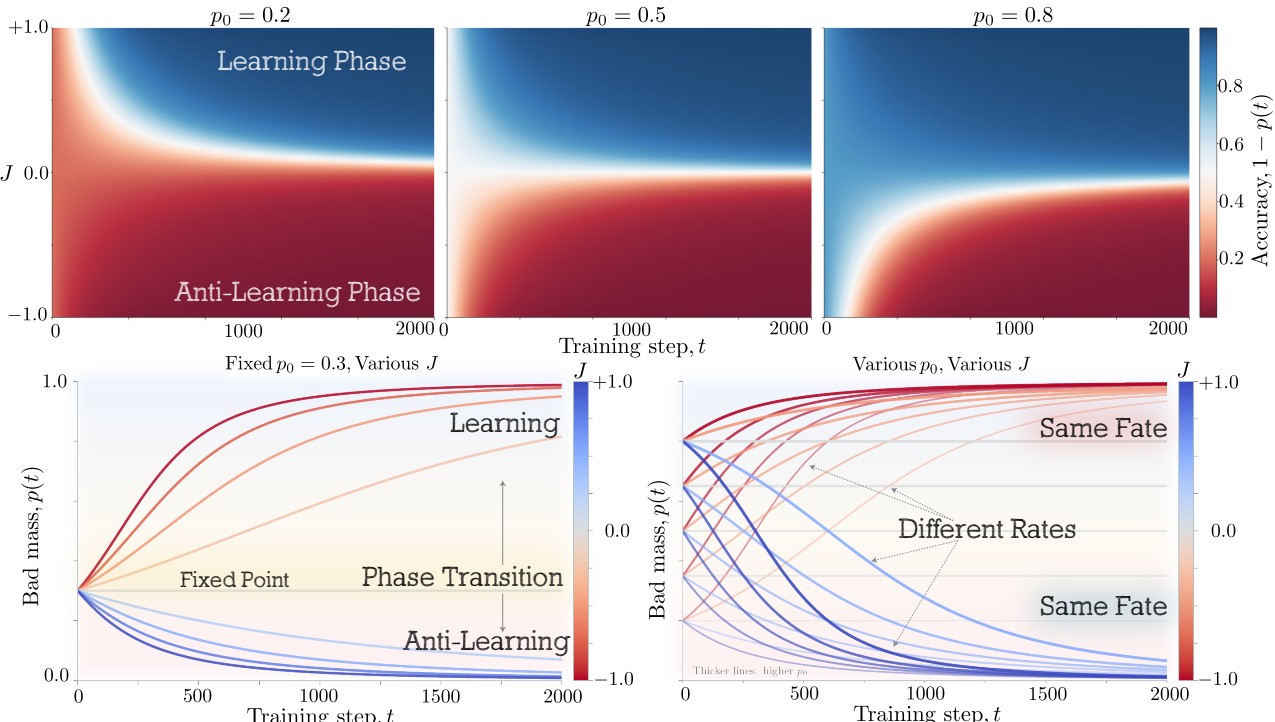

*Figure 3.* **Rate, not fate (mean-field prediction).** After an appropriate time change, trajectories collapse: when $J > 0$, all dynamics converge to $p \to 0$; when $J < 0$, to $p \to 1$. Noise primarily controls the clock.

GRPO implemented in VeRL (Sheng et al., 2024). Unless stated otherwise, we use group size $G = 8$, train for two epochs (1,410 gradient steps), and set the KL penalty coefficient to $\beta = 0$ to isolate reward-driven behavior. We report $\mathbb{E}[\text{pass@1}]$ on the validation set over five random seeds using the *noise-free* unit tests for evaluation. Full hyperparameters are in Appendix M.

**Synthetic verifier noise.** Training rewards are obtained by first computing oracle correctness $z \in \{0, 1\}$ from the unit tests and then flipping it with a noisy checker:

$$r = \begin{cases} 1 \text{ w.p. TPR} & \text{if } z = 1, \\ 1 \text{ w.p. FPR} & \text{if } z = 0, \end{cases} \qquad J = \text{TPR} - \text{FPR}.$$

We sweep $J \in [-0.1, 1]$ using several (TPR, FPR) factorizations at fixed $J$.

### 6.2. Results

**Phase transition at $J = 0$.** Figure 1 shows a clear boundary at $J = 0$: runs with $J > 0$ improve pass@1, $J \approx 0$ exhibits near-neutral behavior, and $J < 0$ degrades, consistent with Theorem 4.2.

**Rate, not fate in practice.** Among $J > 0$ settings, larger $J$ yields faster improvement over the same training horizon, consistent with the "rate" effect in Equations (5) and (6).

At fixed $J = 0.3$, high-FPR configurations are more damaging than high-$\delta_{\text{FN}}$ configurations (Table 1), qualitatively matching the asymmetric-variance behavior in Figure 7.

**Beyond GRPO and code generation.** To check that the transition is not an artifact of the per-completion GRPO abstraction or of programming tasks, we also ran PPO on the math-reasoning benchmark GSM8K using DeepSeek-LLM-7B-chat; see Figure 6. This setting includes token-level credit assignment, unlike critic-free GRPO, yet it exhibits the same qualitative sign-of-$J$ behavior. The positive-$J$ regimes rise from roughly $24\%$ at initialization to about $63$–$64\%$ after two epochs, the near-neutral regime remains nearly flat around $21$–$22\%$, and the negative-$J$ regime collapses to about $1$–$3\%$. Thus, positive verifier alignment leads to learning, near-zero alignment leads to stagnation, and anti-aligned verification leads to anti-learning even when explicit token-level credit assignment is present.

**Interpreting "rate, not fate" in finite training.** The phrase should be read as a mean-field statement about the reward-driven drift: for $J > 0$, noisy verification changes the speed along the same attracting direction. In finite practical runs, group size $G$, training horizon, and nonstationary verifier behavior can still affect how sharply this asymptotic picture is expressed. The experiments therefore support the directional prediction most directly, while larger sensitivity

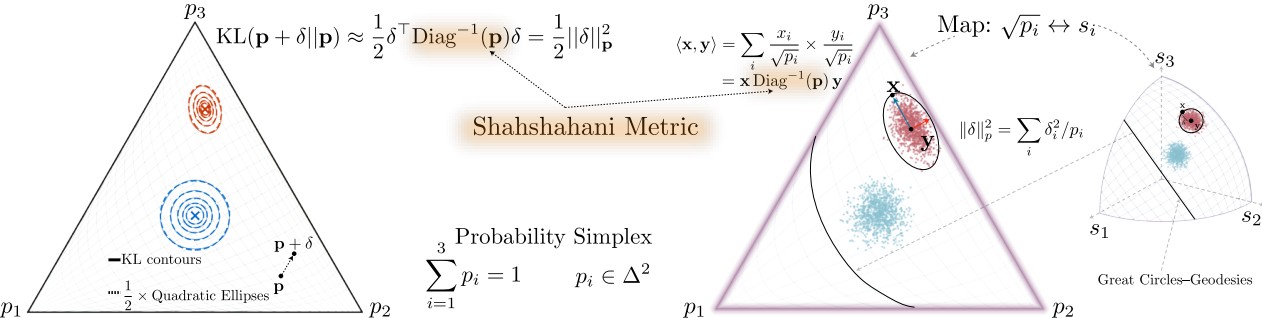

*Figure 4.* **Geometry of the probability simplex.** GRPO induces a mass-conserving replicator flow on $\Delta^{K+M-1}$. The softmax Jacobian $\mathfrak{J}(\mathbf{p})$ projects to the simplex tangent space and corresponds to the inverse Shahshahani/Fisher metric, connecting GRPO to natural-gradient dynamics.

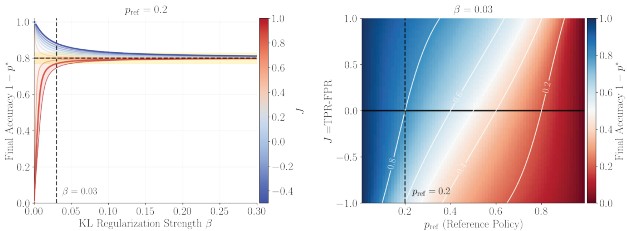

*Figure 5.* **KL regularization yields an interior equilibrium.** Any $\beta > 0$ smooths the $J = 0$ phase transition by balancing reward-driven drift with KL anchoring, producing a stable $p^{\star} \in (0, 1)$ whose location depends on the sign of $J$.

*Table 1.* **Validation after two epochs.** Pass@1 (higher is better) for representative noise regimes. "$\Delta$" is improvement over the initial base model.

| $J$ | (FPR, $\delta_{\mathrm{FN}}$) | Pass@1 | $\Delta$ |
|------|------|------|------|
| $-0.1$ | $(0.60, 0.50)$ | $0.16\%$ | $-12.6\%$ |
| $0.0$ | $(0.50, 0.50)$ | $13.4\%$ | $+0.6\%$ |
| $0.3$ | $(0.00, 0.70)$ | $16.0\%$ | $+3.2\%$ |
| $0.3$ | $(0.70, 0.00)$ | $14.6\%$ | $+1.8\%$ |
| $0.7$ | $(0.20, 0.10)$ | $18.6\%$ | $+5.8\%$ |
| $1.0$ | $(0.00, 0.00)$ | $20.8\%$ | $+8.0\%$ |

sweeps are complementary evidence about rates.

## 7. Limitations and Concluding Remarks

Under GRPO-style group normalization, noisy verifiers induce a sharp and actionable threshold: learning is directionally correct if and only if $J = \mathrm{TPR} - \mathrm{FPR} > 0$. When $J > 0$, noise mainly slows convergence; when $J < 0$, RLVR becomes anti-learning and systematically pushes probability mass toward incorrect modes. We hope this "rate or fate" lens helps practitioners diagnose verifier quality early (e.g., by estimating $\mathrm{TPR}$ and $\mathrm{FPR}$ on held-out data) and guides the design of more robust RLVR pipelines.

**Operationalizing $J$ in practice.** In practice, $J$ can be estimated on a small audited calibration split by comparing verifier decisions against trusted correctness labels and computing $\widehat{\mathrm{TPR}} - \widehat{\mathrm{FPR}}$. For thresholded verifiers, choose the threshold that maximizes this separation, not precision alone. If the confidence interval for $\hat{J}$ overlaps zero, the verifier provides little directional signal and should be improved before relying on RLVR updates.

Our work comes with its own limitations that naturally suggest future research directions.

**Limitations.** Our analysis is intentionally minimal and focuses on a per-prompt mean-field view. (i) The bandit/mode abstraction coarse-grains the sequence space; although it captures how group normalization redistributes probability mass, it ignores parameter sharing across prompts and modes. (ii) We focus on binary rewards and characterize noise via $(\mathrm{TPR}, \mathrm{FPR})$; richer graders (multi-level rewards, preference judges) may introduce additional failure modes. As noted in Section 5.3, the first-order drift extends to graded rewards by replacing $J$ with the class-conditional mean gap $\Gamma$, but richer judges may also introduce prompt-dependent bias, calibration error, or nonstationary behavior not captured by the present model. (iii) Mean-field dynamics approximate the large-group ($G$) limit; finite-group sampling adds stochasticity that can interact with mode collapse and regularization. (iv) Our experiments inject *synthetic* noise and train for a limited horizon; longer training and other domains (e.g., math or judge-based rewards) remain important to test. (v) We set $\beta = 0$ in the main experiments to isolate the reward-driven verifier-noise dynamics. Analytically, adding a KL penalty introduces a restoring drift toward the reference policy, which can stabilize training and replace boundary collapse with a regularized interior equilibrium. However, a strong KL term can also damp the task-driven motion induced by the verifier signal, thereby changing the rate, sharpness, and limiting point of the dy-

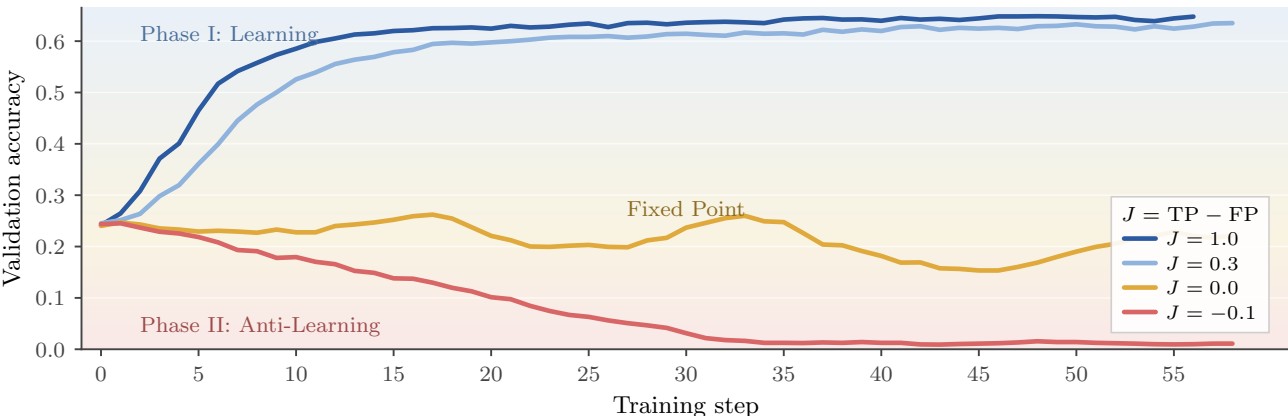

*Figure 6.* **PPO on Math confirms the sign-of-$J$ verifier-noise transition beyond the GRPO bandit abstraction.** Validation accuracy during PPO training of DeepSeek-LLM-7B-chat on GSM8K under different noisy-verifier regimes, where $J = \mathrm{TPR} - \mathrm{FPR}$ is Youden's index. Even though PPO includes token-level credit assignment, the qualitative behavior matches the predicted RLVR phase transition: informative verifiers with $J > 0$ improve accuracy, near-random verifiers with $J \approx 0$ stagnate, and anti-informative verifiers with $J < 0$ induce anti-learning and accuracy collapse. This result supports the view that the sign-of-$J$ behavior is not an artifact of the per-completion mean-field abstraction or of GRPO-specific group normalization, but a robust effect of verifier noise in RLVR.

namics. For this reason, we treat a systematic KL sweep as complementary to the present study and leave it for future work.

**Future directions.** The framework developed here serves two complementary purposes. First, it makes the role of verifier noise explicit, enabling principled choices of training schedules and stopping criteria based on how noise reshapes the mean-field learning dynamics. A natural next step is *noise-aware RLVR*: adaptive data collection, test selection, and active-learning style verification policies that allocate effort where additional signal most improves learnability.

Second, the geometric view of RLVR methods like GRPO as a controlled flow on the probability simplex suggests a path from diagnosis to design. Rather than tuning heuristics, we can *engineer* update rules by targeting desired vector-field properties. An immediate objective is to prevent diversity collapse by constructing advantage-shaping or regularization schemes that induce anti-collision dynamics among correct modes while preserving bad-mass decay.

The flow perspective is also constructive. In simplex coordinates, a desired restoring field toward the uniform distribution over correct modes can be translated, to first order, into an advantage-shaping term. This principle motivates our follow-up work on Geometric GRPO (Rad et al., 2026): vanilla GRPO creates a collision field among correct modes, driving the within-good distribution toward a simplex vertex, or a winner-take-all reasoning rut. G²RPO edits this field with an inverse-probability granularity bonus that upweights underrepresented correct modes, together with a neutrality correction that preserves the bad-mass learning channel. Empirically, this increases correct-mode coverage, prevents the late-stage entropy crash under GRPO, and im-

proves pass@1, showing how mean-field flow can guide algorithm design rather than merely diagnose failures.

More broadly, the framework enables *reverse engineering*: predicting the qualitative behavior of a proposed algorithm from its induced flow before running large-scale experiments.

## Impact Statement

This work studies when reinforcement learning with verifiable rewards remains reliable under imperfect verification (e.g., incomplete unit tests or noisy judges), and when it can systematically fail by reinforcing incorrect behavior. A clearer understanding of these failure modes can help practitioners design verifiers and training procedures that avoid harmful degradation, improving the robustness and reliability of deployed models. At the same time, more reliable RLVR can also accelerate capability gains in domains like code generation; as with other progress in model training, the techniques and insights here could be misused to build more capable systems without commensurate safeguards. We encourage using these results as a diagnostic tool to detect misleading reward signals early and to prioritize verifier improvements and monitoring when applying RLVR in safety-critical settings.

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

## A. Binary bad-mass ODE: convergence rates

This appendix provides a short derivation of Corollary 4.3 for the binary mean-field ODE

$$\dot{p}(t) = -\eta \, \frac{J}{\sigma(p(t))} \left[ p(t)\big(1 - p(t)\big) \right]^2,$$

$$\sigma(p) = \sqrt{q(p)\big(1 - q(p)\big)},$$

$$q(p) = (1 - \delta_{\text{FN}}) - Jp.$$

Assume $J > 0$ and consider the late-time regime where $p(t) \to 0$.

**Case 1: $\delta_{\text{FN}} > 0$.** Then $q(0) = 1 - \delta_{\text{FN}} \in (0, 1)$ and hence $\sigma(p) \to \sigma(0) = \sqrt{(1 - \delta_{\text{FN}})\delta_{\text{FN}}} > 0$. For small $p$, we have $(1 - p)^2 \approx 1$ and thus

$$\dot{p}(t) \approx -\eta \, \frac{J}{\sigma(0)} \, p(t)^2.$$

Solving $\dot{p} = -cp^2$ gives $p(t) = \mathcal{O}(t^{-1})$.

**Case 2: $\delta_{\text{FN}} = 0$.** Then $q(p) = 1 - Jp$ and $1 - q(p) = Jp$, so $\sigma(p) = \sqrt{(1 - Jp)\,Jp} \approx \sqrt{Jp}$ as $p \to 0$. Again using $(1 - p)^2 \approx 1$,

$$\dot{p}(t) \approx -\eta \, \frac{J}{\sqrt{Jp(t)}} \, p(t)^2 = -\eta \, \sqrt{J} \, p(t)^{3/2}.$$

Solving $\dot{p} = -cp^{3/2}$ yields $p(t) = \mathcal{O}(t^{-2})$.

**Remark.** The same local analysis around $p \to 1$ gives analogous polynomial rates for $J < 0$ (anti-learning), with $1 - p(t)$ decaying as $t^{-1}$ or $t^{-2}$ depending on the boundary variance behavior.

## B. LLM as Multi-arm Bandit

### B.1. Multi-armed bandits.

The multi-armed bandit (MAB) problem is a model in optimization and probability that focuses on the exploration–exploitation trade-off. In this problem setup, a decision maker repeatedly selects one of $K$ actions ("arms"); upon pulling arm $a_t \in [K]$ at round $t$, a stochastic reward $R_t(a_t)$ is observed, drawn from an unknown distribution with mean $\mu_{a_t}$. The goal is to maximize cumulative reward or equivalently minimize regret, despite this uncertainty:

$$R_T = T\mu_* - \sum_{t=1}^{T} \mu_{a_t}, \qquad \mu_* := \max_{a \in [K]} \mu_a.$$

This setting captures a wide range of real systems where feedback is noisy, delayed, or partial: online recommendation, A/B testing, adaptive science experiments, and (as

emphasized in this work) coarse-grained evaluation of generative models. Classical algorithms balance information gathering with reward maximization (e.g., optimism/UCB, posterior sampling, or gradient-based updates), and their guarantees hinge on the number of arms, reward signal quality, and the horizon $T$. In our context, the bandit abstraction serves as a tractable surrogate for complex, high-dimensional decision spaces while preserving the essential statistical structure of learning under uncertainty (Lattimore & Szepesvári, 2020).

### B.2. Bandit Abstraction for LLMs

In the context of generative AI, specifically large language models, a given problem (such as a request or prompt) can yield multiple potential solutions, particularly when these models operate with a non-zero temperature setting. Recall that the temperature parameter influences output of the final layer of the model, where it directly affects the selection of subsequent tokens from the logit vectors using the softmax mechanism adjusted by the specified temperature. This selection process at the token level results in the generation of various sequences, some of which are correct (if the domain is verifiable), while others may be incorrect. Although the space of possible sequences that an LLM can generate is theoretically infinite akin to the hypothetical scenario of a monkey randomly typing and eventually producing a proof of the Goldbach conjecture, the total number of answers is finite due to the maximum response length that is feasible for the model to generate these answers.

For a fixed prompt $x$, an LLM samples a completion $y$ from $\pi_\omega(y \mid x)$. With nonzero temperature, the raw support over *all* token sequences can be very large (in principle, unbounded). In practice, inference and training impose a maximum generation length $L_{\max}$ (e.g., max_new_tokens) and an end-of-sequence token $\langle \text{eos} \rangle$. Let $\mathcal{V}$ denote the finite vocabulary. The admissible completions are then drawn from the truncated set

$$\mathcal{Y}_{\leq L_{\max}} = \bigcup_{\ell=1}^{L_{\max}} \mathcal{V}^\ell,$$

$$|\mathcal{Y}_{\leq L_{\max}}| \leq \sum_{\ell=1}^{L_{\max}} |\mathcal{V}|^\ell$$

$$= \frac{|\mathcal{V}|^{L_{\max}+1} - |\mathcal{V}|}{|\mathcal{V}| - 1}.$$

so the effective support is *finite*. (Stop-sequences and $\langle \text{eos} \rangle$ further reduce this set in practice.) Given a fixed prompt $x$, a large language model (LLM) samples an output sequence $y$ from its conditional policy $\pi_\omega(y \mid x)$ (with base parameters $\omega$). In case of truncation, we can write the truncated policy as

$$\pi_\omega^{(L)}(y \mid x) \propto \pi_\omega(y \mid x) \, \mathbf{1}\{y \in \mathcal{Y}_{\leq L_{\max}}\}.$$

## B.3. Coarse-graining into Modes

For a nonzero sampling temperature, the model typically admits many distinct answers to the same prompt, often spanning a very large (potentially infinite) support. However, in the practice, due to the limitation on the output length (controlled by max tokens), the space of possible solution is practically *coarsen* into a finite collection of representatives *reasoning modes* (or solution prototypes). By clustering the reasoning-equivalent response together as a one reasoning mode/arm, we can map outputs via a surjective map

$$\phi: \; \mathcal{Y}_{\leq L_{\max}} \longrightarrow \mathcal{H} \; = \; \{h_1, \ldots, h_{K+M}\},$$

where each mode $h \in \mathcal{H}$ represents a literal or semantic/evaluative equivalence class (e.g., logically equivalent answers, rubric-equivalent or string matching equivalency).

In the next step, we can partition the modes into *good* (correct) and *bad* (incorrect) solutions,

$$\mathcal{H} \; = \; \mathcal{H}^+ \cup \mathcal{H}^-, \qquad |\mathcal{H}^+| = K, \; |\mathcal{H}^-| = M.$$

Without loss of generality, we index good modes by $i \in \{1, \ldots, K\}$ and bad modes by $i \in \{K+1, \ldots, K+M\}$. Sampling a response is now equivalent to pulling one arm from a $(K+M)$-armed bandit with pull probabilities $\pi_\theta(h_i \,|\, x)$. We then work with the induced categorical policy over modes,

$$\pi_\theta(h_i \,|\, x) \; = \; \frac{\exp(\theta_i)}{\sum_{j=1}^{K+M} \exp(\theta_j)} \; = \; \text{softmax}(\theta)_i,$$

where $\theta = (\theta_1, \ldots, \theta_{K+M})$ are *effective logits* that summarize, for the fixed prompt $x$, the aggregate probability mass the base model places on each mode. These logits are not a one-to-one reparameterization of $\omega$; rather, they are low-dimensional coordinates (unique up to an additive constant) on the probability simplex over $\mathcal{H}$ induced by $\pi_\omega(\cdot \,|\, x)$.

In this work, since we are interested mostly in the total probability of bad arms, as we discussed in the Section 3, we define the bad arms mass probability by partitioning modes into good (correct) and bad (incorrect), $\mathcal{H} = \mathcal{H}^+ \cup \mathcal{H}^-, \; |\mathcal{H}^+| = K, \; |\mathcal{H}^-| = M$, such that $p = \sum_{h \in \mathcal{H}^-} \pi_\theta(h \,|\, x)$

## C. Noisy Rewards and Youden's $J$ Index

Recall the definition of noise that we had in:

$$\delta_{\text{FN}} = \Pr(r = 0 \,|\, \text{good}), \qquad \delta_{\text{FP}} = \Pr(r = 1 \,|\, \text{bad}),$$

and the Youden's Index, as

$$J := 1 - \delta_{\text{FN}} - \delta_{\text{FP}} \; = \; \text{TPR} - \text{FPR} \in [-1, 1].$$

where $p = \Pr(\text{bad})$ denote the current bad mass (so $\Pr(\text{good}) = 1 - p$). With this setup, the expected reward of a single pull is

$$\begin{aligned}
q(p) &:= \; \mathbb{E}[r] \\
&= \; \mathbb{E}[r \,|\, \text{good}]\,(1-p) + \mathbb{E}[r \,|\, \text{bad}]\,p \\
&= \; (1-p)\,(1-\delta_{\text{FN}}) + p\,\delta_{\text{FP}} \\
&= \; (1 - \delta_{\text{FN}}) \; - \; J\,p.
\end{aligned} \tag{10}$$

Since $r$ is Bernoulli with mean $q$, its variance is

$$\sigma^2(p) \; := \; \text{Var}(r) \; = \; q(p)\big(1 - q(p)\big). \tag{11}$$

we can also directly verify this property:

$$\begin{aligned}
\text{Var}(r) &= \; (1-p)(1-\delta_{\text{FN}})\,\delta_{\text{FN}} \\
&\quad + p\,\delta_{\text{FP}}(1-\delta_{\text{FP}}) + p(1-p)\,J^2 \\
&= \; \big((1-\delta_{\text{FN}}) - Jp\big)\big(\delta_{\text{FN}} + Jp\big) \\
&= \; q(1-q).
\end{aligned} \tag{12}$$

Notice that $q, \sigma(p) \equiv q, \sigma(\delta_{\text{FN}}, \delta_{FP}, p)$, but for brevity, we denote it as $\sigma(p)$ and $q(p)$.

It is good to notice that since $q(p) \in [\delta_{\text{FP}}, 1 - \delta_{\text{FN}}]$, the variance term $\sigma(p) = \sqrt{q(p)\big(1 - q(p)\big)}$ is maximized at $q(p) = \frac{1}{2}$, provided $\frac{1}{2}$ lies inside this interval. Solving $q(p) = \frac{1}{2}$ for $p$ yields

$$p^\star = \text{clip}\big(\frac{\frac{1}{2} - \delta_{\text{FN}}}{J}, 0, 1\big)$$

such that the value of $p$ that maximizes $\sigma(p)$, assuming $J = 1 - \delta_{\text{FN}} - \delta_{\text{FP}} > 0$. In the edge case where $\frac{1}{2} \notin [\delta_{\text{FP}}, 1 - \delta_{\text{FN}}]$ (i.e., for extremely noisy graders), the maximum of $\sigma(p)$ occurs at the boundary: $q = \delta_{\text{FP}}$ or $q = 1 - \delta_{\text{FN}}$, whichever is closer to $\frac{1}{2}$. Equivalently, $p^\star$ clips to 0 or 1 in this regime (see Fig. 7).

Group-based policy typically updates normalized rewards within a prompt-specific group of $G$ rollouts. While alternatives exist (e.g., leave-one-out baselines (Kool et al., 2019) or centered-but-unstandardized variants (Liu et al., 2025)), we adopt a simple $z$-score normalization (as in GRPO (Shao et al., 2024)):

$$\tilde{r} \; = \; \frac{r - q(p)}{\sigma(p)}. \tag{13}$$

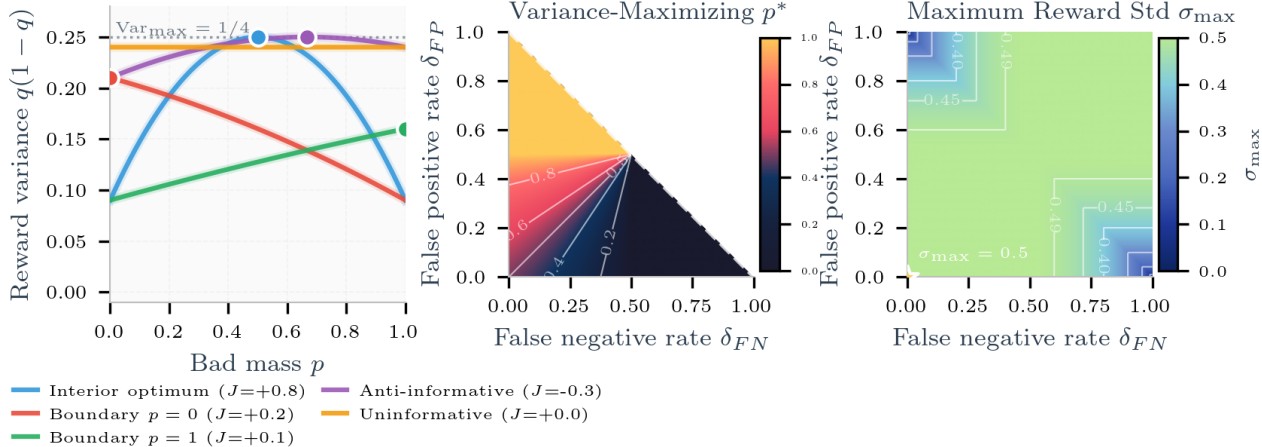

*Figure 7.* **Reward-variance geometry under noisy Bernoulli rewards.** (Left) Reward variance $\mathrm{Var}(r) = q(p)\big(1 - q(p)\big)$ as a function of bad mass $p$ for representative noise settings; markers indicate $p^\star = \arg\max_{p\in[0,1]} \mathrm{Var}(r)$ (equivalently $q(p) = \frac{1}{2}$ when attainable, otherwise the boundary $p \in \{0,1\}$). (Middle) Heatmap of $p^\star(\delta_{\mathrm{FN}}, \delta_{\mathrm{FP}})$ in the informative region $J > 0$, with the dashed diagonal marking the phase boundary $J = 1 - \delta_{\mathrm{FN}} - \delta_{\mathrm{FP}} = 0$ and contours showing level sets of $p^\star$. (Right) Maximum achievable reward standard deviation $\sigma_{\max}(\delta_{\mathrm{FN}}, \delta_{\mathrm{FP}}) = \max_{p\in[0,1]} \sqrt{q(p)(1 - q(p))}$ with contours. Throughout, $q(p) = (1 - \delta_{\mathrm{FN}}) - Jp$ and $J = 1 - \delta_{\mathrm{FN}} - \delta_{\mathrm{FP}}$.

Conditioning on the latent correctness, this yields

$$
\tilde{r} \mid \text{good} = \begin{cases} \dfrac{1-q}{\sigma}, & \text{w.p. } 1 - \delta_{\mathrm{FN}}, \\[2mm] \dfrac{-q}{\sigma}, & \text{w.p. } \delta_{\mathrm{FN}}, \end{cases}
$$

$$
\tilde{r} \mid \text{bad} = \begin{cases} \dfrac{1-q}{\sigma}, & \text{w.p. } \delta_{\mathrm{FP}}, \\[2mm] \dfrac{-q}{\sigma}, & \text{w.p. } 1 - \delta_{\mathrm{FP}}. \end{cases}
$$

Taking expectations gives the block-symmetric conditional means

$$
\mathbb{E}[\tilde{r} \mid \text{good}] = \frac{Jp}{\sigma(p)}, \qquad \mathbb{E}[\tilde{r} \mid \text{bad}] = -\frac{J(1-p)}{\sigma(p)}, \tag{14}
$$

and global centering holds automatically:

$$
\mathbb{E}[\tilde{r}] = (1-p)\,\mathbb{E}[\tilde{r} \mid \text{good}] + p\,\mathbb{E}[\tilde{r} \mid \text{bad}] = 0, \tag{15}
$$

which is desirable for stable, scale-invariant updates. These expressions demonstrate that Youden's index $J$ governs the sign and magnitude of the expected normalized reward for good versus bad arms.

*Remark* C.1. If one omits division by $\sigma(p)$ (the "centered-only" modification of GRPO (Liu et al., 2025)), then

$$
\begin{aligned} \mathbb{E}[\tilde{r} \mid \text{good}] &= Jp, \\ \mathbb{E}[\tilde{r} \mid \text{bad}] &= -J(1-p), \tag{16} \\ \mathbb{E}[\tilde{r}] &= 0. \end{aligned}
$$

*Remark* C.2. Some works use a $\{\pm 1\}$-valued reward $S := 2r - 1$. Then $\mathbb{E}[S] = 2q - 1$ and $\mathrm{Var}(S) = 4q(1-q)$.

Equations (13)–(14) map to this reward by the linear rescaling $S = 2r - 1$; the resulting normalization differs only by a constant factor of 2.

*Remark* C.3. For noise-free case, $J = 1$, the expectation values take a simpler form

$$
\mathbb{E}[\tilde{r} \mid \text{good}] = \sqrt{\frac{p}{1-p}},
$$

$$
\mathbb{E}[\tilde{r} \mid \text{bad}] = \sqrt{\frac{1-p}{p}},
$$

$$
\mathbb{E}[\tilde{r} \mid \text{good}] - \mathbb{E}[\tilde{r} \mid \text{bad}] = \frac{1}{\sqrt{p(1-p)}}.
$$

# D. Mean Field Dynamics.

In this section, we analyze the evolution of good and bad arms using a mean-field approximation of the multi-armed bandit (MAB) system (see Appendix B). Our derivation accounts for a general noisy environment where $J \in [-1, 1]$. The noise-free scenario is treated as a specialization of this framework; specifically, by setting $J = 1$, we recover the standard clean-reward dynamics without requiring further modification.

## D.1. Dynamics of the Bad Arms

Consider a $(K+M)$-arm bandit comprising $K$ *good* arms and $M$ *bad* arms $\{b_1, \ldots, b_M\}$. Let the policy be defined as $\mathbf{p} = \text{softmax}(\boldsymbol{\theta}) \in \Delta^{K+M-1}$ where:

$$\mathbf{p} = (p_1, \ldots, p_K, \, p_{b_1}, \ldots, p_{b_M}),$$
$$p := \sum_{m=1}^{M} p_{b_m} \in [0, 1],$$
$$\alpha := 1 - p.$$

and we define the normalized within-block coordinates:

$$p_j = \alpha \, y_j, \qquad j \leq K,$$
$$\mathbf{y} = (y_1, \ldots, y_K) \in \Delta^{K-1},$$
$$p_{b_m} = p \, z_m, \qquad m \leq M,$$
$$\mathbf{z} = (z_1, \ldots, z_M) \in \Delta^{M-1}.$$

Equivalently, the probability vector factors as follows:

$$\mathbf{p} = \begin{bmatrix} \alpha \, \mathbf{y} \\ p \, \mathbf{z} \end{bmatrix},$$
$$\|\mathbf{y}\|_2^2 = \sum_{j=1}^{K} y_j^2 \in \left[ \tfrac{1}{K}, 1 \right],$$
$$\|\mathbf{z}\|_2^2 = \sum_{m=1}^{M} z_m^2 \in \left[ \tfrac{1}{M}, 1 \right].$$

Let the conditional expected advantage reward be denoted by $A_i := \mathbb{E}[\tilde{r} \mid I = i]$, with $\mathbf{A} = (A_1, \ldots, A_K, A_{b_1}, \ldots, A_{b_M})$ and the mean advantage $\bar{A} := \langle \mathbf{p}, \mathbf{A} \rangle = \sum_i p_i A_i$. We assume *block symmetry* within both blocks, a condition that arises naturally in the binary-reward scenario analyzed in Appendix C (refer to (14)):

$$A_j = a_{\text{g}}(p), \qquad j \leq K,$$
$$A_{b_m} = a_{\text{b}}(p), \qquad m \leq M, \qquad (17)$$
$$\Delta r(p) := a_{\text{b}}(p) - a_{\text{g}}(p).$$

It follows that $\bar{A} = \alpha \, a_{\text{g}}(p) + p \, a_{\text{b}}(p)$, which implies $a_{\text{g}}(p) - \bar{A} = -p \, \Delta r(p)$ and $a_{\text{b}}(p) - \bar{A} = \alpha \, \Delta r(p)$.

**Proposition D.1** (Expected directions in $\boldsymbol{\theta}$ and $\mathbf{p}$ space). *Given $\mathbf{p} = \text{softmax}(\boldsymbol{\theta})$ and the softmax Jacobian $\mathfrak{J}(\mathbf{p}) := \text{Diag}(\mathbf{p}) - \mathbf{p}\mathbf{p}^\top$, for a step size $\eta$ and group size $G$:*

$$\mathbb{E}[\Delta \boldsymbol{\theta} \mid \mathbf{p}] = \eta \, \mathfrak{J}(\mathbf{p}) \, \mathbf{A},$$
$$\mathbb{E}[\Delta \mathbf{p} \mid \mathbf{p}] \approx \mathfrak{J}(\mathbf{p}) \, \mathbb{E}[\Delta \boldsymbol{\theta} \mid \mathbf{p}]$$
$$= \eta \, \mathfrak{J}(\mathbf{p})^2 \, \mathbf{A}.$$

*to the first order in $\Delta \boldsymbol{\theta}$.*

*Proof Sketch.* Let $e_i$ denote the $i$-th standard basis vector. Define $\pi_\theta(i) = p_i = \exp(\theta_i)/\sum_k \exp(\theta_k)$ as the softmax policy. For a realized arm $I$, the gradient is:

$$\nabla_{\boldsymbol{\theta}} \log \pi_\theta(I) = e_I - \mathbf{p}.$$

The REINFORCE estimator ((Williams, 1992)) for $\nabla_{\boldsymbol{\theta}} \mathbb{E}[r]$ is $g = \tilde{r} \, (e_I - \mathbf{p})$. Taking the conditional expectation given $\mathbf{p}$ yields:

$$\mathbb{E}[g \mid \mathbf{p}] = \sum_i p_i \, \mathbb{E}[\tilde{r} \mid I = i] \, (e_i - \mathbf{p})$$
$$= \left( \text{Diag}(\mathbf{p}) - \mathbf{p}\mathbf{p}^\top \right) \mathbf{A}$$
$$= \mathfrak{J}(\mathbf{p}) \, \mathbf{A}.$$

This confirms the stated form and the first identity in Proposition D.1. For the second part, refer to Lemma I.1.

*Remark* D.2 (The Importance of Coupling Terms). Retaining the full Jacobian, including the rank-one term $\mathbf{p}\mathbf{p}^\top$, is essential because it couples all arms through collision terms. Specifically, the total bad-mass drift depends on the collisions within both the good and bad blocks via $\|\mathbf{y}\|_2^2$ and $\|\mathbf{z}\|_2^2$ (see (24)). Omitting the $\mathbf{p}\mathbf{p}^\top$ term would spuriously decouple the blocks and result in incomplete mean-field dynamics.

**Corollary D.3** (First-order softmax pushforward). *For a small logit increment $\Delta \boldsymbol{\theta}$:*

$$\Delta \mathbf{p} = \mathfrak{J}(\mathbf{p}) \, \Delta \boldsymbol{\theta}$$
$$= \text{Diag}(\mathbf{p}) \, \Delta \boldsymbol{\theta} - \mathbf{p} \, \mu,$$
$$\mu := \langle \mathbf{p}, \Delta \boldsymbol{\theta} \rangle = \sum_k p_k \, \Delta \theta_k.$$

*which implies $\Delta p_i = p_i \left( \Delta \theta_i - \mu \right)$.*

Applying Proposition D.1 and the relation for $\bar{A}$, we obtain the following expectations (where conditioning on $\mathbf{p}$ is suppressed for brevity):

$$\mathbb{E}[\Delta \theta_j] = -\eta \, p(1-p) \, \Delta r(p) \, y_j, \qquad j = 1, \ldots, K, \qquad (18)$$

$$\mathbb{E}[\Delta \theta_{b_m}] = \eta \, p(1-p) \, \Delta r(p) \, z_m, \qquad m = 1, \ldots, M. \qquad (19)$$

The expected step therefore follows the block-form direction:

$$\mathbb{E}[\Delta\boldsymbol{\theta}] \;=\; \eta\,p(1-p)\,\Delta r(p) \begin{bmatrix} -\mathbf{y} \\ \mathbf{z} \end{bmatrix}. \qquad (20)$$

Since $\mathfrak{J}(\mathbf{p})\mathbf{1} = 0$, the update is centered:

$$\sum_i \mathbb{E}[\Delta\theta_i] = \eta\,\mathbf{1}^\top \mathfrak{J}(\mathbf{p})\mathbf{A} = 0. \qquad (21)$$

Moreover, within each block the logit increment is collinear with the current within-block distribution:

$$\mathbb{E}[\Delta\theta_j] - y_j \sum_{k=1}^K \mathbb{E}[\Delta\theta_k] = 0,\; \mathbb{E}[\Delta\theta_{b_m}] - z_m \sum_{\ell=1}^M \mathbb{E}[\Delta\theta_{b_\ell}] = 0.$$

In other words, there is no arm-specific drift *within* a block in logit space; arms move in lockstep proportional to $\mathbf{y}$ (good block) and $\mathbf{z}$ (bad block).

Following (14), the expected advantages relative to the noise level $J = \text{TPR} - \text{FPR}$ are expressed as:

$$\begin{aligned} a_{\text{g}}(p) &= \frac{Jp}{\sigma(p)}, \\ a_{\text{b}}(p) &= -\frac{J(1-p)}{\sigma(p)}, \\ \Delta r(p) &= -\frac{J}{\sigma(p)}. \end{aligned} \qquad (22)$$

**Total Bad-Mass Drift**  By Corollary D.3, the softmax-centering scalar $\mu$ becomes:

$$\mu = \eta\,p(1-p)\,\Delta r(p)\left(p\,\|\mathbf{z}\|_2^2 - (1-p)\,\|\mathbf{y}\|_2^2\right). \quad (23)$$

Summing the bad components provides the total bad-mass drift:

$$\mathbb{E}[\Delta p] = -\eta\,\frac{J}{\sigma(p)}\,[p(1-p)]^2\left(\|\mathbf{y}\|_2^2 + \|\mathbf{z}\|_2^2\right). \quad (24)$$

In the case where $M = 1$, then $\mathbf{z} = (1)$ and $\|\mathbf{z}\|_2^2 = 1$, which recovers the $(K+1)$ formula.

**Within-Bad Dynamics in Normalized Coordinates**  Using the identity $\Delta z_m = \frac{1}{p}(\Delta p_{b_m} - z_m\,\Delta p)$ and substituting the first-order drift, we find:

$$\mathbb{E}[\Delta z_m] = -\eta\,\frac{J}{\sigma(p)}\,p(1-p)\,z_m\left(z_m - \|\mathbf{z}\|_2^2\right), \qquad (25)$$
$$m = 1, \ldots, M.$$

In vector form, this is expressed as

$$\mathbb{E}[\Delta\mathbf{z}] = \eta\,p(1-p)\,\Delta r(p)\left(\mathbf{z}\odot\mathbf{z} - \|\mathbf{z}\|_2^2\,\mathbf{z}\right).$$

Consequently, for an informative grader ($J > 0$), the bad-block dynamics exhibit the opposite sign of the good-block collision field, tending to spread bad mass toward a uniform distribution on $\Delta^{M-1}$.

## D.2. Dynamics of the Good Arms

Regarding the good arms, the combination of Corollary D.3 with (18) and (23) provides the probability increments for the good block:

$$\begin{aligned} \mathbb{E}[\Delta\mathbf{p}_{\text{good}}] = & -\eta\,p(1-p)^2\,\Delta r(p) \\ & \times\left(\mathbf{y}\odot\mathbf{y} + [p\,\|\mathbf{z}\|_2^2 - (1-p)\,\|\mathbf{y}\|_2^2]\mathbf{y}\right). \end{aligned} \qquad (26)$$

In componentwise form, substituting $p_j = (1-p)y_j$, we obtain:

$$\begin{aligned} \mathbb{E}[\Delta p_j] = & (1-p)\,y_j\left(\mathbb{E}[\Delta\theta_j] - \mu\right) \\ = & -\eta\,p(1-p)^2\,\Delta r(p)\,y_j\left(y_j + p\,\|\mathbf{z}\|_2^2 - (1-p)\,\|\mathbf{y}\|_2^2\right), \end{aligned} \qquad (27)$$

$$j = 1, \ldots, K.$$

Summing (27) over all $j$ and applying the constraint $\sum_j y_j = 1$ yields:

$$\begin{aligned} \sum_{j=1}^K \mathbb{E}[\Delta p_j] = & -\eta\,[p(1-p)]^2\,\Delta r(p)\left(\|\mathbf{y}\|_2^2 + \|\mathbf{z}\|_2^2\right) \\ = & -\mathbb{E}[\Delta p]. \end{aligned}$$

Consequently, the drift for the total good mass $\alpha := 1 - p$ is given by:

$$\mathbb{E}[\Delta\alpha] = -\mathbb{E}[\Delta p] = -\eta\,[p(1-p)]^2\,\Delta r(p)\left(\|\mathbf{y}\|_2^2 + \|\mathbf{z}\|_2^2\right). \qquad (28)$$

By utilizing the relationship $y_j = p_j/\alpha$, we can apply the exact identity:

$$\Delta y_j = \frac{1}{\alpha}\left(\Delta p_j - y_j\,\Delta\alpha\right). \qquad (29)$$

Substituting (27) through (28) into (29) and simplifying leads to the within-good drift:

$$\mathbb{E}[\Delta y_j] = -\eta\,p\,\alpha\,\Delta r(p)\,y_j\left(y_j - \|\mathbf{y}\|_2^2\right), j = 1, \ldots, K. \qquad (30)$$

In vector form, this is expressed as:

$$\mathbb{E}[\Delta\mathbf{y}] = -\eta\,p(1-p)\,\Delta r(p)\left(\mathbf{y}\odot\mathbf{y} - \|\mathbf{y}\|_2^2\,\mathbf{y}\right). \qquad (31)$$

Notably, $\sum_j \mathbb{E}[\Delta y_j] = 0$, which confirms that the simplex remains invariant as expected. The fixed points of (31) are located at the barycenter and the vertices. When $\Delta r(p) < 0$, a condition signifying an informative grader that favors good arms over bad arms, the uniform point becomes unstable and the vertices act as attractors.

Substituting $\Delta r(p) = -J/\sigma(p)$ from (22) into (30) results in:

$$\mathbb{E}[\Delta y_j] \;=\; \eta \, \frac{J}{\sigma(p)} \, p(1-p) \, y_j \left( y_j - \|\mathbf{y}\|_2^2 \right).$$

For $J > 0$, arms where $y_j > \|\mathbf{y}\|_2^2$ will grow while those where $y_j < \|\mathbf{y}\|_2^2$ shrink, representing a deterministic sharpening within the good block.

### D.3. From Expectation-Based Updates to ODEs: The Small-Step Bridge

Consider an expectation-level logit update:

$$\boldsymbol{\theta}^{(t+1)} \;=\; \boldsymbol{\theta}^{(t)} + \eta \, g(\mathbf{p}^{(t)}), \qquad \Delta\boldsymbol{\theta} \;=\; \eta \, g(\mathbf{p}),$$

where $g$ represents the per-step expected gradient. Through the softmax mapping $\mathbf{p} = \mathrm{softmax}(\boldsymbol{\theta})$, a small logit update is defined as $\Delta\mathbf{p} = \mathfrak{J}(\mathbf{p})\,\Delta\theta$, with $\mathfrak{J}(\mathbf{p}) = \mathrm{Diag}(\mathbf{p}) - \mathbf{p}\mathbf{p}^\top$. Substituting $\Delta\theta = \eta \, g(\mathbf{p})$ yields the expected increment for the policy:

$$\Delta\mathbf{p} \;=\; \eta \left( \mathrm{Diag}(\mathbf{p}) - \mathbf{p}\mathbf{p}^\top \right) g(\mathbf{p}) \;+\; O(\eta^2).$$

**Option 1: Unit Time per Iteration** We may treat the iteration index itself as continuous time. Let $t \in \mathbb{R}$ denote a continuous extension of the discrete counter, where a single algorithmic update corresponds to a unit time step $\Delta t = 1$. By defining $\mathbf{p}(t) \approx \mathbf{p}^{(t)}$, the relationship is:

$$\frac{\mathbf{p}^{(t+1)} - \mathbf{p}^{(t)}}{\Delta t} \;\approx\; \frac{d\mathbf{p}}{dt}(t) \;=\; \dot{\mathbf{p}}(t).$$

Aligning this with the discrete update $\mathbf{p}^{(t+1)} - \mathbf{p}^{(t)} = \Delta\mathbf{p}$ results in the following ordinary differential equation (ODE):

$$\dot{\mathbf{p}}(t) \;=\; \eta \left( \mathrm{Diag}(\mathbf{p}(t)) - \mathbf{p}(t)\mathbf{p}(t)^\top \right) g(\mathbf{p}(t)). \quad (32)$$

The expectation-level GRPO update thus serves as a forward-Euler discretization of the continuous-time dynamics in (32) with a unit step size.

This ODE provides an accurate proxy within the small-learning-rate regime. The local truncation error of the Euler step satisfies $\|\mathbf{p}^+ - \mathbf{p} - \dot{\mathbf{p}}\| = O(\eta^2)$, and given that $\max_a \eta \, |g_a(\mathbf{p})| \ll 1$, no coordinate of $\mathbf{p}$ shifts excessively in a single iteration. Geometrically, (32) remains a natural-gradient (Shahshahani) flow:

$$\dot{\mathbf{p}} \;=\; \eta \, \mathbf{G}(\mathbf{p}) \, \nabla_\theta \mathcal{L}, \qquad \mathbf{G}(\mathbf{p}) \;=\; \mathrm{Diag}(\mathbf{p}) - \mathbf{p}\mathbf{p}^\top,$$

where $\mathbf{G}(\mathbf{p})$ represents the Fisher metric tensor on the simplex. The factor $\eta$ scales the velocity along this geometric flow. By approximating discrete differences with derivatives,

(31) and (32) transform into the coupled ODEs:

$$\dot{\mathbf{y}} = \kappa(p) \left( \mathbf{y} \odot \mathbf{y} - \|\mathbf{y}\|_2^2 \mathbf{y} \right),$$
$$\dot{\mathbf{z}} = -\kappa(p) \left( \mathbf{z} \odot \mathbf{z} - \|\mathbf{z}\|_2^2 \mathbf{z} \right),$$
$$\dot{p} = -\eta \frac{J}{\sigma(p)} [p(1-p)]^2 \left( \|\mathbf{y}\|_2^2 + \|\mathbf{z}\|_2^2 \right),$$

where we define the proportionality factor $\kappa(p) := \eta \frac{J}{\sigma(p)} p(1-p)$.

**Option 2: Alternative Time Rescaling** An alternative approach involves absorbing the learning rate directly into the time variable. By defining a rescaled time $\mathsf{t} = \eta t$, each discrete update advances $\mathsf{t}$ by $\Delta\mathsf{t} = \eta$. Using the chain rule for $\mathbf{p}(\mathsf{t}) := \mathbf{p}(t)$, we find:

$$\frac{d\mathbf{p}}{d\mathsf{t}} \;=\; \frac{1}{\eta} \frac{d\mathbf{p}}{dt} \;=\; \left( \mathrm{Diag}(\mathbf{p}) - \mathbf{p}\mathbf{p}^\top \right) g(\mathbf{p}),$$

which simplifies (32) to:

$$\frac{d\mathbf{p}}{d\mathsf{t}} \;=\; \left( \mathrm{Diag}(\mathbf{p}(\mathsf{t})) - \mathbf{p}(\mathsf{t})\mathbf{p}(\mathsf{t})^\top \right) g(\mathbf{p}(\mathsf{t})). \quad (34)$$

This represents the standard gradient-flow limit.

*Remark* D.4. While the trajectories in policy space remain identical across both time parametrizations, this work utilizes the unit time per iteration notation to maintain the visibility of mean-field correction terms as they relate to $\eta$.

## E. Maximal Learnability

We quantify a prompt's *learnability* by the instantaneous rate at which GRPO reduces its latent bad-mode mass $p = \Pr(\mathrm{bad} \mid x)$. Under the block-symmetric mean-field approximation derived in Appendix D (see (24), the (unregularized) one-step drift of $p$ takes the form)

$$|\Delta p| \;\propto\; \Delta(p) \, [p(1-p)]^2, \Delta(p) := \mathbb{E}[\tilde{r} \mid \mathrm{good}] - \mathbb{E}[\tilde{r} \mid \mathrm{bad}], \quad (35)$$

up to an overall positive step-size constant and smooth factors that vary slowly with $p$. Here $\tilde{r}$ denotes the group-normalized reward.

**Normalized separation under noisy rewards.** With $z$-score normalization (13), the conditional means (14) imply a simple closed form for the separation in normalized units:

$$\begin{aligned}
\Delta(p) &= \frac{J}{\sigma(p)}, \\
J &= 1 - \delta_{\mathrm{FN}} - \delta_{\mathrm{FP}}, \\
\sigma(p) &= \sqrt{q(p)\left(1 - q(p)\right)}, \\
q(p) &= (1 - \delta_{\mathrm{FN}}) - Jp.
\end{aligned} \quad (36)$$

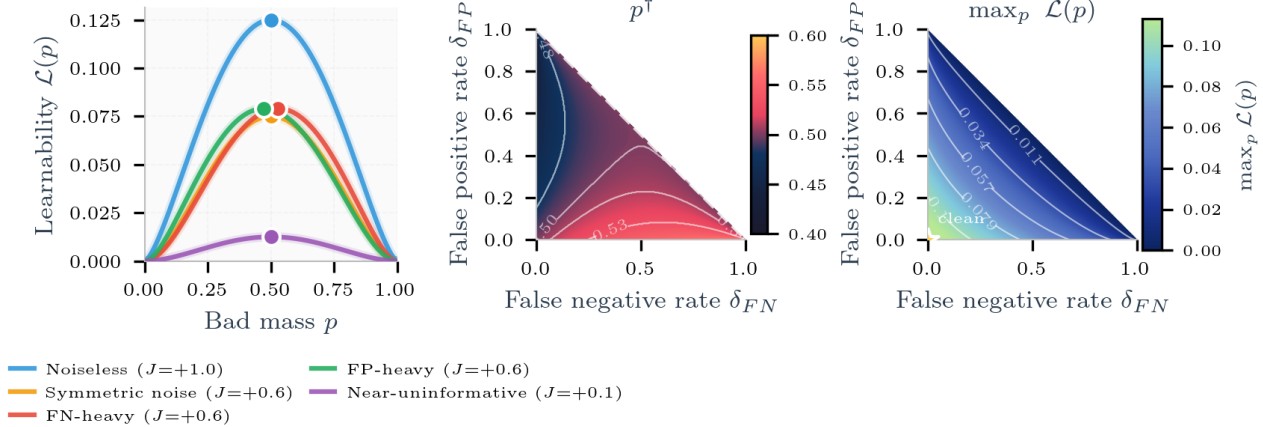

*Figure 8.* **Learnability-maximizing bad mass $p^\dagger$ under group $z$-scored rewards.** We plot the *instantaneous learnability speed* $\mathcal{L}(p) = \frac{J}{\sigma(p)}[p(1-p)]^2$, which controls the magnitude of the one-step bad-mass drift $|\Delta p| \propto \mathcal{L}(p)$ (up to a positive constant), where $\sigma(p) = \sqrt{q(p)(1-q(p))}$, $q(p) = (1-\delta_{\text{FN}}) - Jp$, and $J = 1 - \delta_{\text{FN}} - \delta_{\text{FP}}$. (A) Curves of $\mathcal{L}(p)$ versus $p$ for representative noise settings; markers indicate $p^\dagger = \arg\max_{p \in [0,1]} \mathcal{L}(p)$. (B) Heatmap of $p^\dagger(\delta_{\text{FN}}, \delta_{\text{FP}})$ in the informative region $J > 0$ (masked for $J \leq 0$); the dashed diagonal marks $J = 0$ and contours indicate level sets of $p^\dagger$. (C) Heatmap of the maximum instantaneous learnability $\max_p \mathcal{L}(p)$, showing how noise reduces the peak achievable drift. Symmetric noise ($\delta_{\text{FN}} = \delta_{\text{FP}}$) preserves the midpoint optimum $p^\dagger = \frac{1}{2}$, while asymmetric noise shifts the maximizer away from $\frac{1}{2}$ by reweighting the signal through $\sigma(p)^{-1}$.

Consequently, the *learnability speed* (i.e., the $p$-dependent component of the drift) is

$$\mathcal{L}(p; \delta_{\text{FN}}, \delta_{\text{FP}}) := \frac{J}{\sigma(p)}[p(1-p)]^2,$$

$$|\Delta p| \propto \mathcal{L}(p; \delta_{\text{FN}}, \delta_{\text{FP}}). \tag{37}$$

Throughout this section we focus on the *informative* regime $J > 0$ (the grader is better than random). When $J = 0$ the signal vanishes, and when $J < 0$ it is anti-informative and must be corrected (Remark E.1).

**Noiseless case ($J = 1$).** When $\delta_{\text{FN}} = \delta_{\text{FP}} = 0$, we have $q(p) = 1 - p$ and $\sigma(p) = \sqrt{p(1-p)}$, hence

$$\mathcal{L}(p; 0, 0) = \frac{1}{\sqrt{p(1-p)}}[p(1-p)]^2 = [p(1-p)]^{3/2}. \tag{38}$$

This is maximized at $p^\star = \frac{1}{2}$. Thus, the largest single-step reduction in bad mass occurs on "medium-difficulty" prompts where the model is roughly 50–50 between good and bad solutions. At the extremes $p \to 0$ (almost always good) or $p \to 1$ (almost always bad), the factor $p(1-p)$ vanishes and learning slows down sharply: additional GRPO steps make only marginal progress on highly saturated prompts.

**Noisy grading: what changes.** With noise, the learnability speed becomes

$$\mathcal{L}(p; \delta_{\text{FN}}, \delta_{\text{FP}}) = \frac{J}{\sqrt{q(p)(1-q(p))}}[p(1-p)]^2,$$

$$q(p) = (1-\delta_{\text{FN}}) - Jp. \tag{39}$$

This expression highlights two distinct effects:

1. **Global shrinkage via $J$.** Increasing noise decreases $J = 1 - \delta_{\text{FN}} - \delta_{\text{FP}}$, uniformly reducing learnability. In the limit $\delta_{\text{FN}} + \delta_{\text{FP}} \to 1$, the grader becomes uninformative and $\mathcal{L} \to 0$ for all $p$.

2. **Reweighting via $\sigma(p)^{-1}$.** Group $z$-scoring divides by the reward standard deviation $\sigma(p)$, so the effective learning signal is amplified when the reward distribution is highly concentrated (small $\sigma$) and attenuated when it is maximally noisy (large $\sigma$). Equivalently, the normalized separation is $\Delta(p) = J/\sigma(p)$.

*Symmetric noise.* If $\delta_{\text{FN}} = \delta_{\text{FP}} = \delta$, then $J = 1 - 2\delta$ and $q(1-p) = 1 - q(p)$, implying $\sigma(1-p) = \sigma(p)$. Since $p(1-p)$ is also symmetric, $\mathcal{L}(1-p) = \mathcal{L}(p)$, and the maximizer remains at the symmetry point $p^\star = \frac{1}{2}$. Moreover, $q(\frac{1}{2}) = \frac{1}{2}$ and $\sigma(\frac{1}{2}) = \frac{1}{2}$, yielding the explicit peak value

$$\mathcal{L}_{\text{max}}(\delta, \delta) = \mathcal{L}(\tfrac{1}{2}; \delta, \delta) = \frac{J}{1/2}\left(\frac{1}{4}\right)^2 = \frac{1-2\delta}{8}. \tag{40}$$

Hence symmetric noise does *not* shift the most-learnable difficulty, but it reduces the maximal attainable learning speed, collapsing to zero as $\delta \to \frac{1}{2}$.

*Asymmetric noise.* When $\delta_{\mathrm{FN}} \neq \delta_{\mathrm{FP}}$, the symmetry $p \leftrightarrow 1 - p$ is broken and the maximizer shifts to an interior point $p^\dagger \in (0,1)$ that balances the mixture factor $[p(1-p)]^2$ against the normalization term $\sigma(p)$. Differentiating $\log \mathcal{L}$ yields the stationary condition

$$p^\dagger \text{ solves } \quad 2\,\frac{1-2p}{p(1-p)} + \frac{J}{2}\,\frac{1-2q(p)}{q(p)\big(1-q(p)\big)} = 0, \tag{41}$$
$$q(p) = (1 - \delta_{\mathrm{FN}}) - Jp.$$

with boundary clipping if no interior maximizer exists. Intuitively, the term $[p(1-p)]^2$ favors intermediate difficulty, while the factor $1/\sigma(p)$ reweights the signal in a way that can skew the optimum when false negatives and false positives are imbalanced. Equation (41) admits an explicit solution but not a simple elementary one in general: after substituting $q(p) = (1 - \delta_{\mathrm{FN}}) - Jp$ and clearing denominators, the condition reduces to a cubic polynomial in $p$. Thus $p^\dagger$ can be written in closed form via Cardano's formula, although the expression is cumbersome; in practice, we select the real root in $p \in (0,1)$ (or clip to $\{0,1\}$ if the maximizer lies on the boundary). A notable simplification occurs under symmetric noise $\delta_{\mathrm{FN}} = \delta_{\mathrm{FP}}$, where the invariance $\mathcal{L}(1 - p) = \mathcal{L}(p)$ forces $p^\dagger = \frac{1}{2}$. Fig. 8 shows $p\dagger$ for the full range of noises in the learning phase.

*Remark* E.1 (Uninformative or adversarial graders). If $J = 0$ (i.e., $\delta_{\mathrm{FN}} + \delta_{\mathrm{FP}} = 1$), then $\Delta(p) = J/\sigma(p) = 0$ and the mean update provides no systematic signal to reduce bad mass. If $J < 0$, the grader is worse than random and $\Delta(p)$ flips sign; equivalently, swapping labels $r \mapsto 1 - r$ restores an effective $J' > 0$ and recovers the analysis above.

## F. Lyapunov analysis and the role of $J$

We consider a bandit configuration comprising $K$ good arms and $M$ bad arms, as discussed in Appendix D, defined by the probability vector:

$$\mathbf{p} = (p_1, \ldots, p_K,\ p_{b_1}, \ldots, p_{b_M}) \in \Delta^{K+M-1},$$
$$\sum_{j=1}^{K} p_j + \sum_{m=1}^{M} p_{b_m} = 1.$$

We define the total mass of the good and bad blocks respectively as:

$$s(\mathbf{p}) := \sum_{j=1}^{K} p_j \in [0,1],$$
$$P_{\mathrm{bad}}(\mathbf{p}) := \sum_{m=1}^{M} p_{b_m} = 1 - s(\mathbf{p}),$$
$$\mathfrak{J}(\mathbf{p}) = \mathrm{Diag}(\mathbf{p}) - \mathbf{p}\mathbf{p}^\top.$$

Let $\mathbf{1} \in \mathbb{R}^{K+M}$ denote the all-ones vector and let

$$\mathbf{1}_G := (\underbrace{1, \ldots, 1}_{K}, \underbrace{0, \ldots, 0}_{M}) \in \mathbb{R}^{K+M}$$

serve as the indicator vector for the good block.

For $s(\mathbf{p}) \in (0,1)$, it is convenient to introduce within-block normalized coordinates:

$$y_j := \frac{p_j}{s(\mathbf{p})}\ (j \leq K), \qquad z_m := \frac{p_{b_m}}{P_{\mathrm{bad}}(\mathbf{p})}\ (m \leq M).$$

In this representation, $\in \Delta^{K-1}$ and $\in \Delta^{M-1}$, such that

$$\mathbf{p} = (s\,,\ (1 - s)\,).$$

**Block symmetry and GRPO parametrization.** We assume a structure of block symmetry and state dependence defined by:

$$A_j(\mathbf{p}) = a_{\mathrm{g}}\big(s(\mathbf{p})\big), \qquad j \leq K,$$
$$A_{b_m}(\mathbf{p}) = a_{\mathrm{b}}\big(s(\mathbf{p})\big), \qquad m \leq M.$$

The resulting gap between good and bad arms is denoted as:

$$\Delta(s) := a_{\mathrm{g}}(s) - a_{\mathrm{b}}(s).$$

Under the GRPO specialization examined in this work, we set:

$$a_{\mathrm{g}}(s) = \frac{J\,s}{\sigma(s)}, \qquad a_{\mathrm{b}}(s) = -\frac{J\,(1-s)}{\sigma(s)}, \qquad \sigma(s) > 0.$$

This formulation implies that:

$$\Delta(s) = \frac{J}{\sigma(s)}. \tag{42}$$

The advantage vector can then be expressed as:

$$\mathbf{A}(\mathbf{p}) = a_{\mathrm{b}}(s(\mathbf{p}))\,\mathbf{1} + \Delta\big(s(\mathbf{p})\big)\,\mathbf{1}_G.$$

**GRPO mean-field flow.** Our analysis focuses on the GRPO mean-field ordinary differential equation (ODE):

$$\dot{\mathbf{p}} \;=\; \eta \, \mathfrak{J}(\mathbf{p})^2 \, \mathbf{A}(\mathbf{p}), \qquad \eta > 0. \tag{43}$$

**Theorem F.1** (Dichotomy by the sign of $J$; exchange of stability at $J = 0$)**.** *Assume the block-symmetric structure defined above, with $\Delta$ given by* (42)*. Define a scalar potential $F : \Delta^{K+M-1} \to \mathbb{R}$ such that:*

$$F(\mathbf{p}) \;:=\; F\big(s(\mathbf{p})\big), \qquad F'(s) \;=\; \Delta(s). \tag{44}$$

*Given that $s(\mathbf{p}) \in [0,1]$ and $\Delta$ is integrable on $[0,1]$, $F$ remains bounded on $\Delta^{K+M-1}$. For any constant $C$ satisfying $C \geq \sup_{\mathbf{p}} F(\mathbf{p})$, we define the standard decreasing Lyapunov function:*

$$V(\mathbf{p}) \;:=\; C - F(\mathbf{p}) \;\geq\; 0. \tag{45}$$

*Along any trajectory $\mathbf{p}(t)$ of* (43) *originating at $s(0) \in (0,1)$, the following properties hold:*

1. ***Lyapunov identity.*** *For all $t \geq 0$:*

$$\frac{d}{dt} V\big(\mathbf{p}(t)\big) \;=\; -\eta \left\| \mathfrak{J}\big(\mathbf{p}(t)\big) \, \mathbf{A}\big(\mathbf{p}(t)\big) \right\|_2^2 \;\leq\; 0, \tag{46}$$

   *where equality holds if and only if $\mathfrak{J}(\mathbf{p}(t)) \, \mathbf{A}(\mathbf{p}(t)) = 0$. This is equivalent to:*

$$\frac{d}{dt} F\big(\mathbf{p}(t)\big) \;=\; \eta \left\| \mathfrak{J}\big(\mathbf{p}(t)\big) \, \mathbf{A}\big(\mathbf{p}(t)\big) \right\|_2^2 \;\geq\; 0. \tag{47}$$

2. ***Explicit field and the sign of $\dot{s}$.*** *Let $s = s(\mathbf{p})$ and $P_{\mathrm{bad}} := 1 - s$. The components of the field are given by:*

$$\big[ \mathfrak{J}(\mathbf{p}) \, \mathbf{A}(\mathbf{p}) \big]_j \;=\; p_j \, P_{\mathrm{bad}} \, \Delta(s), \qquad j \leq K,$$

$$\big[ \mathfrak{J}(\mathbf{p}) \, \mathbf{A}(\mathbf{p}) \big]_{b_m} \;=\; -p_{b_m} \, s \, \Delta(s), \qquad m \leq M.$$

   *Consequently, for $J \neq 0$:*

$$\dot{s}(t) = \frac{1}{\Delta\big(s(t)\big)} \frac{d}{dt} F\big(\mathbf{p}(t)\big)$$

$$= \eta \, \Delta\big(s(t)\big) \Bigg( P_{\mathrm{bad}}(t)^2 \sum_{j=1}^{K} p_j(t)^2 \tag{48}$$

$$+ \; s(t)^2 \sum_{m=1}^{M} p_{b_m}(t)^2 \Bigg).$$

   *In normalized coordinates, this simplifies to:*

$$\dot{s} \;=\; \eta \, \Delta(s) \, [s(1-s)]^2 \Big( \|\mathbf{y}\|_2^2 + \|\mathbf{z}\|_2^2 \Big). \tag{49}$$

   *Thus, for any $J \neq 0$, the sign of $\dot{s}(t)$ is identical to the sign of $J$.*

3. ***Global behavior and exchange of stability.***

   - *If $J > 0$, $s(t)$ is strictly increasing such that $s(t) \uparrow 1$. In this case, every $\omega$–limit point lies in the good face $\Delta_G := \{ \mathbf{p} : p_{b_1} = \cdots = p_{b_M} = 0 \}$.*

   - *If $J < 0$, $s(t)$ is strictly decreasing such that $s(t) \downarrow 0$. Here, every $\omega$–limit point lies in the bad face $\Delta_B := \{ \mathbf{p} : s(\mathbf{p}) = 0 \}$.*

   - *If $J = 0$, the advantage $A_i(\mathbf{p})$ is constant across all arms, implying $\dot{\mathbf{p}} \equiv 0$. Under these conditions, every point in the simplex is an equilibrium.*

   *As $J$ crosses zero, the global attractor switches from $\Delta_B$ to $\Delta_G$. This represents a parameter-driven exchange of stability, or phase transition, at $J = 0$.*

4. ***Quantitative tail behavior.*** *For $J \neq 0$ and bounded $\sigma$, and following the bounds established in Appendix C, we observe $0 < \sigma_{\min} \leq \sigma(s) \leq \sigma_{\max} \leq \frac{1}{4}$. By Jensen's inequality, we obtain:*

$$|\dot{s}(t)| \;\geq\; \eta \, \frac{|J|}{\sigma_{\max}} \Big( \frac{1}{K} + \frac{1}{M} \Big) s(t)^2 \big(1 - s(t)\big)^2. \tag{50}$$

   *For $J > 0$, there exists a time $T_{1/2}$ such that for all $t \geq T_{1/2}$, $s(t) \geq \frac{1}{2}$ and:*

$$P_{\mathrm{bad}}(t) \leq \Bigg( \frac{1}{P_{\mathrm{bad}}(T_{1/2})}$$

$$+ \; \frac{\eta J}{4\sigma_{\max}} \Big( \frac{1}{K} + \frac{1}{M} \Big) \big(t - T_{1/2}\big) \Bigg)^{-1}. \tag{51}$$

   *Similarly, for $J < 0$, $s(t)$ follows an $\mathcal{O}(1/t)$ decay after a finite transient period.*

*Proof.* Let $F(s) := \int_0^s \Delta(u) \, du$. Since $F(\mathbf{p}) = F(s(\mathbf{p}))$ and $s(\mathbf{p}) = \sum_{j \leq K} p_j$, it follows that $\nabla F(\mathbf{p}) = \Delta(s) \, \mathbf{1}_G$. Utilizing the fact that $\mathfrak{J}(\mathbf{p})\mathbf{1} = \mathbf{0}$, we find:

$$\mathfrak{J}(\mathbf{p}) \, \nabla F(\mathbf{p}) = \Delta(s) \, \mathfrak{J}(\mathbf{p}) \mathbf{1}_G = \mathfrak{J}(\mathbf{p}) \mathbf{A}(\mathbf{p}).$$

Applying the chain rule and the symmetry of $\mathfrak{J}(\mathbf{p})$ yields:

$$\frac{d}{dt} F(\mathbf{p}(t)) = \nabla F(\mathbf{p}(t))^\top \dot{\mathbf{p}}(t) = \eta \left\| \mathfrak{J}(\mathbf{p}(t)) \mathbf{A}(\mathbf{p}(t)) \right\|_2^2,$$

which confirms (47) and, by extension, (46).

The coordinate-wise expressions in (ii) arise from the identity $[\mathfrak{J}(\mathbf{p})\mathbf{A}]_i = p_i(A_i - \bar{A})$, where $\bar{A} = s \, a_{\mathrm{g}}(s) + (1 - s) \, a_{\mathrm{b}}(s)$. The resulting differential inequality in (iv) is obtained by combining (48) with Jensen's lower bounds and the definition $\Delta(s) = J/\sigma(s)$. Integrating over $[T_{1/2}, t]$ while assuming $s(t) \geq \frac{1}{2}$ produces the bound in (51). $\square$

*Remark* F.2 (Interpretation). The dynamics described by (48) and (49) indicate that the direction of mass transfer between blocks is governed exclusively by the sign of $J$. For $J > 0$, the good face $\Delta_G$ is globally attracting, whereas $J < 0$ renders the bad face $\Delta_B$ the attractor. At the critical value $J = 0$, the system undergoes a phase transition characterized by a degenerate continuum of equilibria.

**Decomposition dynamics and Shahshahani structure.** For $s \in (0, 1)$, we decompose the flow into coordinates $(\mathbf{y}, \mathbf{z}, s)$. Equation (43) then reduces to the following system:

$$
\begin{aligned}
\dot{\mathbf{y}} &= \kappa(s)\left(\mathbf{y} \odot \mathbf{y} - \|\mathbf{y}\|_2^2\, \mathbf{y}\right), \\
\dot{\mathbf{z}} &= -\kappa(s)\left(\mathbf{z} \odot \mathbf{z} - \|\mathbf{z}\|_2^2\, \mathbf{z}\right), \\
\dot{s} &= \eta\, \Delta(s)\, [s(1-s)]^2\left(\|\mathbf{y}\|_2^2 + \|\mathbf{z}\|_2^2\right).
\end{aligned}
\tag{52}
$$

where $\kappa(s) := \eta\, \Delta(s)\, s(1-s)$. This reveals that $\Phi(\mathbf{y}) := \frac{1}{2}\|\mathbf{y}\|_2^2$ serves as a state-scaled Shahshahani-gradient potential on $\Delta^{K-1}$, while $\Psi(\mathbf{z}) := \frac{1}{2}\|\mathbf{z}\|_2^2$ acts as the corresponding potential on $\Delta^{M-1}$ but with an inverted sign. For $J > 0$, we observe $\dot{\Phi} \geq 0$ and $\dot{\Psi} \leq 0$. These inequalities reverse when $J < 0$.

**Probabilistic interpretation of the potential $\Phi(\mathbf{y})$.** The function $\Phi(\mathbf{y}) = \frac{1}{2}\sum_{i=1}^{K} y_i^2$ represents the collision probability, also known as the Herfindahl–Hirschman index. It quantifies the concentration within the good block, where $\Phi(\mathbf{y}) = 1/2K$ at the uniform distribution and $\Phi(\mathbf{y}) = 1/2$ at any vertex. An increase in $\Phi$ signifies that mass is condensing onto a smaller subset of arms, while a decrease indicates a more uniform distribution.

The induced intra-good dynamics are expressed as $\dot{\mathbf{y}} = \kappa(s)\, \mathrm{grad}_{\mathrm{Shah}}\, \Phi(\mathbf{y})$. This shows that $\mathbf{y}$ follows a Shahshahani gradient flow of $\Phi$. Consequently, if $J > 0$, the collision probability increases over time, leading to a rich-get-richer effect where the distribution concentrates toward a vertex. Conversely, if $J < 0$, the flow promotes diversity by pushing toward a uniform distribution.

**Coordination-game correspondence.** The replicator field $\dot{y}_i \propto y_i(y_i - \|\mathbf{y}\|_2^2)$ corresponds to the replicator dynamics of a symmetric pure coordination game. In this context, the payoff for each arm is equal to its current population fraction. Our Shahshahani-gradient identity formally states that the replicator dynamics ascend this potential when $\kappa > 0$ and descend it when $\kappa < 0$.

# G. Bad arm dynamics under PPO/GRPO style importance sampling and clipping

We demonstrate that, within the small-step regime, importance sampling (IS) as employed in PPO and GRPO does not alter the leading-order mean-field ODE established in Appendix C. Specifically, IS modifies the conditional mean drift only at order $O(\eta^2)$, ensuring that the $O(\eta)$ ODE limit remains identical to the on-policy (REINFORCE) system.

**Setup.** Consider a softmax policy $\pi_\theta$ over arms $i \in \{b, 1, \ldots, K\}$. During an update, samples are drawn from an "old" policy $\pi_{\mathrm{old}}$ with probability vector $\mathbf{p} = \mathbf{p}_{\mathrm{old}}$. The parameters are then updated to a "new" policy $\pi_{\mathrm{new}}$ with probability vector $\mathbf{p}^+ = \mathbf{p}_{\mathrm{new}}$. We define the exact importance ratio as:

$$
\rho_i := \frac{\pi_{\mathrm{new}}(i)}{\pi_{\mathrm{old}}(i)} = \frac{p_i^+}{p_i}.
$$

This ratio is well-defined because the softmax function has full support ($p_i > 0$). Let $\tilde{r}$ denote the scalar signal used in the update, and let:

$$
A_i(\mathbf{p}) := \mathbb{E}[\tilde{r} \mid I = i, \mathbf{p}].
$$

We assume $\sup_{i, \mathbf{p}} |A_i(\mathbf{p})| < \infty$. In this context, the small-step regime implies $\|\Delta\theta\| = O(\eta)$ and consequently $\|\mathbf{p}^+ - \mathbf{p}\| = O(\eta)$, a result supported by Lemma I.1.

**IS score-function update.** The IS-corrected score-function update is given by:

$$
g_{\mathrm{IS}} := \tilde{r}\, \rho_I\, \nabla_\theta \log \pi_{\mathrm{new}}(I), \qquad I \sim \pi_{\mathrm{old}}, \qquad \Delta\theta = \eta\, g_{\mathrm{IS}}.
$$

For a softmax policy, where $\nabla_\theta \log \pi_{\mathrm{new}}(i) = e_i - \mathbf{p}^+$, the update simplifies to $g_{\mathrm{IS}} = \tilde{r}\, \rho_I\, (e_I - \mathbf{p}^+)$.

**Proposition G.1** (IS affects the mean logit drift only at $O(\eta^2)$). *Define the on-policy vector field:*

$$
G(\mathbf{p}) := \sum_i p_i\, A_i(\mathbf{p})\, (e_i - \mathbf{p}).
$$

*In the small-step regime, the following holds:*

$$
\mathbb{E}[\Delta\theta \mid \mathbf{p}] = \eta\, G(\mathbf{p}) + O(\eta^2).
$$

*Proof.* Taking the conditional expectation and applying $\mathbb{E}[\tilde{r} \mid I = i, \mathbf{p}] = A_i(\mathbf{p})$, we have:

$$
\mathbb{E}[g_{\mathrm{IS}} \mid \mathbf{p}] = \sum_i p_i\, \rho_i\, A_i(\mathbf{p})\, (e_i - \mathbf{p}^+).
$$

By utilizing the identity $p_i \rho_i = p_i^+$, the expression becomes:

$$
\mathbb{E}[g_{\mathrm{IS}} \mid \mathbf{p}] = \sum_i p_i^+\, A_i(\mathbf{p})\, (e_i - \mathbf{p}^+) =: \widetilde{G}(\mathbf{p}^+; \mathbf{p}),
$$

$$
\widetilde{G}(\mathbf{q}; \mathbf{p}) := \sum_i q_i\, A_i(\mathbf{p})\, (e_i - \mathbf{q}).
$$

For a fixed $\mathbf{p}$, the map $\mathbf{q} \mapsto \widetilde{G}(\mathbf{q}; \mathbf{p})$ is a smooth polynomial in $\mathbf{q}$ with bounded coefficients. It is therefore locally

Lipschitz in $\mathbf{q}$. Given that $\|\mathbf{p}^+ - \mathbf{p}\| = O(\eta)$, it follows that:

$$\widetilde{G}(\mathbf{p}^+; \mathbf{p}) = \widetilde{G}(\mathbf{p}; \mathbf{p}) + O(\|\mathbf{p}^+ - \mathbf{p}\|) = G(\mathbf{p}) + O(\eta).$$

Multiplying by $\eta$ yields the desired result: $\mathbb{E}[\Delta\theta \mid \mathbf{p}] = \eta\, G(\mathbf{p}) + O(\eta^2)$. $\qquad\square$

**Expanded form and relation to Appendix C.** This conclusion can also be derived by examining the identity $\rho_i = 1 + \Delta p_i/p_i$, where $\Delta p_i := p_i^+ - p_i$. This implies that $\mathbb{E}[\rho_i \mid \mathbf{p}] = 1 + \mathbb{E}[\Delta p_i \mid \mathbf{p}]/p_i$. Since replacing $\mathbf{p}^+$ with $\mathbf{p}$ introduces only an $O(\eta)$ error, its contribution after multiplying by $\eta$ is $O(\eta^2)$. Consequently:

$$\mathbb{E}[\Delta\theta \mid \mathbf{p}] = \eta \sum_i \Big( p_i + \mathbb{E}[\Delta p_i \mid \mathbf{p}] \Big) A_i(\mathbf{p})\,(e_i - \mathbf{p}) + O(\eta^2). \tag{53}$$

By letting $\bar{A} := \sum_i p_i A_i(\mathbf{p})$ and $\bar{A}' := \sum_i \mathbb{E}[\Delta p_i \mid \mathbf{p}] A_i(\mathbf{p})$, the coordinate-wise updates are expressed as:

$$\mathbb{E}[\Delta\theta_i \mid \mathbf{p}] = \eta\, p_i \big( A_i(\mathbf{p}) - \bar{A} \big) \tag{54}$$

$$+ \eta \Big( \mathbb{E}[\Delta p_i \mid \mathbf{p}]\, A_i(\mathbf{p}) - p_i\, \bar{A}' \Big) + O(\eta^2). \tag{55}$$

This structure highlights the "extra terms" that appear when the IS mean update is expanded. To show these terms are second order, we apply the softmax pushforward from Lemma I.1, yielding $\mathbb{E}[\Delta\mathbf{p} \mid \mathbf{p}] = \mathfrak{J}(\mathbf{p})\,\mathbb{E}[\Delta\theta \mid \mathbf{p}] + O(\eta^2)$. Because $\mathbb{E}[\Delta\theta \mid \mathbf{p}] = O(\eta)$, the term $\mathbb{E}[\Delta p_i \mid \mathbf{p}]$ is also $O(\eta)$, confirming that the correction in (54) is $O(\eta^2)$.

**Block-symmetric specialization.** Under the assumption of block symmetry, we have $A_{b_m}(\mathbf{p}) = a_{\mathrm{b}}(p)$ and $A_j(\mathbf{p}) = a_{\mathrm{g}}(p)$ for $m = 1, \ldots, M$ and $j = 1, \ldots, K$. The mean advantage is then $\bar{A} = p\, a_{\mathrm{b}}(p) + (1-p) a_{\mathrm{g}}(p)$. The leading terms in (54) result in:

$$\mathbb{E}[\Delta\theta_{b_m} \mid \mathbf{p}] = \eta\, p(1-p) \big( a_{\mathrm{b}}(p) - a_{\mathrm{g}}(p) \big) z_m + O(\eta^2),$$

$$\mathbb{E}[\Delta\theta_j \mid \mathbf{p}] = -\eta\, p(1-p) \big( a_{\mathrm{b}}(p) - a_{\mathrm{g}}(p) \big) y_j + O(\eta^2).$$

These expressions match the REINFORCE dynamics to first order in $\eta$.

**Corollary G.2** (Invariance of the leading-order mean-field ODE). *In the small-step regime, the conditional expectation of the probability update satisfies:*

$$\mathbb{E}[\Delta\mathbf{p} \mid \mathbf{p}] = \eta\, \mathfrak{J}(\mathbf{p})\, G(\mathbf{p}) + O(\eta^2).$$

*Consequently, the leading-order mean-field ODE for good and bad arm dynamics remains as derived in Appendix C.*

**Clipping in PPO and GRPO.** When clipping is applied to the ratios such that $\widehat{\rho}_i = \mathrm{clip}(\rho_i, 1 - \varepsilon, 1 + \varepsilon')$, the ratio $\rho_i$ approaches 1 as $\eta \to 0$. For sufficiently small $\eta$, the clipping mechanism remains inactive. Therefore, clipping does not influence the leading-order ODE.

**Conclusion.** In the small-step regime, the inclusion of importance sampling and clipping with fixed thresholds does not change the leading-order mean-field ODE. The impact of these techniques is confined to the $O(\eta^2)$ terms.

# H. KL Regularization

In this section, we analyze how KL regularization modifies the policy dynamics in our multi-armed bandit abstraction. Many practical algorithms (e.g., PPO/GRPO-style methods) add a KL penalty to keep the learned policy close to a reference policy, and we study the resulting mean-field drift in our multi-good/multi-bad model.

We begin by recalling the notation for the *multi-good/bad-arm* model introduced in Appendix D. Given $K$ good arms and $M$ bad arms, we represent the policy as

$$\mathbf{p} = (p_1, \ldots, p_K,\ p_{b_1}, \ldots, p_{b_M}) \in \Delta^{K+M-1}.$$

The *total bad mass* and the within-block compositions are

$$p = \sum_{m=1}^{M} p_{b_m} \in [0, 1],$$

$$y_j = \frac{p_j}{1 - p}, \qquad j \le K,$$

$$z_m = \frac{p_{b_m}}{p}, \qquad m \le M.$$

Equivalently,

$$p_j = (1 - p)\, y_j \quad (j \le K), \qquad p_{b_m} = p\, z_m \quad (m \le M),$$

with $\mathbf{y} \in \Delta^{K-1}$ and $\mathbf{z} \in \Delta^{M-1}$.

**Reference policy.** We define the KL reference policy $\mathbf{p}^{\mathrm{ref}}$ analogously:

$$\mathbf{p}^{\mathrm{ref}} = \Big( (1 - p_{\mathrm{ref}})\, y_1^{\mathrm{ref}}, \ldots, (1 - p_{\mathrm{ref}})\, y_K^{\mathrm{ref}},\ p_{\mathrm{ref}}\, z_1^{\mathrm{ref}}, \ldots, p_{\mathrm{ref}}\, z_M^{\mathrm{ref}} \Big),$$

where $p_{\mathrm{ref}} \in (0, 1)$, $\mathbf{y}^{\mathrm{ref}} \in \Delta^{K-1}$, and $\mathbf{z}^{\mathrm{ref}} \in \Delta^{M-1}$. We also use the bad-mass log-odds

$$\ell(p) := \log \frac{p}{1 - p}, \qquad \ell_{\mathrm{ref}} := \log \frac{p_{\mathrm{ref}}}{1 - p_{\mathrm{ref}}}.$$

**Replicator/natural-gradient form (recall).** We view a penalty $\Phi(\mathbf{p})$ as inducing a Shahshahani (natural-gradient) flow on the simplex. Concretely, under a small step size $\eta$,

adding an objective term $-\beta\,\Phi(\mathbf{p})$ produces the replicator drift

$$\dot{p}_i \;=\; -\eta\beta\,p_i\Big(\partial_{p_i}\Phi(\mathbf{p}) \;-\; \sum_k p_k\,\partial_{p_k}\Phi(\mathbf{p})\Big). \quad (56)$$

Equivalently, if we interpret $\Delta\theta_i$ as a small logit increment, then to first order

$$\Delta p_i \;=\; p_i\Big(\Delta\theta_i - \sum_k p_k\Delta\theta_k\Big) \;+\; O(\|\Delta\theta\|^2). \quad (57)$$

Thus, choosing *centered* increments with $\sum_k p_k\Delta\theta_k = 0$ yields $\Delta p_i = p_i\Delta\theta_i$ at first order, which makes it easy to realize a desired replicator drift.

**Lemma H.1** (Exact reverse-KL decomposition (multi-bad setting)). *The reverse KL divergence decomposes into a two-class term (bad vs. good) plus within-block terms:*

$$D_{\mathrm{KL}}(\mathbf{p}\|\mathbf{p}^{\mathrm{ref}}) = \; p\log\frac{p}{p_{\mathrm{ref}}} + (1-p)\log\frac{1-p}{1-p_{\mathrm{ref}}}$$
$$+ (1-p)\,D_{\mathrm{KL}}(\mathbf{y}\|\mathbf{y}^{\mathrm{ref}}) + p\,D_{\mathrm{KL}}(\mathbf{z}\|\mathbf{z}^{\mathrm{ref}}).$$

*Proof.* Substituting $p_j = (1-p)y_j$ and $p_{b_m} = p z_m$, we get

$$\log\frac{p_j}{p_j^{\mathrm{ref}}} = \log\frac{1-p}{1-p_{\mathrm{ref}}} + \log\frac{y_j}{y_j^{\mathrm{ref}}},$$
$$\log\frac{p_{b_m}}{p_{b_m}^{\mathrm{ref}}} = \log\frac{p}{p_{\mathrm{ref}}} + \log\frac{z_m}{z_m^{\mathrm{ref}}}.$$

Summing $\sum_{j\le K} p_j(\cdot)$ and $\sum_{m\le M} p_{b_m}(\cdot)$ yields the stated decomposition. $\square$

**Two KL choices.** In practice, one may penalize either (A) only the two-class (inter-block) divergence or (B) the full reverse-KL divergence.

**Proposition H.2** (Two-class KL penalty (bad vs. good only)). *Let*

$$\Phi_{2c}(p) := p\log\frac{p}{p_{\mathrm{ref}}} + (1-p)\log\frac{1-p}{1-p_{\mathrm{ref}}}.$$

*The Shahshahani/natural-gradient (replicator) flow for $-\beta\,\Phi_{2c}$ satisfies*

$$\dot{p}\Big|_{\mathrm{KL},2c} = -\eta\beta\,p(1-p)\big(\ell - \ell_{\mathrm{ref}}\big),$$
$$\dot{\ell}\Big|_{\mathrm{KL},2c} = -\eta\beta\big(\ell - \ell_{\mathrm{ref}}\big). \quad (58)$$

*A centered small-step logit update realizing this flow (i.e., $\sum_i p_i\,\Delta\theta_i^{\mathrm{KL}} = 0$) is*

$$\Delta\theta_{b_m}^{\mathrm{KL}} = -\eta\,\beta(1-p)\,(\ell - \ell_{\mathrm{ref}}), \qquad m \le M,$$
$$\Delta\theta_j^{\mathrm{KL}} = +\eta\,\beta p\,(\ell - \ell_{\mathrm{ref}}), \qquad j \le K.$$

*Because $\Delta\theta_i^{\mathrm{KL}}$ is constant within each block, this KL term induces no deterministic drift in $\mathbf{y}$ or $\mathbf{z}$; it acts only on the total bad mass $p$.*

*Proof.* View $\Phi_{2c}$ as a function on the full simplex via $p = \sum_{m=1}^{M} p_{b_m}$. Then

$$\partial_{p_{b_m}}\Phi_{2c}(\mathbf{p}) = \Phi'_{2c}(p), \qquad \partial_{p_j}\Phi_{2c}(\mathbf{p}) = 0,$$

and

$$\sum_i p_i\,\partial_{p_i}\Phi_{2c}(\mathbf{p}) = \sum_{m=1}^{M} p_{b_m}\,\Phi'_{2c}(p) = p\,\Phi'_{2c}(p).$$

Plugging into (56) yields

$$\dot{p}_{b_m} = -\eta\beta\,p_{b_m}\Big(\Phi'_{2c}(p) - p\,\Phi'_{2c}(p)\Big) = -\eta\beta\,p_{b_m}(1-p)\,\Phi'_{2c}(p),$$

$$\dot{p}_j = -\eta\beta\,p_j\Big(0 - p\,\Phi'_{2c}(p)\Big) = +\eta\beta\,p_j p\,\Phi'_{2c}(p).$$

Summing over bad arms yields

$$\dot{p} = \sum_{m=1}^{M}\dot{p}_{b_m} = -\eta\beta\,p(1-p)\,\Phi'_{2c}(p).$$

A direct derivative computation gives

$$\Phi'_{2c}(p) = \log\frac{p}{p_{\mathrm{ref}}} - \log\frac{1-p}{1-p_{\mathrm{ref}}} = \ell - \ell_{\mathrm{ref}},$$

which implies $\dot{p}|_{\mathrm{KL},2c} = -\eta\beta\,p(1-p)(\ell - \ell_{\mathrm{ref}})$. Since $\dot{\ell} = \dot{p}/[p(1-p)]$, we obtain $\dot{\ell}|_{\mathrm{KL},2c} = -\eta\beta(\ell - \ell_{\mathrm{ref}})$.

Finally, there is no within-block drift. For any $m, r \le M$,

$$\frac{d}{dt}\log\frac{p_{b_m}}{p_{b_r}} = \frac{\dot{p}_{b_m}}{p_{b_m}} - \frac{\dot{p}_{b_r}}{p_{b_r}}$$
$$= -\eta\beta(1-p)\Phi'_{2c}(p) + \eta\beta(1-p)\Phi'_{2c}(p) = 0. \quad (59)$$

so all ratios $p_{b_m}/p_{b_r}$ are constant and hence $\mathbf{z}$ is constant; the same argument applies to $\mathbf{y}$.

For the logit realization, take the *centered* increment

$$\Delta\theta_i^{\mathrm{KL}} = -\eta\beta\Big(\partial_{p_i}\Phi_{2c} - \sum_k p_k\partial_{p_k}\Phi_{2c}\Big),$$

which gives exactly the stated block-constant increments (using $\Phi'_{2c}(p) = \ell - \ell_{\mathrm{ref}}$). By centering, $\sum_i p_i\Delta\theta_i^{\mathrm{KL}} = 0$, and then (57) implies

$$\Delta p = \sum_{m=1}^{M}\Delta p_{b_m} = \sum_{m=1}^{M} p_{b_m}\Delta\theta_{b_m}^{\mathrm{KL}} = -\eta\beta\,p(1-p)\,(\ell - \ell_{\mathrm{ref}}),$$

matching the continuous-time drift. $\square$

*Remark* H.3. The reader may notice that the previous proof started from the replicator dynamics (56). Another, less geometric, approach to model the effect of the KL term is to mimic the derivation of original ODEs in Appendix D.

The contribution of the KL term $-\beta\,\Phi_{2c}(p)$ added to the objective function to the gradient with respect to $\theta$ is

$$-\beta\,\Phi'_{2c}(p)\nabla_\theta p.$$

Notice that $p$ is the sum of the bad-arm masses $p_{b_m}$ ($1 \le m \le M$); and in general, the partial derivatives of the softmax are given by $\nabla_\theta p_I = p_I(e_I - \mathbf{p})$. Thus

$$\nabla_\theta p = \begin{bmatrix} \mathbf{0} \\ \mathbf{P}_{\text{bad}} \end{bmatrix} - p \begin{bmatrix} \mathbf{P}_{\text{good}} \\ \mathbf{P}_{\text{bad}} \end{bmatrix} = \begin{bmatrix} -p\,\mathbf{P}_{\text{good}} \\ (1-p)\,\mathbf{P}_{\text{bad}} \end{bmatrix}.$$

Multiplying by the learning rate, we conclude that $\Delta\theta$ changes by

$$-\eta\,\beta\,\Phi'_{2c}(p) \begin{bmatrix} -p\,\mathbf{P}_{\text{good}} \\ (1-p)\,\mathbf{P}_{\text{bad}} \end{bmatrix}$$

where we are in the $(K, M)$-block form coming from (good, bad) arms. Now, to obtain the contribution of the KL term to arm probabilities, denoted by $\Delta\mathbf{p}^{\text{KL}}$, we need to multiply the last vector by $\mathfrak{J}(\mathbf{p}) = \text{Diag}(\mathbf{p}) - \mathbf{p}\mathbf{p}^\top$ which yields:

$$-\eta\,\beta\,\Phi'_{2c}(p) \begin{bmatrix} -p\,\mathbf{P}_{\text{good}} \odot \mathbf{P}_{\text{good}} \\ (1-p)\,\mathbf{P}_{\text{bad}} \odot \mathbf{P}_{\text{bad}} \end{bmatrix}$$
$$+\,\eta\,\beta\,\Phi'_{2c}(p)\left(-p\,\|\,\mathbf{P}_{\text{good}}\,\|^2 + (1-p)\,\|\,\mathbf{P}_{\text{bad}}\,\|^2\right)\mathbf{p}.$$

The sum of last $M$ components, corresponding to the bad arms, is

$$-\eta\,\beta\,\Phi'_{2c}(p) \cdot (1-p)\,\|\,\mathbf{P}_{\text{bad}}\,\|^2$$
$$+\,\eta\,\beta\,\Phi'_{2c}(p)\left(-p\,\|\,\mathbf{P}_{\text{good}}\,\|^2 + (1-p)\,\|\,\mathbf{P}_{\text{bad}}\,\|^2\right)\cdot p,$$

which simplifies as

$$-\eta\,\beta\,\Phi'_{2c}(p) \cdot \left((1-p)^2\,\|\,\mathbf{P}_{\text{bad}}\,\|^2 + p^2\,\|\,\mathbf{P}_{\text{good}}\,\|^2\right)$$
$$=-\eta\,\beta\,\Phi'_{2c}(p) \cdot p^2(1-p)^2\left(\|\mathbf{y}\|_2^2 + \|\mathbf{z}\|_2^2\right).$$

Consequently, in this approach, the ODE for the total bad arm mass changes from (9) to

$$\dot{p} = -\eta\,\frac{J}{\sigma(p)}\,[p(1-p)]^2\,C(y,z)$$
$$-\eta\,\beta\,[p(1-p)]^2\left(\text{logit}(p) - \text{logit}(p_{\text{ref}})\right)C(y,z). \tag{60}$$

**Proposition H.4** (Full reverse-KL penalty (multi-bad setting))**.** *Let* $\Phi_{\text{full}}(\mathbf{p}) := D_{\text{KL}}(\mathbf{p}\|\mathbf{p}^{\text{ref}})$. *The replicator flow for* $-\beta\,\Phi_{\text{full}}$ *induces the bad-mass drift*

$$\dot{p}\Big|_{\text{KL,full}} = -\eta\beta\,p(1-p)\Big(\ell - \ell_{\text{ref}}$$
$$-\,D_{\text{KL}}(\mathbf{y}\|\mathbf{y}^{\text{ref}}) + D_{\text{KL}}(\mathbf{z}\|\mathbf{z}^{\text{ref}})\Big),$$

*and hence*

$$\dot{\ell}\Big|_{\text{KL,full}} = -\eta\beta\Big(\ell - \ell_{\text{ref}} - D_{\text{KL}}(\mathbf{y}\|\mathbf{y}^{\text{ref}}) + D_{\text{KL}}(\mathbf{z}\|\mathbf{z}^{\text{ref}})\Big).$$

*A canonical centered logit increment follows the natural-gradient form*

$$\Delta\theta_i^{\text{KL}} = -\eta\beta\Big(\log\frac{p_i}{p_i^{\text{ref}}} - \sum_k p_k \log\frac{p_k}{p_k^{\text{ref}}}\Big),$$

*which satisfies* $\sum_i p_i \Delta\theta_i^{\text{KL}} = 0$ *and yields the stated* $p$-*drift.*

*Moreover, in* $(\mathbf{y}, \mathbf{z})$-*coordinates, the full reverse-KL induces independent within-block replicator pulls:*

$$\dot{y}_j\Big|_{\text{KL,full}} = -\eta\beta\,y_j\left(\log\frac{y_j}{y_j^{\text{ref}}} - \sum_{i=1}^K y_i \log\frac{y_i}{y_i^{\text{ref}}}\right),$$

$$\dot{z}_m\Big|_{\text{KL,full}} = -\eta\beta\,z_m\left(\log\frac{z_m}{z_m^{\text{ref}}} - \sum_{r=1}^M z_r \log\frac{z_r}{z_r^{\text{ref}}}\right).$$

*Proof outline (with the main algebra).* For $\Phi_{\text{full}}(\mathbf{p}) = \sum_i p_i \log\frac{p_i}{p_i^{\text{ref}}}$ we have $\partial_{p_i}\Phi_{\text{full}} = \log\frac{p_i}{p_i^{\text{ref}}} + 1$; the constant $+1$ cancels under the mean-subtraction in (56), yielding

$$\dot{p}_i = -\eta\beta\,p_i\left(\log\frac{p_i}{p_i^{\text{ref}}} - \sum_k p_k \log\frac{p_k}{p_k^{\text{ref}}}\right).$$

To get $\dot{p}$, sum over bad indices:

$$\dot{p} = \sum_{m=1}^M \dot{p}_{b_m} = -\eta\beta\Big(\sum_{m=1}^M p_{b_m}\log\frac{p_{b_m}}{p_{b_m}^{\text{ref}}} - p\sum_k p_k\log\frac{p_k}{p_k^{\text{ref}}}\Big).$$

Using the block parametrization,

$$\log\frac{p_{b_m}}{p_{b_m}^{\text{ref}}} = \log\frac{p}{p_{\text{ref}}} + \log\frac{z_m}{z_m^{\text{ref}}},$$
$$\log\frac{p_j}{p_j^{\text{ref}}} = \log\frac{1-p}{1-p_{\text{ref}}} + \log\frac{y_j}{y_j^{\text{ref}}}. \tag{61}$$

we obtain

$$\sum_{m=1}^M p_{b_m}\log\frac{p_{b_m}}{p_{b_m}^{\text{ref}}} = p\log\frac{p}{p_{\text{ref}}} + p\,D_{\text{KL}}(\mathbf{z}\|\mathbf{z}^{\text{ref}}),$$

and by Lemma H.1,

$$\sum_k p_k\log\frac{p_k}{p_k^{\text{ref}}} = \Phi_{2c}(p) + (1-p)\,D_{\text{KL}}(\mathbf{y}\|\mathbf{y}^{\text{ref}}) + p\,D_{\text{KL}}(\mathbf{z}\|\mathbf{z}^{\text{ref}}).$$

Substituting and simplifying yields

$$\dot{p} = -\eta\beta\,p(1-p)\Big(\log\frac{p}{p_{\text{ref}}} - \log\frac{1-p}{1-p_{\text{ref}}}$$
$$-\,D_{\text{KL}}(\mathbf{y}\|\mathbf{y}^{\text{ref}}) + D_{\text{KL}}(\mathbf{z}\|\mathbf{z}^{\text{ref}})\Big). \tag{62}$$

and the log-odds form follows since $\log\frac{p}{p_{\text{ref}}} - \log\frac{1-p}{1-p_{\text{ref}}} = \ell - \ell_{\text{ref}}$. Finally, the $(\mathbf{y}, \mathbf{z})$ equations follow by differentiating $y_j = p_j/(1-p)$ and $z_m = p_{b_m}/p$ and substituting the

above $\dot{p}_i$ expressions; the same cancellations as in Proposition H.2 reduce each block to its own replicator pull toward the corresponding within-block reference. □

*Remark* H.5 (Comparison of KL penalties). The two-class penalty $\Phi_{2c}(p)$ depends *only* on the aggregate bad mass $p$ (equivalently, on $\ell = \log \frac{p}{1-p}$). Consequently, its Shahshahani field is *block-constant*: all good arms receive the same logit increment and all bad arms receive the same logit increment. This implies that the within-block compositions are invariant,

$$\dot{\mathbf{y}}\Big|_{\mathrm{KL},2c} = 0, \qquad \dot{\mathbf{z}}\Big|_{\mathrm{KL},2c} = 0,$$

and the KL regularizer acts solely as a logistic contraction of $\ell$ toward $\ell_{\mathrm{ref}}$:

$$\dot{\ell}\Big|_{\mathrm{KL},2c} = -\eta\beta(\ell - \ell_{\mathrm{ref}}).$$

In contrast, the full reverse-KL splits as

$$D_{\mathrm{KL}}(\mathbf{p}\|\mathbf{p}^{\mathrm{ref}}) = \Phi_{2c}(p) + (1-p)D_{\mathrm{KL}}(\mathbf{y}\|\mathbf{y}^{\mathrm{ref}}) \\ + pD_{\mathrm{KL}}(\mathbf{z}\|\mathbf{z}^{\mathrm{ref}}). \tag{63}$$

so it produces *two* effects simultaneously: (i) independent within-block replicator pulls that damp deviations of $\mathbf{y}$ and $\mathbf{z}$ from $\mathbf{y}^{\mathrm{ref}}$ and $\mathbf{z}^{\mathrm{ref}}$, and (ii) an additional scalar feedback into the bad-mass drift. Concretely, the effective restoring force on the log-odds becomes

$$\dot{\ell}\Big|_{\mathrm{KL,full}} = -\eta\beta\Big(\ell - \ell_{\mathrm{ref}} - D_{\mathrm{KL}}(\mathbf{y}\|\mathbf{y}^{\mathrm{ref}}) + D_{\mathrm{KL}}(\mathbf{z}\|\mathbf{z}^{\mathrm{ref}})\Big).$$

Thus, if the bad block is more "misaligned" than the good block ($D_{\mathrm{KL}}(\mathbf{z}\|\mathbf{z}^{\mathrm{ref}}) > D_{\mathrm{KL}}(\mathbf{y}\|\mathbf{y}^{\mathrm{ref}})$), the KL term strengthens the push to *decrease* $p$; whereas a larger within-good mismatch can partially counteract that push because it enters with a negative sign. In particular, when $\mathbf{y} = \mathbf{y}^{\mathrm{ref}}$ and $\mathbf{z} = \mathbf{z}^{\mathrm{ref}}$, the full reverse-KL reduces to the two-class behavior.

### H.1. Full Mean-Field ODE for the Bad Mass with KL

We now integrate the reward-driven mean-field drift with the KL penalties derived above. For this analysis, we omit the clipping or importance-ratio factors typically found in PPO/GRPO: our previous results indicate these contribute only second-order corrections and do not alter the fundamental phase portrait of the mean-field drift.

Define the following quantities:

$$\alpha(p) := \eta \frac{J}{\sigma(p)} p(1-p),$$

$$s_2 := \|\mathbf{y}\|_2^2 \in \left[\frac{1}{K}, 1\right],$$

$$t_2 := \|\mathbf{z}\|_2^2 \in \left[\frac{1}{M}, 1\right].$$

Based on the reward term alone, the baseline drift of the *total bad mass* is

$$\mathbb{E}[\Delta p]_{\mathrm{reward}} = -\alpha(p)\, p(1-p)\, (s_2 + t_2) \\ = -\eta \frac{J}{\sigma(p)} [p(1-p)]^2 (s_2 + t_2). \tag{64}$$

**Combined dynamics.** Combining the reward drift with each KL choice yields the governed ODEs

$$\dot{p} = -\eta \frac{J}{\sigma(p)} [p(1-p)]^2 (s_2 + t_2) - \eta\beta\, p(1-p)(\ell - \ell_{\mathrm{ref}})$$
(two-class KL)

$$\dot{p} = -\eta \frac{J}{\sigma(p)} [p(1-p)]^2 (s_2 + t_2) - \eta\beta\, p(1-p)\Big(\ell - \ell_{\mathrm{ref}} \tag{65}$$
$$- D_{\mathrm{KL}}(\mathbf{y}\|\mathbf{y}^{\mathrm{ref}}) + D_{\mathrm{KL}}(\mathbf{z}\|\mathbf{z}^{\mathrm{ref}})\Big)$$
(full reverse-KL)

### H.2. Nullcline, Interior Equilibrium, and the Prevention of Collapse

Using $s_2 = \|\mathbf{y}\|_2^2$, $t_2 = \|\mathbf{z}\|_2^2$, and $\ell(p) = \log\frac{p}{1-p}$, the two-class KL ODE can be written as

$$\dot{p} = \eta\, p(1-p)\left\{-\frac{J}{\sigma(p)} p(1-p)\, (s_2 + t_2) - \beta(\ell - \ell_{\mathrm{ref}})\right\}.$$

This formulation allows us to characterize the stationary behavior of the system.

**Definition H.6** (Nullcline). The interior nullcline $\{\dot{p} = 0\}$ on the interval $(0, 1)$ is defined by the graph

$$\beta(\ell(p) - \ell_{\mathrm{ref}}) = -\frac{J}{\sigma(p)} p(1-p)\, (s_2 + t_2). \tag{66}$$

**Theorem H.7** (Existence, uniqueness, and stability of the interior equilibrium). *For any penalty strength $\beta > 0$ and fixed within-block distributions $(\mathbf{y}, \mathbf{z})$:*

1. *There exists a unique equilibrium $p^\star \in (0, 1)$ satisfying (66).*

2. *This equilibrium $p^\star$ is globally asymptotically stable on $(0, 1)$.*

*Proof sketch.* On the open interval $(0, 1)$, the right-hand side of (66) is continuous and bounded (under mild regularity of $\sigma$). In contrast, the log-odds $\ell(p)$ is strictly increasing, with $\ell(p) \to -\infty$ as $p \downarrow 0$ and $\ell(p) \to +\infty$ as $p \uparrow 1$. Existence and uniqueness follow from the intermediate value theorem and monotonicity. Asymptotic stability follows by linearization: the bracketed term in $\dot{p}$ is strictly decreasing in $p$ near the intersection, implying $\partial_p \dot{p}(p^\star) < 0$. □

**Corollary H.8** (KL regularization prevents collapse for $J < 0$). *When $J < 0$, the right-hand side of* (66) *is strictly positive, implying $\ell(p^\star) > \ell_{\mathrm{ref}}$ and thus $p^\star > p_{\mathrm{ref}}$. While the unregularized dynamics ($\beta = 0$) would drive $p(t) \uparrow 1$ (collapse onto the bad-arm block), any $\beta > 0$ ensures the existence of a stable $p^\star < 1$. Consequently, the long-run accuracy $1 - p^\star$ remains strictly positive and is necessarily higher than in the $\beta = 0$ regime.*

**Asymptotic behavior of $p^\star$.** Let $\sigma_\star := \sigma(p^\star)$. We examine the equilibrium under varying KL strengths:

- **Strong KL** ($\beta \to \infty$). Expanding (66) around $p_{\mathrm{ref}}$ yields

$$p^\star = p_{\mathrm{ref}} - \frac{J}{\beta} \frac{\left[ p_{\mathrm{ref}}(1 - p_{\mathrm{ref}}) \right]^2}{\sigma(p_{\mathrm{ref}})} (s_2 + t_2) + O\!\left( \tfrac{1}{\beta^2} \right). \tag{67}$$

  For $J < 0$, the correction term is positive and scales as $O(\beta^{-1})$, indicating that $p^\star$ approaches $p_{\mathrm{ref}}$ from above as $\beta$ increases.

- **Weak KL** ($\beta \downarrow 0$, $J < 0$). Setting $\varepsilon := 1 - p$ and balancing leading-order terms near $p = 1$ gives

$$1 - p^\star \sim \frac{\beta}{c} \log \frac{c}{\beta}, \qquad c := -\frac{J}{\sigma(1)} (s_2 + t_2) > 0. \tag{68}$$

  This confirms that any non-zero $\beta$ is sufficient to prevent total collapse ($p^\star < 1$).

# I. Properties of the Simplex

This section collects the geometric and algebraic facts about the probability simplex that we repeatedly use in the main text and appendix. Let $d \geq 2$ be an integer and let $\mathbf{1} \in \mathbb{R}^d$ denote the all-ones vector. We consider the probability simplex

$$\Delta^{d-1} := \left\{ \mathbf{p} \in \mathbb{R}^d_{\geq 0} : \mathbf{1}^\top \mathbf{p} = 1 \right\},$$

whose affine tangent space at any interior point $\mathbf{p}$ is

$$T_{\mathbf{p}} \Delta^{d-1} := \left\{ \mathbf{v} \in \mathbb{R}^d : \mathbf{1}^\top \mathbf{v} = 0 \right\}.$$

All definitions and statements below apply verbatim to any simplex-valued variable (e.g. $\mathbf{p}$ or $\mathbf{y}$) by renaming.

A recurring character in simplex geometry is the matrix

$$\mathfrak{J}(\mathbf{p}) := \mathrm{Diag}(\mathbf{p}) - \mathbf{p}\,\mathbf{p}^\top,$$

which simultaneously plays three roles: (i) it is the Jacobian of the softmax map, (ii) it projects directions back to the simplex tangent space, and (iii) it is the inverse of the Shahshahani metric when restricted to the tangent. We build these facts in steps.

**Lemma I.1** (Softmax differential is tangent; Jacobian form). *Let $\mathbf{p} = \mathrm{softmax}(\boldsymbol{\theta}) \in \Delta^{d-1}$ with components $p_i(\boldsymbol{\theta}) = e^{\theta_i}/Z(\boldsymbol{\theta})$, where $Z(\boldsymbol{\theta}) = \sum_{j=1}^{d} e^{\theta_j}$. Then*

$$\frac{\partial p_i}{\partial \theta_j} = p_i(\delta_{ij} - p_j) \quad \Longrightarrow \quad \mathrm{d}\mathbf{p} = \mathfrak{J}(\mathbf{p})\,\mathrm{d}\boldsymbol{\theta}.$$

*Moreover, $\mathfrak{J}(\mathbf{p})\mathbf{1} = 0$ and $\mathbf{1}^\top \mathfrak{J}(\mathbf{p}) = 0^\top$, so $\mathrm{Im}\,\mathfrak{J}(\mathbf{p}) \subseteq T_{\mathbf{p}}\Delta^{d-1}$; i.e., the softmax differential always lies in the tangent space.*

Lemma I.1 already explains why replicator-like dynamics naturally appear when working in logits: any infinitesimal change in $\boldsymbol{\theta}$ is automatically mapped to a tangent direction in probability space via $\mathfrak{J}(\mathbf{p})$.

**Corollary I.2** (Rank, nullspace, and image). *When $\mathbf{p}$ lies in the interior ($p_i > 0$ for all $i$), the matrix $\mathfrak{J}(\mathbf{p})$ is symmetric positive semidefinite and satisfies*

$$\mathrm{null}\big(\mathfrak{J}(\mathbf{p})\big) = \mathrm{span}\{\mathbf{1}\},$$
$$\mathrm{rank}\big(\mathfrak{J}(\mathbf{p})\big) = d - 1,$$
$$\mathrm{Im}\,\mathfrak{J}(\mathbf{p}) = T_{\mathbf{p}}\Delta^{d-1}.$$

*In particular, $\mathfrak{J}(\mathbf{p})$ acts as a linear automorphism of the tangent space.*

The only "forbidden" direction is $\mathbf{1}$ (which corresponds to shifting all logits by a constant and hence does not change $\mathbf{p}$). On the tangent space, the directions that actually change probabilities, $\mathfrak{J}(\mathbf{p})$ is full rank.

**Lemma I.3** (A right inverse of $\mathfrak{J}$ on the tangent)**.** *For an interior point* $\mathbf{p}$*, and for any vector* $v \in T_{\mathbf{p}}\Delta^{d-1}$,

$$\mathfrak{J}(\mathbf{p})\,\mathrm{Diag}(\mathbf{p})^{-1}v \;=\; \left(I - \mathbf{p}\,\mathbf{1}^{\top}\right)v \;=\; v.$$

*Equivalently,* $\mathrm{Diag}(\mathbf{p})^{-1}$ *behaves like* $\mathfrak{J}(\mathbf{p})^{-1}$ *once we restrict attention to tangent directions.*

*Proof.* By direct computation,

$$\mathfrak{J}(\mathbf{p})\,\mathrm{Diag}(\mathbf{p})^{-1} = (\mathrm{Diag}(\mathbf{p}) - \mathbf{p}\mathbf{p}^{\top})\,\mathrm{Diag}(\mathbf{p})^{-1} = I - \mathbf{p}\,\mathbf{1}^{\top},$$

using $\mathbf{p}^{\top}\mathrm{Diag}(\mathbf{p})^{-1} = \mathbf{1}^{\top}$. If $v \in T_{\mathbf{p}}\Delta^{d-1}$, then $\mathbf{1}^{\top}v = 0$, so $(I - \mathbf{p}\,\mathbf{1}^{\top})v = v$. $\qquad\square$

Up to now, $\mathfrak{J}(\mathbf{p})$ has appeared as a purely algebraic object (a Jacobian with a convenient nullspace). Next we endow the simplex with a Riemannian metric for which $\mathfrak{J}(\mathbf{p})$ becomes the natural "inverse metric" on the tangent space.

**Definition I.4** (Shahshahani metric)**.** For an interior point $\mathbf{p}$, the Shahshahani inner product on $T_{\mathbf{p}}\Delta^{d-1}$ is defined as

$$\langle u, v \rangle_{\mathrm{Shah}} := u^{\top}\mathrm{Diag}(\mathbf{p})^{-1}v = \sum_{i=1}^{d} \frac{u_i v_i}{p_i}, \, u, v \in T_{\mathbf{p}}\Delta^{d-1}. \tag{69}$$

**Geometric and information-theoretic intuition.** The Shahshahani metric rescales each coordinate by $1/\sqrt{p_i}$: moving a small component is "expensive," and the induced norm

$$\|\delta\|_{\mathbf{p}}^2 = \langle \delta, \delta \rangle_{\mathrm{Shah}} = \sum_{i=1}^{d} \frac{\delta_i^2}{p_i}$$

measures *relative* (multiplicative) change. A convenient way to visualize geodesics is the square-root embedding $p \mapsto \sqrt{p}$ (componentwise), which maps the simplex interior to the positive orthant of the unit sphere. Under this embedding, Shahshahani geodesics become great-circle arcs, and the induced geodesic distance is the Bhattacharyya angle

$$d_{\mathrm{Shah}}(x, y) \;=\; 2\arccos\!\left(\sum_{i=1}^{d} \sqrt{x_i y_i}\right). \tag{70}$$

This geometry is also tied to information theory: the metric tensor is the Hessian of the convex potential $\psi(\mathbf{p}) = \sum_{i=1}^{d} p_i \log p_i$ (negative Shannon entropy),

$$\nabla^2 \psi(\mathbf{p}) \;=\; \mathrm{Diag}\!\left(\tfrac{1}{p_1}, \ldots, \tfrac{1}{p_d}\right), \tag{71}$$

so the forward KL divergence has the local quadratic expansion

$$D_{\mathrm{KL}}(\mathbf{p} + \delta \,\|\, \mathbf{p}) \;=\; \tfrac{1}{2}\delta^{\top}\mathrm{Diag}(\mathbf{p})^{-1}\delta + o(\|\delta\|^2),\, \mathbf{1}^{\top}\delta = 0, \tag{72}$$

which is precisely why this metric underlies mirror descent and multiplicative-weights updates.

Once a metric is fixed, gradients become metric-dependent: the Shahshahani gradient is the unique tangent vector whose inner product against any tangent direction matches the directional derivative.

**Riemannian gradient under the Shahshahani metric.** For a function $F : \Delta^{d-1} \to \mathbb{R}$, the Shahshahani gradient $\mathrm{grad}_{\mathrm{Shah}}F(\mathbf{p}) \in T_{\mathbf{p}}\Delta^{d-1}$ is defined by

$$\langle \mathrm{grad}_{\mathrm{Shah}}F(\mathbf{p}), u \rangle_{\mathrm{Shah}} \;=\; \nabla F(\mathbf{p})^{\top}u, \qquad \forall u \in T_{\mathbf{p}}\Delta^{d-1},$$

where $\nabla F(\mathbf{p})$ denotes the Euclidean gradient in the ambient space.

The next corollary shows the pleasant surprise: under the Shahshahani metric, the gradient is obtained by applying $\mathfrak{J}(\mathbf{p})$ to the Euclidean gradient, so the same matrix that appears in the softmax Jacobian also governs natural-gradient flow on the simplex.

**Corollary I.5** (Natural gradient on the simplex)**.** *If* $F : \Delta^{d-1} \to \mathbb{R}$ *is* $C^1$ *and* $\mathbf{p}$ *is interior, then*

$$\mathrm{grad}_{\mathrm{Shah}}F(\mathbf{p}) = \mathfrak{J}(\mathbf{p})\,\nabla F(\mathbf{p})$$
$$= \mathbf{p} \odot \left(\nabla F(\mathbf{p}) - \langle \mathbf{p}, \nabla F(\mathbf{p}) \rangle\,\mathbf{1}\right) \in T_{\mathbf{p}}\Delta^{d-1}.$$

*Proof.* Let $g := \mathfrak{J}(\mathbf{p})\nabla F(\mathbf{p})$. From Lemma I.1, we have $g \in T_{\mathbf{p}}\Delta^{d-1}$. For any $u \in T_{\mathbf{p}}\Delta^{d-1}$, symmetry of $\mathfrak{J}(\mathbf{p})$ and Lemma I.3 yield

$$\langle g, u \rangle_{\mathrm{Shah}} = g^{\top}\mathrm{Diag}(\mathbf{p})^{-1}u$$
$$= \nabla F(\mathbf{p})^{\top}\mathfrak{J}(\mathbf{p})\,\mathrm{Diag}(\mathbf{p})^{-1}u$$
$$= \nabla F(\mathbf{p})^{\top}u.$$

Thus $g$ satisfies the defining property of $\mathrm{grad}_{\mathrm{Shah}}F(\mathbf{p})$, proving the claim. The componentwise form follows by expanding $\mathfrak{J}(\mathbf{p})\nabla F(\mathbf{p}) = \mathrm{Diag}(\mathbf{p})\nabla F(\mathbf{p}) - \mathbf{p}(\mathbf{p}^{\top}\nabla F(\mathbf{p}))$. $\qquad\square$

The following lemma states that $\mathfrak{J}(\mathbf{p})$ is strictly positive on tangent directions, so it can safely serve as an inverse metric (and as a preconditioner) as long as we stay in the simplex interior.

**Lemma I.6** (Positive definiteness of $\mathfrak{J}$ on the tangent)**.** *For any interior* $\mathbf{p}$ *and any nonzero* $v \in T_{\mathbf{p}}\Delta^{d-1}$,

$$v^{\top}\mathfrak{J}(\mathbf{p})\,v \;=\; \sum_{i=1}^{d} p_i v_i^2 - \left(\sum_{i=1}^{d} p_i v_i\right)^2 \;=\; \mathrm{Var}_{i \sim p}(v_i) \;>\; 0.$$

*Thus,* $\mathfrak{J}(\mathbf{p})$ *is symmetric positive definite when restricted to the tangent space.*

**Remark on boundary points.** If some components satisfy $p_i = 0$, the statements remain valid after restricting to the support of $\mathbf{p}$ and the corresponding lower-dimensional face (where $\mathrm{Diag}(\mathbf{p})^{-1}$ is well-defined).

So far we have described the geometry (metric) and the resulting notion of gradient (natural gradient). We now connect this viewpoint to the standard KL-regularized update used by mirror descent / multiplicative weights, and show that it matches the Shahshahani natural-gradient flow to first order.

**Proposition I.7** (Entropic mirror-ascent on the simplex). *For $\mathbf{p} \in \Delta^{d-1}$, a vector $A \in \mathbb{R}^d$, and a step size $\eta > 0$, the KL-regularized maximization problem*

$$\mathbf{p}^+ = \arg\max_{\mathbf{q} \in \Delta^{d-1}} \left\{ \langle A, \mathbf{q} \rangle - \tfrac{1}{\eta} D_{\mathrm{KL}}(\mathbf{q} \,\|\, \mathbf{p}) \right\} \quad (73)$$

*satisfies the following properties:*

1. **Existence and uniqueness.** *The objective is strictly concave on the relative interior of the face determined by $\mathbf{p}$, hence the maximizer $\mathbf{p}^+$ exists and is unique.*

2. **Closed-form update (multiplicative weights).** *Let $Z := \sum_{j=1}^{d} p_j \exp(\eta A_j)$. Then*

   $$p_i^+ = \frac{p_i \exp(\eta A_i)}{Z}, \qquad i = 1, \ldots, d. \quad (74)$$

   *Equivalently, $\log p_i^+ = \log p_i + \eta A_i - \log Z$.*

3. **Trust-region equivalence.** *For any $\rho > 0$, there exists $\lambda > 0$ such that $\mathbf{p}^+$ also solves $\max_{\mathbf{q} \in \Delta^{d-1}} \{ \langle A, \mathbf{q} \rangle : D_{\mathrm{KL}}(\mathbf{q}\|\mathbf{p}) \leq \rho \}$ with $\eta = 1/\lambda$.*

4. **Optimal value.** *The maximum value of the objective equals $\tfrac{1}{\eta} \log Z$.*

5. **Invariance and support.** *The update is invariant under shifts $A \mapsto A + c\mathbf{1}$ and does not create new support in one step.*

6. **Improvement via Jeffreys divergence.** *With $\bar{A} := \langle \mathbf{p}, A \rangle$ and $J_{\mathrm{KL}}(\mathbf{p}^+, \mathbf{p}) := D_{\mathrm{KL}}(\mathbf{p}^+\|\mathbf{p}) + D_{\mathrm{KL}}(\mathbf{p}\|\mathbf{p}^+)$,*
   $$\eta \langle A, \mathbf{p}^+ - \mathbf{p} \rangle = J_{\mathrm{KL}}(\mathbf{p}^+, \mathbf{p}) \geq 0.$$

7. **First-order expansion (replicator direction).** *For small $\eta$,*
   $$\mathbf{p}^+ - \mathbf{p} = \eta \, \mathfrak{J}(\mathbf{p}) \, A + O(\eta^2). \quad (75)$$

8. **Local objective gain.** $\langle A, \mathbf{p}^+ - \mathbf{p} \rangle = \eta \, \mathrm{Var}_{i \sim p}(A_i) + O(\eta^2)$.

*Proof.* These claims follow from standard Lagrangian optimality conditions and a Taylor expansion of (74). In particular, stationarity yields $q_i \propto p_i e^{\eta A_i}$ and hence (74), while the Jeffreys identity follows by summing $D_{\mathrm{KL}}(\mathbf{p}^+\|\mathbf{p})$ and $D_{\mathrm{KL}}(\mathbf{p}\|\mathbf{p}^+)$. $\square$

**Closing the loop.** The final corollary makes the connection explicit: mirror-ascent is an Euler discretization of Shahshahani natural-gradient flow, which is exactly the replicator-form dynamics that appear in our mean-field ODEs.

**Corollary I.8** (Natural-gradient interpretation). *The mirror-ascent step* (73) *is equivalent, to first order, to an Euler step of the Shahshahani natural-gradient flow:*

$$\dot{\mathbf{p}} = \mathfrak{J}(\mathbf{p})A = \mathbf{p} \odot \big( A - \langle \mathbf{p}, A \rangle \, \mathbf{1} \big).$$

# J. Inner Dynamics of the Good Arms

In Appendix D, we established the mean-field dynamics of the good and bad blocks in the explicit $(K+M)$-arm model (with $K$ good arms and $M$ bad arms), and in particular we introduced the within-block coordinates

$$p(t) := \sum_{m=1}^{M} p_{b_m}(t) \in [0, 1],$$

$$y_j(t) := \frac{p_j(t)}{1 - p(t)}, \qquad j \leq K,$$

$$z_m(t) := \frac{p_{b_m}(t)}{p(t)}, \qquad m \leq M.$$

In this section, we go into the details of the *inner good-arm* dynamics, i.e. the evolution of $\mathbf{y}(t) \in \Delta^{K-1}$. (The corresponding inner bad-arm dynamics for $\mathbf{z}(t)$ is the sign-reversed analogue and is recorded separately.)

To start, we re-derive the inner good dynamics ODE in a more intuitive way. Let $p_i = \exp(\theta_i)/Z$ with

$$Z = \sum_{k=1}^{K} \exp(\theta_k) + \sum_{m=1}^{M} \exp(\theta_{b_m}), \qquad p := \sum_{m=1}^{M} p_{b_m},$$

and define

$$y_j := \frac{p_j}{1 - p}, \qquad j \in [K].$$

Then

$$y_j = \frac{\exp(\theta_j)}{\sum_{k=1}^{K} \exp(\theta_k)} = \big( \mathrm{softmax}(\boldsymbol{\theta}_{\mathrm{good}}) \big)_j, \quad (76)$$

so $\mathbf{y} = (y_1, \cdots, y_K)$ depends only on $\boldsymbol{\theta}_{\mathrm{good}}$ (the bad-block logits cancel by normalization).

**Lemma J.1** (Pushforward from logits to within-good composition). *For any small increment $\Delta\boldsymbol{\theta} = (\Delta\boldsymbol{\theta}_{\mathrm{good}}, \Delta\boldsymbol{\theta}_{\mathrm{bad}})$,*

$$\Delta\mathbf{y} = \big( \mathrm{Diag}(\mathbf{y}) - \mathbf{y}\mathbf{y}^\top \big) \Delta\boldsymbol{\theta}_{\mathrm{good}}$$
$$= \mathbf{y} \odot \big( \Delta\boldsymbol{\theta}_{\mathrm{good}} - \langle \mathbf{y}, \Delta\boldsymbol{\theta}_{\mathrm{good}} \rangle \mathbf{1} \big). \quad (77)$$

and in particular $\partial \mathbf{y}/\partial \theta_{b_m} = \mathbf{0}$ *for all* $m \in [M]$ *(equivalently,* $\partial \mathbf{y}/\partial \boldsymbol{\theta}_{\mathrm{bad}} = \mathbf{0}$*).*

Proof. *From* (76), *write* $y_j = \exp(\theta_j - L)$ *with* $L := \log \sum_{k \leq K} \exp(\theta_k)$, *so* $\Delta y_j = y_j(\Delta \theta_j - \Delta L)$ *and* $\Delta L = \sum_{k \leq K} y_k \Delta \theta_k$. *Stacking over* $j$ *gives* (77).

Similar to our discussion in Appendix D, assume block symmetry with $A_j = a_{\mathrm{g}}(p)$ for $j \leq K$ and $A_{b_m} = a_{\mathrm{b}}(p)$ for $m \leq M$, and set $\Delta r(p) = a_{\mathrm{b}}(p) - a_{\mathrm{g}}(p)$. The expected logit step in the good block is

$$\mathbb{E}[\Delta \boldsymbol{\theta}_{\mathrm{good}}] = \kappa(p)\,\mathbf{y}, \qquad \kappa(p) := -\eta\, p(1-p)\,\Delta r(p), \tag{78}$$

so there is no arm-specific preference in $\theta$-space. Applying Lemma J.1 leads to

$$\begin{aligned} \mathbb{E}[\Delta \mathbf{y}] &= \kappa(p)\left(\mathbf{y} \odot \mathbf{y} - \|\mathbf{y}\|_2^2\,\mathbf{y}\right), \\ \mathbb{E}[\Delta y_j] &= \kappa(p)\, y_j\left(y_j - \|\mathbf{y}\|_2^2\right). \end{aligned} \tag{79}$$

If $a_{\mathrm{g}}(p) = \frac{Jp}{\sigma(p)}$ and $a_{\mathrm{b}}(p) = -\frac{J(1-p)}{\sigma(p)}$ then $\Delta r(p) = -J/\sigma(p)$ and (79) becomes

$$\mathbb{E}[\Delta y_j] = \eta\,\frac{J}{\sigma(p)}\,p(1-p)\,y_j\left(y_j - \|\mathbf{y}\|_2^2\right),$$

*Consequences.* If $J > 0$, arms with $y_j > \|\mathbf{y}\|_2^2$ grow while those with $y_j < \|\mathbf{y}\|_2^2$ shrink (deterministic sharpening within the good block). The fixed points in $\mathbf{y}$ are the uniform point and the vertices. If $\Delta r(p) < 0$ (informative grader favoring the good block), the uniform point is unstable and the vertices are attracting.[1]

**Lemma J.2** (Geometry and Lyapunov structure). *Consider the ODE on the simplex*

$$\dot{\mathbf{y}} = \kappa(t)\left(\mathbf{y} \odot \mathbf{y} - \|\mathbf{y}\|_2^2\,\mathbf{y}\right), \tag{80}$$

$$\mathbf{y} = (y_1, \dots, y_K) \in \Delta^{K-1} := \Big\{ \mathbf{y} \geq 0 : \sum_{i=1}^K y_i = 1 \Big\}.$$

*Let* $\tau(t) := \int_0^t \kappa(s)\,ds$ *and write* $\mathbf{y}' = \frac{d\mathbf{y}}{d\tau}$*. (E.g., here in our noisy GRPO dynamics,* $\kappa(t) \propto \frac{J}{\sigma(p(t))} p(t)(1-p(t))$ *up to the chosen mean-field time scaling.)*

(1) ***Simplex invariance.*** $\sum_i \dot{y}_i = 0$*, and if* $y_i(0) \geq 0$ *then* $y_i(t) \geq 0$ *for all* $t$*. Hence* $\Delta^{K-1}$ *is forward invariant.*

[1] It is interesting to notice that if one uses the algorithm that uses natural-gradient/replicator flow instead, $\dot{\mathbf{p}} = \eta\, \mathfrak{J}(\mathbf{p})\mathbf{A}$ with $\mathfrak{J}(\mathbf{p}) = \mathrm{Diag}(\mathbf{p}) - \mathbf{p}\mathbf{p}^\top$, then under block symmetry

$$\dot{y}_j = y_j\big(a_{\mathrm{g}} - \bar{A}_{\mathrm{g}}\big) = 0,$$

so $\mathbf{y}$ is exactly drift-free deterministically and only sampling noise perturbs it.

(2) ***Gradient form.*** *Define*

$$\mathcal{L}(\mathbf{y}) := \frac{1}{3}\sum_{i=1}^K y_i^3 - \frac{1}{4}\Big(\sum_{i=1}^K y_i^2\Big)^2 = \frac{1}{3}s_3 - \frac{1}{4}s_2^2,$$

*such that* $s_2(t) := \|\mathbf{y}(t)\|_2^2 = \sum_i y_i^2$ *and* $s_3(t) := \sum_i y_i^3$*, then* $\nabla\mathcal{L}(\mathbf{y}) = (y_i^2 - \|\mathbf{y}\|_2^2\, y_i)_i$ *and*

$$\dot{\mathbf{y}} = \kappa(t)\,\nabla\mathcal{L}(\mathbf{y}), \qquad \frac{d}{dt}\mathcal{L}\big(\mathbf{y}(t)\big) = \kappa(t)\,\|\nabla\mathcal{L}(\mathbf{y})\|_2^2.$$

*In particular, when* $J > 0$*, then* $\kappa(t) \geq 0$ *and the flow monotonically ascends* $\mathcal{L}$ *(and descends it when* $J < 0$*).*

(3) ***Monotone concentration.***

$$\dot{s}_2 = 2\kappa(t)\Big(\sum_i y_i^3 - s_2^2\Big) \geq 0 \quad \text{whenever } \kappa(t) \geq 0,$$

*because by Cauchy–Schwarz,* $\sum_i y_i^3 \geq (\sum_i y_i^2)^2$ *on the simplex, with equality iff* $\mathbf{y}$ *is uniform on its support. Hence for* $\kappa \geq 0$ *the mass generically concentrates while when* $\kappa < 0$*,* $s_2$ *decreases.*

(4) ***Equilibria.*** *Stationary points satisfy* $y_i \in \{0, s_2\}$ *for all* $i$*. Thus for any* $m \in \{1, \dots, K\}$*, the points with exactly* $m$ *nonzero coordinates all equal to* $1/m$ *are equilibria (uniform on a support of size* $m$*). For* $\kappa > 0$*, the* $m = 1$ *vertices are (Lyapunov) attractors and the others are saddles.*

*Remark* J.3 (Time reparametrization). All phase-portrait statements are independent of $t$ and depend only on the internal time $\tau$. In $\tau$-time the ODE is autonomous:

$$\frac{d\mathbf{y}}{d\tau} = \mathbf{y} \odot \mathbf{y} - \|\mathbf{y}\|_2^2\,\mathbf{y} = \mathbf{y} \odot \mathbf{y} - s_2\mathbf{y}. \tag{81}$$

Solutions in physical time are obtained by composing with $\tau(t)$.[2] The system from (81) can be solved explicitly as:

$$y_i(\tau) = \frac{\frac{d\gamma}{d\tau}}{\frac{1}{y_i(0)} - \gamma(\tau)}$$

where $\gamma(\tau)$ satisfies $\gamma(0) = 0$ and

$$\prod_{i=1}^K \left(\frac{1}{y_i(0)} - \gamma(\tau)\right) = \frac{1}{\prod_{i=1}^K y_i(0)} e^{-\tau}.$$

[2] We can see how the closed form solution of this ODE look like for simple examples. *A: Closed form on the 2-arm face* ($K = 2$). On the line $\{(q, 1-q) : q \in [0,1]\}$ one has $\frac{dq}{d\tau} = q(1-q)(2q-1)$. The separating-variables step gives $\int \frac{dq}{q(1-q)(2q-1)} = \tau + C \implies \log\left(\frac{(2q-1)^2}{q(1-q)}\right) = \tau + \log G_0$, where $G_0 = \frac{(2q_0-1)^2}{q_0(1-q_0)}$. Hence the explicit solution $q(\tau) = \frac{1}{2}\left(1 \pm \sqrt{\frac{G_0\, e^\tau}{4 + G_0\, e^\tau}}\right)$. Choose $+$ if $q_0 > \frac{1}{2}$ (converges to 1 for $\kappa > 0$) and $-$ if $q_0 < \frac{1}{2}$.

Similarly, for *B. One-vs-rest symmetric slice for general* $K \geq 2$ *(full partial fractions)*. Restrict to the 1-D symmetric manifold

By the Implicit Function Theorem, there exists a unique strictly increasing function $\gamma : [0, \infty) \to \left[0, \frac{1}{\max_{1 \le i \le K} y_i(0)}\right)$ with these properties, and it tends to $\frac{1}{\max_{1 \le i \le K} y_i(0)}$ as $\tau \to \infty$. In particular, assuming we have $m$ highest initial values among $y_1(0), \ldots, y_K(0)$, as $\tau \to \infty$, $y_i(\tau)$'s with highest initial values tend to $\frac{1}{m}$ and the rest tend to 0.

We will continue the dynamics of good arms in more details in the following subsection. J.5

## J.1. Evolution of Collision term, $s_2$

**Lemma J.4** (Evolution and bounds for $s_2 = \|\mathbf{y}\|_2^2$). *In $\tau$-time one has*

$$\frac{d}{d\tau} s_2 = 2(s_3 - s_2^2), \qquad \frac{d}{dt} s_2 = 2\kappa(t)(s_3 - s_2^2).$$

*Consequently:*

1. *(Monotonicity) On the simplex, $s_3 \ge s_2^2$ with equality iff $\mathbf{y}$ is uniform on its support. Hence $s_2(\tau)$ is nondecreasing (strictly, away from uniform-on-support points) when $\kappa \ge 0$.*

2. *(Range) $\frac{1}{K} \le s_2(\tau) \le 1$. In the multi-bad setting, defining $t_2(\tau) := \|\mathbf{z}(\tau)\|_2^2 \in [\frac{1}{M}, 1]$, we have the uniform bound*

$$\frac{1}{K} + \frac{1}{M} \le s_2(\tau) + t_2(\tau) \le 2.$$

3. *(Logistic upper differential) Since $0 \le y_i \le 1$, $s_3 \le s_2$,*

$$\mathbf{y}(\tau) = \left(x(\tau), \frac{1-x(\tau)}{K-1}, \ldots, \frac{1-x(\tau)}{K-1}\right), \qquad x \in [0, 1]. \text{ Then}$$

$$\|\mathbf{y}\|_2^2 = x^2 + \frac{(1-x)^2}{K-1},$$

$$\frac{dx}{d\tau} = x^2 - x\|\mathbf{y}\|_2^2 = \frac{1}{K-1} x(1-x)(Kx - 1).$$

Separate variables and partial-fractions decomposition gives: $\frac{(Kx(\tau)-1)^K}{x(\tau)^{K-1}(1-x(\tau))} = \frac{(Kx_0-1)^K}{x_0^{K-1}(1-x_0)} e^\tau := G_0 e^\tau$. with fixing the constant with $x(0) = x_0 \in (0, 1)$. Equilibria on this slice are $x \in \{0, 1/K, 1\}$. For $\kappa > 0$, $x(\tau) \to 0$ if $x_0 < 1/K$, and $x(\tau) \to 1$ if $x_0 > 1/K$.

For $K = 3$ the invariant in (B) gives the explicit solution

$$x(\tau) = \frac{1}{3}(1 + u(\tau) + v(\tau)),$$

$$u(\tau) := \sqrt[3]{z(\tau)(1 + \sqrt{1 - z(\tau)})},$$

$$v(\tau) := \sqrt[3]{z(\tau)(1 - \sqrt{1 - z(\tau)})},$$

$$z(\tau) = \frac{G_0 e^\tau}{27 + G_0 e^\tau}.$$

*hence*

$$\frac{d}{d\tau} s_2 \le 2 s_2(1 - s_2), s_2(\tau) \le \frac{1}{1 + \left(\frac{1-s_2(0)}{s_2(0)}\right) e^{-2\tau}}.$$

*Integrating,*

$$\int_0^\tau s_2(u) \, du \le \frac{1}{2} \log\left(1 + s_2(0)(e^{2\tau} - 1)\right).$$

**Corollary J.5** (Exact logit representation and envelopes for $p$ (multi-bad setting)). *Let $p(\tau) \in (0, 1)$ be the total bad mass and let $\mathbf{z}(\tau) \in \Delta^{M-1}$ be the within-bad composition, so $t_2(\tau) := \|\mathbf{z}(\tau)\|_2^2 \in [1/M, 1]$. In $\tau$-time the $p$-equation reads*

$$\frac{dp}{d\tau} = -p(1 - p)(s_2(\tau) + t_2(\tau)).$$

*For the logit $L(\tau) := \log \frac{p(\tau)}{1-p(\tau)}$,*

$$\frac{dL}{d\tau} = -(s_2(\tau) + t_2(\tau)),$$

$L(\tau) = L(0) - \int_0^\tau (s_2(u) + t_2(u)) \, du$. *Because $s_2 \in \left[\frac{1}{K}, 1\right]$ and $t_2 \in \left[\frac{1}{M}, 1\right]$, comparison yields the envelopes*

$$\frac{p_0}{1 - p_0} e^{-2\tau} \le \frac{p(\tau)}{1 - p(\tau)} \le \frac{p_0}{1 - p_0} e^{-(\frac{1}{K} + \frac{1}{M})\tau},$$

*equivalently*

$$\frac{1}{1 + \frac{1-p_0}{p_0} e^{2\tau}} \le p(\tau) \le \frac{1}{1 + \frac{1-p_0}{p_0} e^{(\frac{1}{K} + \frac{1}{M})\tau}}.$$

**Corollary J.6** (Hitting-time bracket in internal time (multi-bad setting)). *Fix $p_\star \in (0, 1)$. The internal time to reach $p(\tau_\star) = p_\star$ is bounded by*

$$\frac{1}{2} \log \frac{p_0(1 - p_\star)}{(1 - p_0)p_\star} \le \tau_\star \le \frac{1}{\frac{1}{K} + \frac{1}{M}} \log \frac{p_0(1 - p_\star)}{(1 - p_0)p_\star}.$$

*Thus the factor $\|\mathbf{y}\|_2^2 + \|\mathbf{z}\|_2^2$ accelerates the decay of $\log$it $p$ by a multiplicative factor between $\frac{1}{K} + \frac{1}{M}$ (near-uniform within both blocks) and 2 (maximally concentrated within both blocks).*

**Theorem J.7** (General-$(K, M)$ small-heterogeneity expansion). *Let $\text{unif}_K = (1/K, \ldots, 1/K)$ and $\text{unif}_M = (1/M, \ldots, 1/M)$ and write*

$$\mathbf{y} = \text{unif}_K + \mathbf{v}, \quad \sum_{i=1}^K v_i = 0,$$

$$\mathbf{z} = \text{unif}_M + \mathbf{w}, \quad \sum_{m=1}^M w_m = 0.$$

*Define the heterogeneities*

$$\zeta(\tau) := \|\mathbf{v}(\tau)\|_2^2 = s_2(\tau) - \frac{1}{K} \ge 0, \qquad \zeta_0 := \zeta(0),$$

$$\xi(\tau) := \|\mathbf{w}(\tau)\|_2^2 = t_2(\tau) - \frac{1}{M} \geq 0, \qquad \xi_0 := \xi(0).$$

*Then in internal time $\tau$,*

$$\frac{dL}{d\tau} = -\big(s_2(\tau) + t_2(\tau)\big)$$
$$= -\Big(\tfrac{1}{K} + \tfrac{1}{M} + \zeta(\tau) + \xi(\tau)\Big),$$
$$L(\tau) = L(0) - \int_0^\tau \big(s_2(u) + t_2(u)\big)\, du.$$

*Moreover:*

(i) ***Linearized $\zeta$-law (good block).*** *One has the identity*

$$\zeta'(\tau) = \frac{2}{K}\,\zeta(\tau) - 2\,\zeta(\tau)^2 + 2\sum_{i=1}^K v_i(\tau)^3,$$

*and the bound $\big|\sum_i v_i^3\big| \leq \sum_i |v_i|^3 \leq \|\mathbf{v}\|_2^3 = \zeta^{3/2}$. Hence, uniformly while $\zeta(\tau) \leq 1$,*

$$\zeta'(\tau) = \frac{2}{K}\,\zeta(\tau) + O\big(\zeta(\tau)^{3/2}\big).$$

(ii) ***Linearized $\xi$-law (bad block).*** *Because the within-bad dynamics are the sign-reversed collision flow, one analogously has*

$$\xi'(\tau) = -\frac{2}{M}\,\xi(\tau) + 2\,\xi(\tau)^2 - 2\sum_{m=1}^M w_m(\tau)^3,$$

*and $\big|\sum_m w_m^3\big| \leq \|\mathbf{w}\|_2^3 = \xi^{3/2}$. Hence, uniformly while $\xi(\tau) \leq 1$,*

$$\xi'(\tau) = -\frac{2}{M}\,\xi(\tau) + O\big(\xi(\tau)^{3/2}\big).$$

(iii) ***Asymptotic forms.*** *There exist constants $C_K, C_M > 0$ (depending only on $K$ and $M$) such that, for all $\tau \geq 0$ with $\sqrt{\zeta_0}\, e^{\tau/K} \leq \frac{1}{2}$,*

$$\zeta(\tau) = \zeta_0\, e^{\frac{2}{K}\tau} + R_\zeta(\tau),\ |R_\zeta(\tau)| \leq C_K\, \zeta_0^{3/2}\, e^{\frac{3}{K}\tau},$$

*and, for all $\tau \geq 0$ with $\sqrt{\xi_0} \leq \frac{1}{2}$,*

$$\xi(\tau) = \xi_0\, e^{-\frac{2}{M}\tau} + R_\xi(\tau),\ |R_\xi(\tau)| \leq C_M\, \xi_0^{3/2}\, e^{-\frac{3}{M}\tau}.$$

(iv) ***Impact on the $p$-logit.*** *Consequently,*

$$\int_0^\tau s_2(u)\, du = \frac{\tau}{K} + \frac{K}{2}\,\zeta_0\Big(e^{\frac{2}{K}\tau} - 1\Big) + R_{I,y}(\tau),$$
$$\big|R_{I,y}(\tau)\big| \leq C_K'\, \zeta_0^{3/2}\, e^{\frac{3}{K}\tau}.$$

$$\int_0^\tau t_2(u)\, du = \frac{\tau}{M} + \frac{M}{2}\,\xi_0\Big(1 - e^{-\frac{2}{M}\tau}\Big) + R_{I,z}(\tau),$$
$$\big|R_{I,z}(\tau)\big| \leq C_M'\, \xi_0^{3/2}.$$

*and hence*

$$L(\tau) = L(0) - \Big(\tfrac{1}{K} + \tfrac{1}{M}\Big)\tau$$
$$- \frac{K}{2}\,\zeta_0\Big(e^{\frac{2}{K}\tau} - 1\Big) - \frac{M}{2}\,\xi_0\Big(1 - e^{-\frac{2}{M}\tau}\Big) + R_L(\tau).$$

*with $|R_L(\tau)| \leq C''_{K,M}\big(\zeta_0^{3/2} e^{3\tau/K} + \xi_0^{3/2}\big)$ for a constant $C''_{K,M}$.*

*In particular, to first order the only dependence on the initial within-block states is through $\zeta_0 = \|\mathbf{y}(0) - \mathrm{unif}_K\|_2^2$ and $\xi_0 = \|\mathbf{z}(0) - \mathrm{unif}_M\|_2^2$; finer details of $\mathbf{y}(0)$ and $\mathbf{z}(0)$ enter only at order $O(\zeta_0^{3/2})$ and $O(\xi_0^{3/2})$ and higher.*

**Corollary J.8** (Small-$\tau$ expansion; "only $(\zeta_0, \xi_0)$ matters" at first order)**.** *Expanding the expression in Theorem J.7 for small $\tau$ gives*

$$L(\tau) = L(0) - \Big(\tfrac{1}{K} + \tfrac{1}{M}\Big)\tau - (\zeta_0 + \xi_0)\tau - \Big(\tfrac{\zeta_0}{K} - \tfrac{\xi_0}{M}\Big)\tau^2$$
$$+ O(\zeta_0\, \tau^3) + O(\xi_0\, \tau^3) + O(\zeta_0^{3/2}\, \tau) + O(\xi_0^{3/2}\, \tau).$$

*Thus, to linear order in $\tau$, the correction to the baseline $-(\tfrac{1}{K} + \tfrac{1}{M})\tau$ is exactly $-(\zeta_0 + \xi_0)\tau$. Equivalently, at leading order,*

$$\frac{dL}{d\tau} = -(s_2(0) + t_2(0)) + \textit{higher-order corrections.}$$

*Remark J.9* (Scope and sign). (i) The $\tau$-phase portraits for $\mathbf{y}$ are independent of the path of $p(t)$ (since $p$ only reparametrizes time through $\tau$). (ii) If $\kappa(t) \leq 0$ (e.g. $J/\sigma(p) \leq 0$), replace $\tau$ by $-|\tau|$ to flip directions in physical time; the internal-time identities remain valid. (iii) The window $\sqrt{\zeta_0}\, e^{\tau/K} \leq \frac{1}{2}$ describes the regime where the linearized law for $\zeta$ dominates; beyond it, the global envelopes in Cor. J.5 apply.

### J.2. Asymptotic General inner dynamics behavior

**Lemma J.10** (Order preservation and winner identity)**.** *For any $i \neq j$, define $\delta_{ij}(\tau) := y_i(\tau) - y_j(\tau)$. Along $\frac{d\mathbf{y}}{d\tau} = \mathbf{y} \odot \mathbf{y} - s_2\mathbf{y}$,*

$$\delta'_{ij} = \delta_{ij}\big(y_i + y_j - s_2\big),$$
$$\delta_{ij}(\tau) = \delta_{ij}(0)\, \exp\Big(\int_0^\tau \big(y_i + y_j - s_2\big)\, du\Big).$$

*Hence $\operatorname{sign}\delta_{ij}(\tau) = \operatorname{sign}\delta_{ij}(0)$ for all $\tau$. In particular,*

$$m := \arg\max_{1 \leq i \leq K} y_i(0)$$

*remains the unique maximizer for all $\tau > 0$, and each hyperplane $\{y_i = y_j\}$ is invariant.*

**Theorem J.11** (Global convergence and generic basins). *Under* (81), *the trajectory stays in* $\Delta^{K-1}$ *and* $\mathcal{L}$ *is a strict Lyapunov function. The equilibria are*

$$\mathcal{E} = \bigcup_{m=1}^{K} \mathcal{E}_m,$$

$$\mathcal{E}_m := \left\{ \mathbf{y} : y_{i_1} = \cdots = y_{i_m} = \tfrac{1}{m}, \ y_j = 0 \ \textit{otherwise} \right\}.$$

*For generic initial conditions (no coordinate ties), the trajectory converges to the vertex* $\mathbf{e}_m$ *selected by Lemma J.10. The non-vertex equilibria (uniform on* $m$-*subsupports with* $m \geq 2$*) are saddles with co-dimension one stable manifolds coinciding with unions of the tie sets* $\{y_i = y_j\}$.

**Proposition J.12** (Exponential polarization in internal time). *Let* $m = \arg\max y_i(0)$ *and write* $\varepsilon_i(\tau) := y_i(\tau)$ *for* $i \neq m$. *Linearizing* (81) *at the vertex* $\mathbf{e}_m$ *yields*

$$\varepsilon_i' = -\varepsilon_i + O(\varepsilon^2), \qquad 1 - y_m = \sum_{i \neq m} \varepsilon_i.$$

*Hence there exists* $\tau_0$ *and constants* $c_i > 0$ *such that, for all* $\tau \geq \tau_0$,

$$y_i(\tau) = c_i\, e^{-\tau}\bigl(1 + o(1)\bigr),$$

$$1 - y_m(\tau) = \Bigl(\sum_{i \neq m} c_i\Bigr)e^{-\tau}\bigl(1 + o(1)\bigr),$$

$$1 - s_2(\tau) = \Theta\bigl(e^{-\tau}\bigr).$$

*Remark* J.13 (Sharper monotonicity identity). The variance form

$$s_3 - s_2^2 = \sum_{i=1}^{K} y_i\,(y_i - s_2)^2 \geq 0$$

(with equality iff $\mathbf{y}$ is uniform on its support) implies $s_2'(\tau) = 2\bigl(s_3 - s_2^2\bigr) \geq 0$, with strict increase away from the saddle sets.

### J.3. Stability of the Within–Good Equilibria

Consider our the inner-good ODE

$$\dot{\mathbf{y}} = \kappa(p)\,\bigl(\mathbf{y} \odot \mathbf{y} - s_2\,\mathbf{y}\bigr),$$

$$s_2 = \|\mathbf{y}\|_2^2, \qquad \kappa(p) = \frac{J}{\sigma(p)}\,p(1-p). \tag{82}$$

with $\mathbf{y} \in \Delta^{K-1}$ and $p \in (0,1)$ treated as quasi–frozen on the slow time scale. We write $\mathbf{y}^\star \in \Delta^{K-1}$ for an equilibrium of (82) and work on the tangent space $T_{\mathbf{y}^\star}\Delta^{K-1} = \{\delta \in \mathbb{R}^K : \sum_j \delta_j = 0\}$.

**Lemma J.14** (Jacobian on the simplex tangent). *Let* $\mathbf{y}^\star$ *be an equilibrium of* (82) *and write* $s_2^\star = \|\mathbf{y}^\star\|_2^2$. *For*

*perturbations* $\mathbf{y} = \mathbf{y}^\star + \delta$ *with* $\sum_j \delta_j = 0$, *the linearization is*

$$\dot{\delta} = J_y\,\delta, \qquad J_y = \kappa(p)\,\Bigl(\operatorname{diag}(2\mathbf{y}^\star) - 2\mathbf{y}^\star(\mathbf{y}^\star)^\top - s_2^\star I\Bigr), \tag{83}$$

*where* $J_y$ *acts on* $T_{\mathbf{y}^\star}\Delta^{K-1}$ *(the subspace orthogonal to* $\mathbf{1}$*).*

**Proposition J.15** (Stability of the uniform equilibrium). *Consider the uniform equilibrium* $\mathbf{y}^\star = \frac{1}{K}\mathbf{1}$ *of* (82). *Then* $s_2^\star = \frac{1}{K}$ *and the Jacobian reduces to*

$$J_y = \frac{\kappa(p)}{K}\Bigl(I - \tfrac{2}{K}\mathbf{1}\mathbf{1}^\top\Bigr). \tag{84}$$

*The eigenstructure is:*

- *Along the direction* $\mathbf{1}$: *a zero eigenvalue (simplex invariance).*

- *On the* $(K-1)$-*dimensional tangent space* $T_{\mathbf{y}^\star}\Delta^{K-1}$: *eigenvalues* $\lambda = \frac{\kappa(p)}{K}$.

*Hence:*

$$\begin{cases} \kappa(p) > 0 \ (\textit{i.e., } J > 0): & \mathbf{y}^\star \textit{ is linearly unstable (repelling);} \\ \kappa(p) < 0 \ (\textit{i.e., } J < 0): & \mathbf{y}^\star \textit{ is asymptotically stable.} \end{cases}$$

**Proposition J.16** (Stability of the pure–arm equilibria). *Consider a vertex / pure–arm equilibrium* $\mathbf{y}^\star = e_j$ *of* (82), *for some* $j \in \{1, \ldots, K\}$. *Then* $s_2^\star = 1$ *and*

$$J_y = \kappa(p)\,\bigl(\operatorname{diag}(2e_j) - 2e_j e_j^\top - I\bigr). \tag{85}$$

*In coordinates this yields*

$$\dot{\delta}_j = 0, \qquad \dot{\delta}_i = -\kappa(p)\,\delta_i \quad (i \neq j),$$

*so that perturbations orthogonal to* $e_j$ *decay or grow according to the sign of* $\kappa(p)$. *In particular:*

$$\begin{cases} \kappa(p) > 0: & \mathbf{y}^\star = e_j \\ & \textit{is locally asymptotically stable} \\ & \textit{(winner–take–all);} \\ \kappa(p) < 0: & \mathbf{y}^\star = e_j \\ & \textit{is unstable} \\ & \textit{(flow returns toward mixed states).} \end{cases}$$

Since $\kappa(p) = \frac{J}{\sigma(p)}p(1-p)$ and $p(1-p) > 0$ for $p \in (0,1)$, the sign of the Youden index $J$ dictates the symmetry breaking based:

**Corollary J.17** (Stability summary for within–good equilibria). *For the inner dynamics* (82), *the stability types of the canonical equilibria are summarized in Table 2.*

| Equilibrium | $J$ | Local stability |
|---|---|---|
| Uniform $y_j = \frac{1}{K}$ | $> 0$ | **Unstable**: diversity collapses |
| Uniform $y_j = \frac{1}{K}$ | $< 0$ | **Stable**: diversity preserved |
| Vertex $\mathbf{y} = e_j$ | $> 0$ | **Stable**: specialization (single mode) |
| Vertex $\mathbf{y} = e_j$ | $< 0$ | **Unstable**: reverts to a mixture |

*Table 2.* Local stability of uniform and pure–arm equilibria for the within–good ODE (82).

*Remark* J.18 (Role of the Youden index $J$). We can see the effect of noise as

- $J > 0$ (reward alignment): the uniform mixture is destabilized, pure–arm vertices become attractors, and the good arms polarize; diversity collapses.

- $J < 0$ (reward inversion): the uniform mixture is stabilized while vertices are repelling; diversity is maintained.

### J.4. Coupling back to physical time

Let $p(t) \in (0, 1)$ be the *total* bad mass and define $\kappa(t) = \frac{J}{\sigma(p(t))} p(t)\big(1 - p(t)\big)$. Then $d\tau/dt = \kappa(t)$ and (in the multi-bad model) the coupled equations read

$$\dot{\mathbf{y}} = \kappa(t)\big(\mathbf{y} \odot \mathbf{y} - s_2\, \mathbf{y}\big),$$
$$\dot{p} = -\frac{J}{\sigma(p)}\, [p(1-p)]^2 \big(s_2 + t_2\big),$$
$$s_2 = \|\mathbf{y}\|_2^2, \qquad t_2 = \|\mathbf{z}\|_2^2.$$

In internal time,

$$\frac{dp}{d\tau} = -p(1 - p)\big(s_2(\tau) + t_2(\tau)\big).$$

Along the generic $J > 0$ branch we have $s_2(\tau) \to 1$ (good-block polarization) and $t_2(\tau) \to 1/M$ (bad-block mixing), hence

$$\frac{dp}{d\tau} = -\big(1 + \tfrac{1}{M}\big)\, p\,\big(1 + o(1)\big),$$
$$p(\tau) = C\, e^{-(1+1/M)\tau}\,\big(1 + o(1)\big), \qquad \tau \to \infty.$$

for some $C > 0$ determined by the initial condition (e.g. via the logit identity).

Next, since

$$\frac{dt}{d\tau} = \frac{1}{\kappa(t)} = \frac{\sigma\big(p(\tau)\big)}{J\, p(\tau)\,(1 - p(\tau))} = \frac{\sigma\big(p(\tau)\big)}{J\, p(\tau)}\,(1 + o(1)),$$

the physical-time asymptotics are governed by the local behavior of $\sigma(p)$ near $p = 0$.

**Theorem J.19** (Physical-time rates via the local law of $\sigma(p)$ (multi-bad setting)). *Assume $\sigma(p) \sim \sigma_0\, p^\gamma$ as $p \downarrow 0$ with $\sigma_0 > 0$ and $\gamma \in \mathbb{R}$. Let $m = \arg\max_i y_i(0)$ (unique) and suppose we are on the $J > 0$ branch so that $p(t) \downarrow 0$. Write $a := 1 + \frac{1}{M}$ (the asymptotic internal-time slope of $\operatorname{logit} p$). Then, generically, as $t \to \infty$ or to a finite absorption time $t_\infty$,*

*(A)* $\gamma < 1$ : $p(t) \asymp t^{-1/(1-\gamma)}$,
$$1 - y_m(t),\ y_i(t)\ (i \neq m) \asymp t^{-1/[a(1-\gamma)]};$$

*(B)* $\gamma = 1$ : $p(t) \asymp e^{-(aJ/\sigma_0)t}$,
$$1 - y_m(t),\ y_i(t)\ (i \neq m) \asymp e^{-(J/\sigma_0)t};$$

*(C)* $\gamma > 1$ : $p(t) \asymp (t_\infty - t)^{1/(\gamma-1)}$,
$$1 - y_m(t),\ y_i(t)\ (i \neq m) \asymp (t_\infty - t)^{\frac{1}{[a(\gamma-1)]}}.$$

*All implicit constants depend only on $(\mathbf{y}(0), \mathbf{z}(0), p(0), J, \sigma_0, \gamma)$. The power-law exponents for $p(t)$ are universal (independent of $K$ and $M$), while the $\mathbf{y}$-rates depend on $M$ through $a = 1 + 1/M$ (reducing to the one-bad-arm case when $M = 1$).*

**Corollary J.20** (Who wins, and how fast?). *For a general initial condition $\mathbf{y}(0) \in \Delta^{K-1}$ with a unique maximizer $m$, the winning arm is $m$. In internal time, the non-winners decay as $e^{-\tau}$; in physical time, the rates follow Theorem J.19.*

**Corollary J.21** (Sharp $\tau$-envelopes for $p$). *Using the logit $L(\tau) = \log \frac{p(\tau)}{1-p(\tau)}$,*

$$\frac{dL}{d\tau} = -(s_2(\tau) + t_2(\tau)).$$

*On the generic $J > 0$ branch, $s_2(\tau) \uparrow 1$ and $t_2(\tau) \downarrow 1/M$ (with exponentially decaying gaps), so*

$$L(\tau) = L(0) - \Big(1 + \frac{1}{M}\Big)\tau + O(1), \quad p(\tau) = \Theta\big(e^{-(1+1/M)\tau}\big).$$

*Thus physical-time rates reduce to integrating $dt/d\tau \sim \sigma(p(\tau))/(Jp(\tau))$, i.e. to the local exponent $\gamma$.*

### J.5. Dynamics of $\mathbf{y}$

Let $q := \mathbf{y}(0) \in \Delta^{K-1}$ be the initial composition. Reparametrize time by

$$\tau(t) = \int_0^t \kappa(s)\, ds, \qquad \kappa(t) = \frac{J}{\sigma(p(t))}\, p(t)\big(1 - p(t)\big).$$

In $\tau$-time the inner flow is autonomous:

$$\frac{dy_j}{d\tau} = y_j\big(y_j - s_2\big), \qquad s_2 = \sum_i y_i^2.$$

Introduce $u_j := 1/y_j$. Then $u_j' - s_2\, u_j = -1$, whose solution is[3]

$$u_j(\tau) = e^{S(\tau)}\left(\frac{1}{q_j} - I(\tau)\right),$$

$$S(\tau) := \int_0^\tau s_2(r)\,dr, \qquad I(\tau) := \int_0^\tau e^{-S(s)}\,ds.$$

Notice that, based on the simplex constraint $\sum_{j=1}^K y_j(\tau) = 1$, we can eliminate the common factor $e^{-S(\tau)}$:

$$1 = \sum_{\ell=1}^K y_\ell(\tau) = e^{-S(\tau)} \sum_{\ell=1}^K \frac{1}{\frac{1}{q_\ell} - I(\tau)}. \qquad (86)$$

Hence

$$e^{-S(\tau)} = \left[\sum_{\ell=1}^K \frac{1}{\frac{1}{q_\ell} - I(\tau)}\right]^{-1}. \qquad (87)$$

Substituting this back into the expression for $y_j(\tau)$ gives

$$y_j(\tau) = \frac{e^{-S(\tau)}}{\frac{1}{q_j} - I(\tau)} = \frac{\dfrac{q_j}{1 - I(\tau)\, q_j}}{\sum_{\ell=1}^K \dfrac{q_\ell}{1 - I(\tau)\, q_\ell}}. \qquad (88)$$

---

[3] Writing it in the standard form $u_j'(\tau) + a(\tau)\, u_j(\tau) = b(\tau)$, we have $a(\tau) = -s_2(\tau)$, $b(\tau) = -1$. For a linear ODE $u_j' + a(\tau)u_j = b(\tau)$, the integrating factor is $\mu(\tau) = \exp\!\left(\int_0^\tau a(r)\,dr\right)$. Using $a(\tau) = -s_2(\tau)$ and the notation $S(\tau) := \int_0^\tau s_2(r)\,dr$, we obtain $\mu(\tau) = \exp\!\left(\int_0^\tau -s_2(r)\,dr\right) = e^{-S(\tau)}$. Multiplying the ODE by $\mu(\tau)$ gives

$$e^{-S(\tau)}u_j'(\tau) - s_2(\tau)e^{-S(\tau)}u_j(\tau) = -e^{-S(\tau)}.$$

By the product rule and the definition of $S$,

$$\frac{d}{d\tau}\left(e^{-S(\tau)}u_j(\tau)\right) = e^{-S(\tau)}u_j'(\tau) - s_2(\tau)e^{-S(\tau)}u_j(\tau),$$

so the left-hand side becomes an exact derivative and the equation reduces to

$$\frac{d}{d\tau}\left(e^{-S(\tau)}u_j(\tau)\right) = -e^{-S(\tau)}.$$

Integrating from 0 to $\tau$ yields

$$e^{-S(\tau)}u_j(\tau) - e^{-S(0)}u_j(0) = -\int_0^\tau e^{-S(s)}\,ds.$$

Since $S(0) = 0$, we have $e^{-S(0)} = 1$. Writing the initial composition as $y_j(0) = q_j$, we have $u_j(0) = 1/q_j$. Thus

$$e^{-S(\tau)}u_j(\tau) = \frac{1}{q_j} - \int_0^\tau e^{-S(s)}\,ds.$$

Defining $I(\tau) := \int_0^\tau e^{-S(s)}\,ds$, and multiplying both sides by $e^{S(\tau)}$, we obtain

$$u_j(\tau) = e^{S(\tau)}\left(\frac{1}{q_j} - I(\tau)\right).$$

## J.6. Evolution of the collision term $s_2$

Throughout, let $q := \mathbf{y}(0) \in \Delta^{K-1}$ be the initial within–good composition and $I(\tau)$ the scalar from Eq. (88). For $r \in \{1, 2, 3\}$ define the moment sums

$$\mathsf{M}_r(I) := \sum_{j=1}^K \frac{q_j^r}{(1 - Iq_j)^r},$$

$$\mathsf{M}_1(I) > 0 \quad \text{for } I \in [0, 1/q_*), \qquad q_* := \max_j q_j.$$

**Lemma J.22** (Exact moment formulas and internal-time map). *Under the change of variable $I = I(\tau)$ from Eq. (88),*

$$y_j(\tau(I)) = \frac{\dfrac{q_j}{1 - Iq_j}}{\mathsf{M}_1(I)},$$

$$s_2(\tau(I)) = \frac{\mathsf{M}_2(I)}{\mathsf{M}_1(I)^2}, \qquad s_3(\tau(I)) = \frac{\mathsf{M}_3(I)}{\mathsf{M}_1(I)^3},$$

$$\tau(I) = \int_0^I \mathsf{M}_1(z)\,dz$$
$$= -\sum_{j=1}^K \log(1 - Iq_j),$$

$$S(\tau) := \int_0^\tau s_2(u)\,du$$
$$= \int_0^{I(\tau)} \frac{\mathsf{M}_2(z)}{\mathsf{M}_1(z)}\,dz.$$

*Proof.* Eq. (88) gives the first line immediately. The formulas for $s_2$ and $s_3$ follow by summing $y_j^2$ and $y_j^3$. Since $I'(\tau) = e^{-S(\tau)} = 1/\mathsf{M}_1(I)$, we get $d\tau/dI = \mathsf{M}_1(I)$, and integrate to obtain $\tau(I)$. Finally, $S(\tau) = \int s_2\,d\tau = \int (\mathsf{M}_2/\mathsf{M}_1)\,dI$.

**Proposition J.23** (Two-sided integral bounds; refined logit envelope (multi-bad outer bounds)). *For all $I \in [0, 1/q_*)$,*

$$\frac{1}{K}\,\tau(I) \le S(\tau(I)) \le -\log(1 - Iq_*),$$

$$\tau(I) = -\sum_{j=1}^K \log(1 - Iq_j).$$

*Define also the bad-block collision integral*

$$T(\tau) := \int_0^\tau t_2(u)\,du \text{ with } t_2(u) = \|\mathbf{z}(u)\|_2^2 \in \left[\frac{1}{M}, 1\right],$$

*so that $\frac{\tau}{M} \le T(\tau) \le \tau$. Then for the logit $L(\tau) = \log\frac{p(\tau)}{1-p(\tau)}$,*

$$L(\tau) = L(0) - S(\tau) - T(\tau). \qquad (89)$$

*Consequently,*

$$L(0) - 2\tau \leq L(\tau) \leq L(0) - \left(\tfrac{1}{K} + \tfrac{1}{M}\right)\tau, \qquad (90)$$

*and along the implicit time map $I \mapsto \tau(I)$ we also have*

$$L\big(\tau(I)\big) \geq L(0) - \tau(I) + \log\big(1 - Iq_*\big). \qquad (91)$$

*Proof.* The bounds on $S$ are as in the one-bad case: Cauchy–Schwarz on $\{a_j\} = \{\tfrac{q_j}{1-Iq_j}\}$ gives $\mathsf{M}_1(I)^2 \leq K\,\mathsf{M}_2(I)$; hence $\mathsf{M}_2/\mathsf{M}_1 \geq \mathsf{M}_1/K$. Integrating: $S = \int(\mathsf{M}_2/\mathsf{M}_1)\,dI \geq \tfrac{1}{K}\int \mathsf{M}_1\,dI = \tau/K$. For the upper bound, monotonicity of $t \mapsto \tfrac{t}{1-It}$ implies $\tfrac{q_j^2}{(1-Iq_j)^2} \leq \tfrac{q_*}{1-Iq_*} \cdot \tfrac{q_j}{1-Iq_j}$, so $\mathsf{M}_2 \leq \tfrac{q_*}{1-Iq_*}\mathsf{M}_1$. Integrating yields $S \leq \int \tfrac{q_*}{1-Iq_*}\,dI = -\log(1 - Iq_*)$.

For the logit, use Eq. (89) and the bounds $S(\tau) \in [\tau/K, \tau]$ and $T(\tau) \in [\tau/M, \tau]$ to get (90). For (91), combine $T(\tau) \leq \tau$ with $S(\tau(I)) \leq -\log(1 - Iq_*)$.

*Remark* J.24 (What is "collision"?). Here $s_2 = \sum_i y_i^2$ is the usual collision probability on the simplex. The *collision gap* $s_3 - s_2^2 = \sum_i y_i(y_i - s_2)^2$ drives the slope: in internal time, $\tfrac{ds_2}{d\tau} = 2\big(s_3 - s_2^2\big) \geq 0$, with strict increase off the uniform-on-support sets.

**Proposition J.25** (Pointwise bounds for $s_2'$ (tighter than logistic)). *Let $u(\tau) := \sqrt{s_2(\tau)} \in [1/\sqrt{K}, 1]$ and $y_{\max}(\tau) := \max_i y_i(\tau)$. Then for all $\tau$,*

$$\frac{ds_2}{d\tau} = 2\sum_{i=1}^{K} y_i\,(y_i - s_2)^2 = 2\,(s_3 - s_2^2). \qquad (92)$$

*Moreover,*

$$\frac{ds_2}{d\tau} = 2(s_3 - s_2^2) \leq 2\big(s_2^{3/2} - s_2^2\big) = 2\,s_2^{3/2}\big(1 - \sqrt{s_2}\big), \qquad (93)$$

*using $s_3 = \|\mathbf{y}\|_3^3 \leq \|\mathbf{y}\|_2^3 = s_2^{3/2}$.*

$$\frac{ds_2}{d\tau} \leq 2\,y_{\max}(\tau)\,s_2(\tau)\,\big(K\,s_2(\tau) - 1\big),$$
$$\text{since } \sum_i (y_i - s_2)^2 = s_2\,\big(Ks_2 - 1\big). \qquad (94)$$

*Moreover, writing $\mathbf{y} = \mathrm{unif}_K + v$ with $\sum_i v_i = 0$ and $\eta := \|v\|_2^2 = s_2 - \tfrac{1}{K}$, we have*

$$\frac{ds_2}{d\tau} = 2\Big(\frac{\eta}{K} - \eta^2 + \sum_i v_i^3\Big),$$

$$\Rightarrow \quad 2\Big(\frac{\eta}{K} - \eta^2 - \eta^{3/2}\Big) \leq \frac{ds_2}{d\tau} \leq 2\Big(\frac{\eta}{K} - \eta^2 + \eta^{3/2}\Big), \qquad (95)$$

*since $\big|\sum_i v_i^3\big| \leq \|v\|_3^3 \leq \|v\|_2^3 = \eta^{3/2}$.*

*Proof.* Eq. (92) is the variance identity. For (93), use $\|\mathbf{y}\|_3 \leq \|\mathbf{y}\|_2$ to get $s_3 \leq s_2^{3/2}$. For (94), bound $y_i \leq y_{\max}$ in (92) and note

$$\sum_i (y_i - s_2)^2 = \sum_i y_i^2 - 2s_2 \sum_i y_i + Ks_2^2 = s_2(Ks_2 - 1).$$

For (95), expand $s_3$ around unif: $s_3 = \sum_i (1/K + v_i)^3 = 1/K^2 + 3\eta/K + \sum_i v_i^3$ and subtract $s_2^2 = (1/K + \eta)^2$.

**Corollary J.26** (Implicit envelope for $u(\tau) = \sqrt{s_2(\tau)}$). *Integrating the differential inequality $u'(\tau) \leq u(\tau)^2\big(1 - u(\tau)\big)$ yields the implicit bound*

$$\log\frac{u(\tau)}{1 - u(\tau)} - \frac{1}{u(\tau)} \leq \log\frac{u_0}{1 - u_0} - \frac{1}{u_0} + \tau,$$
$$u_0 = \sqrt{s_2(0)} = \|q\|_2.$$

*This dominates the logistic envelope $s_2(\tau) \leq \big(1 + [(1 - s_2(0))/s_2(0)]e^{-2\tau}\big)^{-1}$ whenever $u_0$ is close to $1/\sqrt{K}$ (near-uniform start).*

**Corollary J.27** (Refined envelopes for $p(\tau)$ (multi-bad outer bounds)). *Using $L(\tau) = L(0) - \int_0^\tau (s_2 + t_2)\,du$ and Proposition J.23:*

$$\frac{1}{1 + \frac{1-p_0}{p_0}\,e^{2\tau}} \leq p(\tau) \leq \frac{1}{1 + \frac{1-p_0}{p_0}\,e^{(\frac{1}{K}+\frac{1}{M})\tau}},$$

*and, with the implicit time $I \mapsto \tau(I)$,*

$$p\big(\tau(I)\big) \geq \frac{1}{1 + \frac{1-p_0}{p_0}\,\exp\Big(\tau(I) - \log(1 - Iq_*)\Big)}.$$

*The last (implicit) lower envelope becomes tight as the good mass polarizes to the maximizer of $q$.*

## K. Inner Dynamics of the Bad Arms

This section is the companion to Section J. When we keep the $M$ bad arms explicitly (instead of aggregating them into a single virtual bad arm), the *within-bad* composition obeys the *same* collision ODE as the within-good composition, but with an overall *sign flip*. As a result, essentially every statement in Section J has a direct bad-block analogue obtained by the substitutions

$$(\mathbf{y}, K) \mapsto (\mathbf{z}, M),$$
$$\kappa \mapsto -\kappa, \qquad \text{(equivalently, } \tau \mapsto -\tau \text{ in internal time).}$$

We record the corresponding results (without reproving them).

### K.1. Within-bad composition and pushforward

Let the policy over $K$ good arms and $M$ bad arms be

$$\mathbf{p} = (p_1, \ldots, p_K, \, p_{b_1}, \ldots, p_{b_M}) \in \Delta^{K+M-1},$$

$$p := \sum_{m=1}^{M} p_{b_m} \in [0, 1].$$

Define the within-bad normalized composition

$$z_m := \frac{p_{b_m}}{p}, \qquad m \in [M], \qquad \mathbf{z} \in \Delta^{M-1}.$$

Writing $p_i = \exp(\theta_i)/Z$ with

$$Z = \sum_{j=1}^{K} e^{\theta_j} + \sum_{m=1}^{M} e^{\theta_{b_m}},$$

the bad normalization cancels the good logits:

$$\begin{aligned} z_m &= \frac{\exp(\theta_{b_m})}{\sum_{\ell=1}^{M} \exp(\theta_{b_\ell})} \\ &= \big(\mathrm{softmax}(\boldsymbol{\theta}_{\mathrm{bad}})\big)_m, \end{aligned} \tag{96}$$

so $\mathbf{z}$ depends only on $\boldsymbol{\theta}_{\mathrm{bad}}$.

**Lemma K.1** (Pushforward from logits to within-bad composition). *For any small increment $\Delta\boldsymbol{\theta} = (\Delta\boldsymbol{\theta}_{\mathrm{good}}, \Delta\boldsymbol{\theta}_{\mathrm{bad}})$,*

$$\begin{aligned} \Delta\mathbf{z} &= \big(\mathrm{Diag}(\mathbf{z}) - \mathbf{z}\mathbf{z}^\top\big) \Delta\boldsymbol{\theta}_{\mathrm{bad}} \\ &= \mathbf{z} \odot \big(\Delta\boldsymbol{\theta}_{\mathrm{bad}} - \langle \mathbf{z}, \Delta\boldsymbol{\theta}_{\mathrm{bad}}\rangle \mathbf{1}\big), \end{aligned} \tag{97}$$

*and in particular $\partial\mathbf{z}/\partial\boldsymbol{\theta}_{\mathrm{good}} = \mathbf{0}$.*

*Proof.* Identical to Lemma J.1 with $(\mathbf{y}, \boldsymbol{\theta}_{\mathrm{good}}, K)$ replaced by $(\mathbf{z}, \boldsymbol{\theta}_{\mathrm{bad}}, M)$.

### K.2. Bad-block drift: the same collision field with opposite sign

Assume the same block symmetry as in Section J / Appendix D:

$$\begin{aligned} A_j &= a_{\mathrm{g}}(p), &j &\leq K, \\ A_{b_m} &= a_{\mathrm{b}}(p), &m &\leq M, \\ \Delta r(p) &:= a_{\mathrm{b}}(p) - a_{\mathrm{g}}(p). \end{aligned}$$

Recall the (good-block) scalar from (78):

$$\kappa(p) := -\eta \, p(1-p) \, \Delta r(p).$$

Then the expected logit drift in the bad block is the sign-reversed analogue of (78):

$$\begin{aligned} \mathbb{E}[\Delta\boldsymbol{\theta}_{\mathrm{bad}}] &= -\kappa(p) \, \mathbf{z}, \\ \text{i.e.} \qquad \mathbb{E}[\Delta\theta_{b_m}] &= -\kappa(p) \, z_m. \end{aligned} \tag{98}$$

**Proposition K.2** (Within-bad mean drift in $\mathbf{z}$-coordinates). *Applying Lemma K.1 to (98) yields*

$$\begin{aligned} \mathbb{E}[\Delta\mathbf{z}] &= -\kappa(p)\Big(\mathbf{z}\odot\mathbf{z} - \|\mathbf{z}\|_2^2\,\mathbf{z}\Big), \\ \mathbb{E}[\Delta z_m] &= -\kappa(p)\, z_m\big(z_m - \|\mathbf{z}\|_2^2\big). \end{aligned} \tag{99}$$

*In the noisy GRPO specialization $a_{\mathrm{g}}(p) = \frac{Jp}{\sigma(p)}$, $a_{\mathrm{b}}(p) = -\frac{J(1-p)}{\sigma(p)}$ (so $\Delta r(p) = -J/\sigma(p)$ and $\kappa(p) = \eta\frac{J}{\sigma(p)}p(1-p)$), this becomes*

$$\mathbb{E}[\Delta z_m] = -\eta\,\frac{J}{\sigma(p)}\,p(1-p)\,z_m\Big(z_m - \|\mathbf{z}\|_2^2\Big). \tag{100}$$

*Consequences (bad-block smoothing vs. polarization).* For $J > 0$ (hence $\kappa(p) > 0$), the sign in (99) implies a *smoothing* effect: components with $z_m > \|\mathbf{z}\|_2^2$ shrink while those with $z_m < \|\mathbf{z}\|_2^2$ grow, pushing $\mathbf{z}$ toward uniformity on its support. For $J < 0$ the direction reverses and the bad block *polarizes* (winner-take-all among bad modes), mirroring the good-block behavior when $J > 0$.

### K.3. Internal-time form and direct correspondence with Section J

As in Section J, define the internal time

$$\tau(t) := \int_0^t \kappa\big(p(s)\big)\,ds,$$

so that (in mean-field ODE form) the bad composition satisfies

$$\frac{d\mathbf{z}}{d\tau} = -\Big(\mathbf{z}\odot\mathbf{z} - \|\mathbf{z}\|_2^2\,\mathbf{z}\Big). \tag{101}$$

Equivalently, with $\rho := -\tau$ one has

$$\frac{d\mathbf{z}}{d\rho} = \mathbf{z}\odot\mathbf{z} - \|\mathbf{z}\|_2^2\,\mathbf{z},$$

which is *exactly* the same autonomous ODE as (81) for $\mathbf{y}$, with $K$ replaced by $M$.

**Lemma K.3** (Geometry / Lyapunov structure for the bad block (sign-reversed)). *Let $\mathbf{z}(\tau) \in \Delta^{M-1}$ solve (101) and define*

$$t_2(\tau) := \|\mathbf{z}(\tau)\|_2^2 = \sum_{m=1}^{M} z_m(\tau)^2, \qquad t_3(\tau) := \sum_{m=1}^{M} z_m(\tau)^3.$$

*Then the statements of the "Geometry and Lyapunov structure" lemma in Section J carry over with $(\mathbf{y}, s_2, s_3, K)$ replaced by $(\mathbf{z}, t_2, t_3, M)$ and with all monotonicities reversed. Concretely:*

(1) **Simplex invariance.** $\sum_m z_m(\tau) = 1$ and $z_m(\tau) \geq 0$ *are preserved.*

(2) **Gradient form with opposite sign.** *With the same potential*

$$\mathcal{L}(\mathbf{z}) := \frac{1}{3}\sum_{m=1}^{M} z_m^3 - \frac{1}{4}\Big(\sum_{m=1}^{M} z_m^2\Big)^2,$$

$$\nabla\mathcal{L}(\mathbf{z}) = \big(z_m^2 - t_2\, z_m\big)_m,$$

*we have*

$$\frac{d\mathbf{z}}{d\tau} = -\nabla\mathcal{L}(\mathbf{z}), \qquad \frac{d}{d\tau}\mathcal{L}\big(\mathbf{z}(\tau)\big) = -\|\nabla\mathcal{L}(\mathbf{z})\|_2^2 \le 0.$$

(3) **Monotone** de-**concentration.**

$$\frac{d}{d\tau}t_2(\tau) = -2\big(t_3(\tau) - t_2(\tau)^2\big) \le 0,$$

*with equality iff* $\mathbf{z}$ *is uniform on its support.*

(4) **Equilibria.** *Stationary points are exactly the uniform points on a support of size* $m$*: for any* $m \in \{1,\dots,M\}$*, any point with exactly* $m$ *nonzero entries, each equal to* $1/m$*, is an equilibrium.*

### K.4. Stability and global limits (bad-block counterpart of Section J.3)

Write the within-bad mean-field ODE in physical time as

$$\dot{\mathbf{z}} = -\kappa\big(p(t)\big)\,\big(\mathbf{z}\odot\mathbf{z} - t_2\,\mathbf{z}\big),$$
$$t_2 = \|\mathbf{z}\|_2^2, \qquad\qquad\qquad (102)$$
$$\kappa(p) = \frac{J}{\sigma(p)}\,p(1-p) \qquad \text{(noisy GRPO).}$$

**Proposition K.4** (Stability of the uniform and vertex equilibria for the bad block)**.** *The stability conclusions of Propositions J.15 and J.16 carry over with* $K \mapsto M$ *and* $\kappa \mapsto -\kappa$*:*

- **Uniform equilibrium.** *For* $\mathbf{z}^\star = \frac{1}{M}\mathbf{1}$*, the nontrivial eigenvalues on the simplex tangent space are* $\lambda = -\kappa(p)/M$*. Hence*

$$J > 0 \;\big(\kappa(p) > 0\big): \quad \mathbf{z}^\star \text{ is asymptotically stable,}$$
$$\text{(bad mass spreads);}$$

$$J < 0 \;\big(\kappa(p) < 0\big): \quad \mathbf{z}^\star \text{ is unstable.}$$

- **Vertex equilibria.** *For a vertex* $\mathbf{z}^\star = e_m$*, the transverse modes have eigenvalues* $+\kappa(p)$*, hence*

$$J > 0 \;\big(\kappa(p) > 0\big): \quad \mathbf{z}^\star = e_m \text{ is unstable,}$$
$$J < 0 \;\big(\kappa(p) < 0\big): \quad \mathbf{z}^\star = e_m \text{ is locally asymptotically stable,}$$
$$\text{(bad-mode collapse).}$$

**Corollary K.5** (Stability summary for within-bad equilibria)**.** *The within-bad stability types are the sign-reversed analogue of Corollary J.17:*

| Equilibrium | $J$ | Stability |
|---|---|---|
| Uniform $z_m = 1/M$ | $> 0$ | Stable (bad mass diffuses) |
| Uniform $z_m = 1/M$ | $< 0$ | Unstable |
| Vertex $\mathbf{z} = e_m$ | $> 0$ | Unstable |
| Vertex $\mathbf{z} = e_m$ | $< 0$ | Stable (bad-mode collapse) |

*Table 3.* Stability of canonical equilibria for the within-bad ODE (102).

**Theorem K.6** (Global limit of $\mathbf{z}$ (bad-block counterpart of Theorem J.11))**.** *Consider* (102) *with an interior initialization* $\mathbf{z}(0)$ *(all coordinates positive). Then:*

- *If* $J > 0$*, the flow is the* reverse *of the good-block collision flow in internal time. Consequently* $\mathbf{z}(t)$ *converges to the uniform point on the full bad simplex:*

$$\mathbf{z}(t) \;\longrightarrow\; \frac{1}{M}\mathbf{1},$$

*and the collision probability* $t_2(t) = \|\mathbf{z}(t)\|_2^2$ *decreases monotonically to* $1/M$*.*

- *If* $J < 0$*, the direction reverses and* $\mathbf{z}(t)$ *follows the* forward *collision flow (on* $\Delta^{M-1}$*). For generic initial conditions (no ties),* $\mathbf{z}(t)$ *converges to the vertex selected by the unique maximizer* $m = \arg\max_m z_m(0)$ *(winner-take-all among bad modes).*

- *If some coordinates of* $\mathbf{z}(0)$ *are exactly zero, the support is invariant and the same statements hold with* $M$ *replaced by the support size (uniform-on-support for* $J > 0$*, vertex-on-support for* $J < 0$*).*

*Proof.* Immediate from Theorem J.11 by the correspondence (101) (time reversal) and $K \mapsto M$. 

**Proposition K.7** (Exponential approach in internal time)**.** *The rate statements in Proposition J.12 transfer to* $\mathbf{z}$ *with the same substitutions:*

- *For* $J > 0$ *(uniform stable), linearization at* $\mathbf{z}^\star = \frac{1}{M}\mathbf{1}$ *yields*

$$\big\|\mathbf{z}(\tau) - \tfrac{1}{M}\mathbf{1}\big\|_2 = \Theta\big(e^{-\tau/M}\big) \quad (\tau \to +\infty),$$

*and hence* $t_2(\tau) - \frac{1}{M} = \Theta(e^{-2\tau/M})$*.*

- *For* $J < 0$ *(vertices stable), after reversing internal time as in the sign discussion of Section J, the non-winners decay as* $e^{-|\tau|}$ *toward the winning vertex, exactly as in Proposition J.12 with* $K \mapsto M$*.*

*Remark* K.8 (Scalar closed-form representation (sign-flipped analogue of (88)))**.** The explicit "one-scalar" representation for $\mathbf{y}(\tau)$ in Section J also carries over

to the bad block with the sign flipped in the denominators. If $q := \mathbf{z}(0) \in \Delta^{M-1}$ and $I(\tau)$ is a strictly increasing scalar with $I(0) = 0$, then

$$z_m(\tau) = \frac{\dfrac{q_m}{1 + I(\tau)\, q_m}}{\displaystyle\sum_{\ell=1}^{M} \frac{q_\ell}{1 + I(\tau)\, q_\ell}}, \tag{103}$$

$$I(\tau) \uparrow \infty \;\Rightarrow\; \mathbf{z}(\tau) \to \tfrac{1}{M}\mathbf{1}.$$

This is the direct sign-reversal analogue of (88); we omit the derivation.

*Remark* K.9 (Effect on the total bad-mass drift in the multi-bad model). In the multi-bad setting the total bad mass $p(t)$ couples to *both* collision terms: in internal time $\tau$,

$$\begin{aligned}
\frac{dp}{d\tau} &= -\, p(1 - p)\big(\|\mathbf{y}(\tau)\|_2^2 + \|\mathbf{z}(\tau)\|_2^2\big), \\
\frac{d}{d\tau} \log \frac{p}{1 - p} &= -\big(\|\mathbf{y}\|_2^2 + \|\mathbf{z}\|_2^2\big).
\end{aligned} \tag{104}$$

Thus $\|\mathbf{z}\|_2^2 \in [1/M, 1]$ enters only as a bounded multiplicative factor in the decay of $\operatorname{logit} p$, recovering the aggregated-bad model when $M = 1$ (where $\|\mathbf{z}\|_2^2 \equiv 1$).

## L. Shahshahani geometry and the within–good flow

Let $\mathbf{y} = (y_1, \dots, y_K) \in \Delta^{K-1}$ denote the within–good composition and define

$$\dot{\mathbf{y}} = \kappa(p)\Big(\mathbf{y}\odot\mathbf{y} - \|\mathbf{y}\|_2^2\, \mathbf{y}\Big), \qquad \kappa(p) := \frac{J}{\sigma(p)}\, p(1-p), \tag{105}$$

where $J$ is the judge–separation, $\sigma(p) > 0$ is the group–normalization scale, and $p \in [0, 1]$ is the bad–mass. Set $s_2 := \|\mathbf{y}\|_2^2 = \sum_i y_i^2$ and $s_3 := \sum_i y_i^3$.

**Shahshahani (Fisher) metric on the simplex.** On the interior of the simplex $\Delta^{K-1}$, the Shahshahani inner product is

$$\langle \mathbf{u}, \mathbf{v}\rangle_{\mathbf{y}} = \sum_{i=1}^{K} \frac{u_i\, v_i}{y_i},$$

$$T_{\mathbf{y}}\Delta^{K-1} = \Big\{\mathbf{v} \in \mathbb{R}^K : \sum_i v_i = 0\Big\}.$$

For a smooth potential $\phi : \Delta^{K-1} \to \mathbb{R}$, the associated *natural gradient* (steepest ascent in this metric) has the replicator form

$$\operatorname{grad}_{\mathrm{Shah}} \phi(y) = \mathbf{y} \odot \Big(\nabla\phi(\mathbf{y}) - \langle\nabla\phi(\mathbf{y}), \mathbf{y}\rangle\, \mathbf{1}\Big). \tag{106}$$

Indeed, with $w := \mathbf{y} \odot (\nabla\phi - c\,\mathbf{1})$ and $c = \langle\nabla\phi, y\rangle$, we have $\sum_i w_i = 0$ and, for any $v \in T_{\mathbf{y}}\Delta^{K-1}$, $\langle w, v\rangle_{\mathbf{y}} = \sum_i (\nabla\phi_i - c)v_i = \langle\nabla\phi, v\rangle$, which characterizes the Riemannian gradient.

**Proposition L.1** (Shahshahani gradient representation). *Let* $\Phi(\mathbf{y}) := \tfrac{1}{2}\|\mathbf{y}\|_2^2 = \tfrac{1}{2}\sum_i y_i^2$. *Then* (105) *is the Shahshahani gradient flow of* $\Phi$, *scaled by* $\kappa(p)$:

$$\dot{\mathbf{y}} = \kappa(p)\, \operatorname{grad}_{\mathrm{Shah}} \Phi(\mathbf{y}).$$

*Equivalently, in coordinates,* $\dot{y}_i = \kappa(p)\, y_i\big(y_i - \|\mathbf{y}\|_2^2\big).$

*Proof.* We have $\nabla\Phi(\mathbf{y}) = \mathbf{y}$ and $\langle\nabla\Phi(\mathbf{y}), \mathbf{y}\rangle = \|\mathbf{y}\|_2^2$. Applying (106) gives $\operatorname{grad}_{\mathrm{Shah}} \Phi(\mathbf{y}) = \mathbf{y} \odot (\mathbf{y} - \|\mathbf{y}\|_2^2\mathbf{1})$, hence the claim. $\square$

**Interpretation (Herfindahl ascent in Fisher units).** The potential $\Phi(\mathbf{y}) = \tfrac{1}{2}\|\mathbf{y}\|_2^2$ is the (half) Herfindahl–Hirschman concentration index. Thus (105) is the *steepest way, in Fisher/Shahshahani geometry*, to increase concentration when $\kappa(p) > 0$ (and to decrease it when $\kappa(p) < 0$). The scalar $\kappa(p) = \frac{J}{\sigma(p)}p(1 - p)$ gates the time–scale: the inner reshuffling freezes at $p \in \{0, 1\}$ and is fastest near $p = \tfrac{1}{2}$; its sign flips with $J$.

**Corollary L.2** (Lyapunov monotonicity and a variance identity). *Along any trajectory of* (105),

$$\frac{d}{dt}\Phi\big(\mathbf{y}(t)\big) = \kappa\big(p(t)\big)\big(s_3 - s_2^2\big)$$
$$= \kappa\big(p(t)\big)\operatorname{Var}_{i\sim\mathbf{y}}(y_i) \geq 0, \qquad \text{whenever } \kappa \geq 0.$$
$$(107)$$

*with equality iff y is uniform on its support* ($y_i \in \{0, 1/m\}$ *on some subset of size m*). *Equivalently,* $\frac{d}{dt}\|\mathbf{y}\|_2^2 = 2\,\kappa(p)\operatorname{Var}_{i\sim\mathbf{y}}(y_i)$.

*Proof.* By the chain rule, $\frac{d}{dt}\Phi = \langle\nabla\Phi, \dot{\mathbf{y}}\rangle = \kappa\sum_i y_i\big(y_i - \|\mathbf{y}\|_2^2\big)y_i = \kappa\,(s_3 - s_2^2)$. Since $s_3 - s_2^2 = \sum_i y_i(y_i - s_2)^2 \geq 0$ and vanishes exactly at support–uniform points, the claim follows. □

**Proposition L.3** (Equilibria, support invariance, and stability). *The rest points of* (105) *are exactly the barycenters of faces: for any subset* $S \subseteq [K]$ *of size m,* $y_i^\star = \frac{1}{m}$ *for* $i \in S$ *and* $y_i^\star = 0$ *otherwise. Moreover:*

(i) *Support invariance. If* $y_i(0) = 0$, *then* $y_i(t) \equiv 0$ *for all t (since* $\dot{y}_i = \kappa\,y_i(\cdot)$).

(ii) *Stability for* $\kappa > 0$. *The unique asymptotically stable equilibria are the vertices (m = 1); all higher–dimensional barycenters (m* $\geq$ *2) are saddles/unstable.*

(iii) *Stability for* $\kappa < 0$. *The roles reverse: the full–uniform point (m = K) is the unique asymptotically stable equilibrium; all others are unstable.*

*Proof sketch.* At equilibrium, $0 = \dot{y}_i = \kappa\,y_i(y_i - s_2)$ implies: either $y_i = 0$ or $y_i = s_2$. If $m$ coordinates are positive, then $1 = \sum_i y_i = m\,s_2 \Rightarrow y_i = s_2 = \frac{1}{m}$ on the support. For stability, note that (105) is a Shahshahani gradient system for the convex $\Phi$, scaled by $\kappa$. When $\kappa > 0$ the flow ascends $\Phi$ and converges to its maximizers on $\Delta^{K-1}$, which are precisely the vertices; when $\kappa < 0$ it descends to the unique minimizer, the full–uniform point. Support invariance follows from the factor $y_i$ in each coordinate. □

## M. Hyperparameters and Training Details

Table 4 summarizes the training and evaluation configuration used in all experiments. We fine-tune a Qwen2.5-3B base model for 1410 optimization steps (2 epochs) with a global batch size of 16. For rollout generation, we sample 8 responses per prompt with temperature 1.0 (top-$p = 1.0$, top-$k = -1$) and truncate prompts/responses at a maximum length of 4000 tokens each. For actor optimization, we use GRPO with symmetric PPO clipping $(\varepsilon_{\text{low}}, \varepsilon_{\text{high}}) = (0.2, 0.2)$ and Adam (learning rate $10^{-6}$,

weight decay 0.1, $(\beta_1, \beta_2) = (0.9, 0.999)$), with gradient-norm clipping 1.0 and a constant learning-rate schedule with 10 warmup steps. Evaluation is performed greedily (temperature 0.0) with the same decoding truncation limits. In our experimental setup, we skipped the KL-regularization term by setting its corresponding coefficient to zero.

*Table 4.* Training configuration.

| Data Configuration | |
| --- | --- |
| Base Model | Qwen2.5-3B |
| Global Batch Size | 16 |
| Train Steps | 1410 |
| Total Epochs | 2 |
| **Rollout Inference** | |
| Rollout Num per Prompt | 8 |
| Temperature | 1.0 |
| Top-p | 1.0 |
| Top-k | $-1$ |
| Max Prompt Length | 4000 |
| Max Response Length | 4000 |
| **Actor Training** | |
| PPO Mini Batch Size | 32 |
| Advantage Estimation Type | GRPO |
| Clipping $\varepsilon_{\text{low}}$ | 0.2 |
| Clipping $\varepsilon_{\text{high}}$ | 0.2 |
| Optimizer | Adam |
| Learning Rate | $10^{-6}$ |
| Weight Decay | 0.1 |
| $(\beta_1, \beta_2)$ | (0.9, 0.999) |
| Gradient Norm Clipping | 1.0 |
| Learning Rate Scheduler | constant |
| Warmup Steps | 10 |
| KL coefficient ($\beta$) | 0.0 |
| **Evaluation Setup** | |
| Temperature | 0.0 |
| Top-p | 1.0 |
| Top-k | $-1$ |
| Max Generation Length | 4000 |

## N. Noise Injection Pseudocode

**Algorithm 1** Noisy Verifier Wrapper

**Require:** Oracle checker $\text{Oracle}(\cdot) \in \{0,1\}$, target $(\text{TPR}, \text{FPR})$
1: **function** NoisyCheck(program)
2:    $z \leftarrow \text{Oracle}(\text{program})$ {ground truth}
3:    **if** $z = 1$ **then**
4:      $r \leftarrow \text{Bernoulli}(\text{TPR})$
5:    **else**
6:      $r \leftarrow \text{Bernoulli}(\text{FPR})$
7:    **end if**
8:
9:    **return** $r$
10: **end function**

## O. Data Sample

### Example Coding Problem: kMarsh

**Problem Statement.** Solve the following coding problem using the programming language `Python`:

Mr. K has a rectangular plot of land which may contain marshes where fenceposts cannot be set. He wants you to find the *perimeter of the largest rectangular fence* that can be built on this land.

For example, in the following $m \times n = 4 \times 4$ grid, $x$ marks a marsh and `.` marks good land:

```
....
..x.
..x.
x...
```

If we number the rows and columns starting with 1, there are two main areas that can be fenced: $(1,1) - (3,2)$ and $(1,2) - (4,4)$. The longest perimeter is 10.

**Function Description.** Complete the function `kMarsh` in the editor below. It should print either an integer or the word `impossible`.

**kMarsh(grid):** • **Input:** an array of strings that represent the grid.
• **Output:** an integer representing the largest perimeter, or the string `impossible`.

**Input Format.**

• The first line contains two space-separated integers $m$ and $n$, the grid rows and columns.

• Each of the next $m$ lines contains $n$ characters describing the land: ''x'' (ASCII 120) if it is a marsh, and ''.'' (ASCII 46) otherwise.

**Constraints.**
$$2 \le m, n \le 500$$

**Output Format.** Print a single integer—the largest perimeter—or `impossible` if no rectangular fence can be built.

**Sample Input 0**

```
4 5
.....
.x.x.
.....
.....
```

**Sample Output 0**

```
14
```

**Explanation 0.** The fence can be built around the entire field.

$$\text{Perimeter} = 2(4 - 1) + 2(5 - 1) = 14.$$

**Sample Input 1**

```
2 2
.x
x.
```

**Sample Output 1**

```
impossible
```

**Explanation 1.** We need a minimum of four corner points to form a fence, hence it is impossible.

**Sample Input 2**

```
2 5
.....
xxxx.
```

**Sample Output 2**

```
impossible
```

**Explanation 2.** The lower row prevents forming a valid rectangle.

The input is provided via `stdin`, and the solution should print its result to `stdout`.

**Task.** Now, solve the problem and return the code.

### Test Cases for kMarsh

**Test Case List (JSON-like format):**

```
[
  {
    'fn_name': None,
    'input': '4 5\n.....\n.x.x.\n.....\n
        .....\n',
```

```
    'output': '14\n',
    'type': 'stdin_stdout'
  },
  {
    'fn_name': None,
    'input': '2 2\n.x\nx.\n',
    'output': 'impossible\n',
    'type': 'stdin_stdout'
  }
]
```

