# OpenReview forum: "Rate or Fate? RLV$^{\varepsilon}$R: Reinforcement Learning with Verifiable Noisy Rewards"
_ICML.cc/2026/Conference — ICML 2026 spotlight_

### Official Review · Reviewer_2fsb · 2026-03-08

**Soundness:** 3
**Presentation:** 2
**Significance:** 3
**Originality:** 3
**Overall Recommendation:** 5
**Confidence:** 3

**Summary:**

The paper studies the effect of noise in reward function on RLVR behavior, by modeling RLVR as a multi-armed bandit problem. The paper discovers that Youden's index $J$ is the key metrics. If it is positive, RLVR still works but noise may slow down convergence; If it is negative, RLVR will collapse. The statement is verified from both theoretical and empirical perspective.

**Compliance With Llm Reviewing Policy:**

Affirmed.

**Final Justification:**

I had two confusions regarding the theory of the paper, and they are fully resolved by the author's response. Now I understand the theory better and am more confident that the analysis is solid. In addition, the theory is applicable to continuous reward instead of binary alone. Therefore, this is a good paper connecting empirical methods and theory. I have raised my score and confidence.

**Key Questions For Authors:**

I might have misunderstood parts of the paper and come to two confusions below. I am happy to raise scores if the questions can be answered.
1. According to Section 5.3, advantage normalization $\sigma(p)$ is useful for convergence rates. However, there are some other RLVR algorithms like [RLOO](https://arxiv.org/abs/2402.14740) that do not use advantage normalization. In binary reward RLVR tasks, RLOO performs similarly as GRPO. Does that contradict the analysis?
2. As I understand, higher Youden's index implies faster convergence rate. Does that imply the reward function should be conservative so that FPR is close to 0? For example, assume a RLVR task has access to a reward oracle that outputs a confidence score $p\in [0,1]$, where higher $p$ means the oracle believes the response is more likely to be correct. The confidence is converted to a binary score by $r = 1[p>t]$, where $t$ is the decision threshold. When $t\to 1$, we are expected to have $TPR\approx 1$ and $FPR\approx 0$, so Youden's index $J\approx 1$ is near optimal. So according to the paper, we should set $t\approx 1$? But this does not make sense because then we have a meaningless reward function that almost always gives 0. How to resolve this paradox?

**Limitations:**

Yes.

**Strengths And Weaknesses:**

**Strength**
1. The paper offers a novel viewpoint that RLVR reward function con contain noise/error, while most prior work treats RLVR reward as ground truth. This is practical especially in coding tasks, in which test cases can be incomplete and thus false positive might exist.
2. The paper builds a strong theoretical analysis to support the claim. While the proof is not thoroughly checked, the result itself is natural, when FPR is high and even surpasses TPR, positive reward signal is no longer reliable and is actually leading the model towards incorrect actions. The claim is further verified with experiments, so that it also holds in practice.
3. The theoretical analysis does not only work for standard GRPO alone. The paper also applies the constructed ODE system to analyze effects of other factors, e.g., task difficulty, normalization and KL regularization.

**Weakness**
1. The theory only applies to binary reward. In practice, especially for coding tasks, reward function might be continuous that considers multiple factors, including correctness, efficiency, etc.
2. The paper can be further improved if suggestions can be made about how to raise the Youden's index, so that convergence can be accelerated.

---

> ### Author Rebuttal · Authors · 2026-03-28
>
> We sincerely thank you the constructive, detailed feedback. We also appreciate the care and effort reflected in your review. Due to the character limit, we have kept our responses brief.
>
> **Response to Q1.**
> We do **not** believe this contradicts the analysis. Section 5.3 concerns **convergence speed within a phase**, not the **phase boundary**. In our mean-field binary-reward model, RLOO and GRPO have the same sign structure and the same $J=0$ threshold; normalization changes only the rate along the same trajectory.
>
> For RLOO, the advantage is just the centered group baseline up to a positive constant:
>
> $$
> \hat A_g^{\mathrm{RLOO}}=r_g-\frac{1}{G-1}\sum_{h\neq g} r_h=\frac{G}{G-1}(r_g-\bar r).
> $$
>
> Accordingly, the bad-mass dynamics are
>
> $$
> \dot p_{\mathrm{RLOO}}(t)=-\eta \frac{G}{G-1}J[p(t)(1-p(t))]^2C_{\mathrm{geo}}(t)+\mathcal O(\eta^2),
> $$
>
> while for GRPO,
>
> $$\dot p_{\mathrm{GRPO}}(t)=-\eta \frac{J}{\sigma(p)}[p(t)(1-p(t))]^2C_{\mathrm{geo}}(t)+\mathcal O(\eta^2),\qquad\sigma(p)=\sqrt{p(1-p)}.
> $$
>
> Hence, for $p\in(0,1)$,
>
> $$
> F_{\mathrm{GRPO}}(p,y,z)=\frac{G-1}{G\sigma(p)}F_{\mathrm{RLOO}}(p,y,z),
> $$
>
> so the two vector fields differ only by a strictly positive scalar factor. Therefore they have the same drift direction, fixed points, and the same $J=0$ phase transition: if $J>0$, both decrease $p(t)$; if $J<0$, both increase it.
>
> So our claim should be read narrowly: **advantage normalization is a rate-conditioning mechanism, not a requirement for the sign-of-$J$ phase behavior.** This is why RLOO can perform similarly to GRPO on binary-reward RLVR tasks without contradicting the analysis.
>
> At the same time, the theory does **not** predict identical learning curves versus optimization step. The two methods share the same phase portrait only up to a positive time reparameterization, so quantitative speed differences should remain. In a simple 1D ODE illustration where GRPO moves from $p(0)=0.60$ to $p(500)=0.45$, the corresponding RLOO trajectory ends near $p(500)\approx0.52$, a moderate gap of about $6$ to $8$ percentage points. Thus similar empirical performance is expected, but exact equality is not.
>
> -----
>
> **Response to Q2**
>
>
>
> Thank you for this insightful question. The apparent paradox comes from conflating **true positive rate** with **precision**. For a thresholded verifier
> $$
> r_t = \mathbf{1}[s > t],
> $$
> where $s \in [0,1]$ is the oracle confidence score, the induced operating characteristics are
> $$
> \mathrm{TPR}(t) = \Pr(s>t \mid Y=1), \qquad \mathrm{FPR}(t) = \Pr(s>t \mid Y=0).
> $$
> As the threshold $t$ increases, the verifier becomes more conservative: both $\mathrm{TPR}(t)$ and $\mathrm{FPR}(t)$ decrease. In particular, $t \to 1$ does **not** imply $\mathrm{TPR} \to 1$; rather, in the extreme conservative limit, the verifier outputs almost all zeros, so typically $\mathrm{TPR}(t)\to 0$ and $\mathrm{FPR}(t)\to 0$. What may improve at a high threshold is **precision** $\Pr(Y=1 \mid s>t)$, not TPR.
>
> Therefore, our theory does **not** say one should choose $t \approx 1$. The relevant quantity is the verifier’s discriminative signal
> $$
> J(t) = \mathrm{TPR}(t) - \mathrm{FPR}(t),
> $$
> which, for a thresholded score, is generally maximized at an **interior** threshold, not at the most conservative one. Indeed, if $F_1$ and $F_0$ are the score CDFs on correct and incorrect responses, then
> $$
> \mathrm{TPR}(t)=1-F_1(t), \qquad \mathrm{FPR}(t)=1-F_0(t),
> $$
> so
> $$
> J(t)=F_0(t)-F_1(t).
> $$
> Hence, as $t \to 1$, both CDFs approach $1$, and thus
> $$
> J(t) \to 0.
> $$
> So the all-zero regime is not optimal: it makes the reward nearly uninformative and collapses the learning signal.
>
>
> This is fully consistent with our analysis. Our result should be interpreted as follows: **for a fixed binary verifier, larger $J$ yields a stronger and more correctly aligned learning signal; but when the verifier is obtained by thresholding a continuous confidence score, the right design choice is to select the threshold that maximizes the induced signal $J(t)$, not to push the threshold to the most conservative limit.** In other words, the theory favors **good separation**, not trivial abstention.
>
> ---
>
> **Response to Weakness Binary reward**:
>
>  Our analysis is **not limited to binary rewards**. The Bernoulli case is only a special instance. For a general continuous reward $r\in[0,1]$, the same derivation goes through:
>
> $$
> \mu_g:=\mathbb{E}[r\mid \mathrm{good}],\qquad
> \mu_b:=\mathbb{E}[r\mid \mathrm{bad}],\qquad
> \Gamma:=\mu_g-\mu_b.
> $$
> With bad-mass $p$ and within-class variances $v_g,v_b$,
>
> hence the effective driving signal remains
>
> $$
> \mathbb{E}[\tilde r\mid \mathrm{good}]-\mathbb{E}[\tilde r\mid \mathrm{bad}]=\frac{\Gamma}{\sigma(p)}.
> $$
> Therefore, all Appendix B-D results extend directly by replacing $J$ with the mean gap $\Gamma$ and replacing the Bernoulli variance $q(1-q)$ with the mixture variance above.
>
> ---
>
> **Response to raise J index**:
>
> We suggest make sure FNR is near zero, which is easier to achieve than reducing FPR.

---

> > ### Author Rebuttal · Reviewer_2fsb · 2026-04-01
> >
> > The questions I raised are my misunderstandings, and they are resolved by the authors. I've raised my score and confidence

---

### Official Review · Reviewer_7CFc · 2026-03-08

**Soundness:** 3
**Presentation:** 3
**Significance:** 2
**Originality:** 3
**Overall Recommendation:** 4
**Confidence:** 4

**Summary:**

This paper studies reinforcement learning with verifiable rewards under noisy binary verification, focusing on GRPO-style training where reward signals can contain both false positives and false negatives. The core idea is to model RLVR as a bandit over recurring reasoning modes and derive a mean-field, replicator-style dynamical system that tracks how probability mass moves between correct and incorrect modes during training.

Beyond the binary case, the paper extends the analysis to multiple good and bad modes on the probability simplex, showing that the dynamics decouple into within-group competition and overall bad-mass evolution. In this generalized view, the same J=0 phase boundary remains the central criterion, while the geometry of the simplex connects the GRPO update to replicator dynamics and natural-gradient flow. The paper also discusses implications for convergence speed, prompt learnability, reward variance asymmetry, and KL regularization, arguing that in the informative regime J>0, noisy verification mainly changes the rate of convergence rather than the eventual direction of learning.

Empirically, the authors validate the theory on Python code generation using Qwen2.5-3B trained with GRPO under synthetic verifier noise.

**Compliance With Llm Reviewing Policy:**

Affirmed.

**Key Questions For Authors:**

1. How robust is the sign-of-𝐽 phase boundary once the per-prompt mean-field abstraction is relaxed?
2. How sensitive are the empirical conclusions to group size 𝐺, training horizon, and KL regularization?
3. How should practitioners estimate or operationalize 𝐽 when “ground-truth correctness” itself is not directly observable?

**Limitations:**

yes

**Strengths And Weaknesses:**

Strengths:
- The paper’s main contribution is conceptually sharp: the sign of J acts as a phase boundary for the direction of learning under the proposed mean-field GRPO dynamics. This is a strong and easily interpretable message, and the derivation from the fitness gap to the bad-mass ODE is presented in a fairly clean way.

Weaknesses:
- The phrase “rate, not fate” is compelling, but it is only established within the mean-field regime and only partially tested in one finite-horizon experimental setup. In the empirical section, training runs for two epochs / 1410 steps with 𝐺 = 8 and 𝛽 = 0, which is a quite specific regime. That is enough to support the directional phenomenon, but not enough to conclude that verifier noise broadly “mainly rescales the clock” in practical RLVR systems.

---

> ### Author Rebuttal · Authors · 2026-03-28
>
> **Response to: _How robust is the sign-of-$J$ phase boundary once the per-prompt mean-field abstraction is relaxed?_**
>
> Thank you for this important question. We view the per-prompt mean-field model as an analytical approximation. It is a tractable lens for RLVR, not a claim that practical RLVR exactly reduces to this abstraction. The key point is that the sign-of-$J$ phase transition also appears empirically beyond the original setting, suggesting it is a robust feature of RLVR under verifier noise rather than an artifact of the model.
>
> To test this, we added two experiments beyond the original coding setup:
> 1. **GRPO on math reasoning**: train on **DAPO-17K**, evaluate on **AIME 2025**, using **Qwen3-4B-Instruct**.
> 2. **PPO on math reasoning**: using **DeepSeek-R1-Distill-7B** on **GSM8K**.
>
> In the new **GRPO math** experiment, we again observe the predicted sign-of-$J$ transition:
>
> | Regime | Step 0 | Step 230 (1 epoch) |
> |---|---:|---:|
> | $J = 1$ | 42.9\% | 60.0\% |
> | $J = 0.3$ | 44.7\% | 55.0\% |
> | $J = 0$ | 44.2\% | 33.54\% |
> | $J = -0.1$ | 42.9\% | 10.8\% |
>
> Thus, when $J>0$, performance improves; near $J=0$, training fails to progress reliably; and when $J<0$, training clearly anti-learns. We observe the same qualitative pattern beyond the original coding setup, including under **PPO**.
>
> **PPO experiment on math reasoning**
> Model: **Deepseek-llm-7b-chat**
> Benchmark: **GSM8K**
>
> | Regime | Step 0 | Step 30 (~1 epoch) | Step 58 (2 epochs) |
> |---|---:|---:|---:|
> | $J=1$ | 24\% | 63\% | 64\% |
> | $J=0.3$ | 24\% | 61\% | 63\% |
> | $J=0$ | 24\% | 21\% | 22\% |
> | $J=-0.1$ | 24\% | 3\% | 1\% |
>
>
>
> ---
>
> **Response to: _How sensitive are the empirical conclusions to group size $G$, training horizon, and KL regularization?_**
>
> Thank you for this important question. We agree that a full sensitivity study over $G$, training horizon, and KL would strengthen the paper. However, we believe it is important to separate two issues:
> (1) whether these parameters affect the **speed / sharpness** of the dynamics, and
> (2) whether they overturn the **qualitative sign-of-$J$ phase boundary**.
>
> Our current evidence is strongest on the second point. The main empirical conclusion is directional: when $J>0$, learning improves; near $J=0$, progress breaks down; and when $J<0$, training anti-learns. The added math-reasoning experiment supports this clearly.
>
> For **Qwen3-4B-Instruct + GRPO**, trained on **DAPO-17K** and evaluated on **AIME 2025** with **$G=8$**, we observe:
>
> | Regime | Step 0 | Step 130 | Step 230 |
> |---|---:|---:|---:|
> | $J=1$ | 42.9\% | 57.5\% | 60.0\% |
> | $J=0.3$ | 44.7\% | 51.1\% | 55.0\% |
> | $J=0$ | 44.2\% | 35.8\% | 33.54\% |
> | $J=-0.1$ | 42.9\% | 31.7\% | 10.8\% |
>
>
>
> For **group size $G$**, the theory suggests that larger $G$ should move the empirical process closer to the mean-field limit, not reverse the sign-of-$J$ effect. Importantly, the phase transition is already clear at the modest value **$G=8$**, indicating that the effect does not require an extreme large-$G$ regime.
>
> To further probe $G$, we ran an additional math-reasoning experiment with the same setup while varying rollout group size and **model size**:
>
> **Qwen3-4B-Instruct / Qwen3-14B-Base**.
>
> | G | Regime | Step 0 | Step 130 | Step 230 (1 epoch) | Delta |
> |---|---|---:|---:|---:|---:|
> | 8  | J=1   | 42.9 / 10.4 | 57.5 / 29.0 | 60.0 / 35.0 | +17.1 / +24.6 |
> | 8  | J=0.3 | 44.7 / 9.6  | 51.1 / 25.6 | 55.0 / 27.3 | +10.3 / +17.7 |
> | 16 | J=1   | 42.2 / 9.7  | 60.2 / 25.4 | 59.8 / 34.1 | +17.6 / +24.4 |
> | 16 | J=0.3 | 42.5 / 11.7 | 56.4 / 24.3 | 54.4 / 28.3 | +11.9 / +16.6 |
>
> For **longer horizons**, our **PPO** experiment extends to **two epochs**, and we observe that the noisy case $J=0.3$ moves very close to the $J=1$ trajectory. This directly supports the **"rate, not fate"** theory and in agreement with the ODE: when $J>0$, verifier noise primarily changes the convergence speed, not the qualitative endpoint, so the gap narrows over time exactly as predicted.
>
>
> For **KL regularization**, our empirical evidence is more limited, so we do not want to overclaim. In the analysis, the KL term adds a drift toward the reference policy, $p_{\mathrm{ref}}$, which can stabilize training but also damp task-driven movement if too strong. This is one reason many practical RLVR setups use weak KL or omit it entirely. Since our goal was to study verifier-noise effects in the more common practical regime, we focused on those settings. We agree that a dedicated KL ablation would be valuable future work.
>
> Overall, while a fuller sweep would strengthen the paper, the current and newly added evidence suggests that the core empirical conclusion, the **sign-of-$J$ phase boundary**, is qualitatively robust, while $G$, horizon, and KL mainly affect how quickly and how sharply that boundary is expressed in practice.

---

> > ### Author Rebuttal · Reviewer_7CFc · 2026-04-01
> >
> > I am very grateful to the authors for their detailed answers to my questions. I have raised my rating accordingly.

---

### Official Review · Reviewer_mi4W · 2026-03-12

**Soundness:** 4
**Presentation:** 3
**Significance:** 3
**Originality:** 4
**Overall Recommendation:** 5
**Confidence:** 4

**Summary:**

This paper provides a simplified theoretical model of GRPO dynamics which allows the authors to accurately predict various effects of noisy binary reward signals. The main simplification made is a per-prompt bandit structure over “arms” formed by broad classes of good or bad reasoning patterns. Using this model, the intuitive property is that noise only affects time to convergence, with three distinct regimes (learning, stagnation, anti-learning) depending only whether the reward is on average correct or not (expressed by the authors in terms of TPR minus FPR) is predicted and then empirically verified in a simple coding setting. Notably, several interesting side effects are also predicted from this model, including diversity collapse among good solutions and a tendency toward uniformity among bad solutions, an interesting dependence of convergence speed on the balance FPR and FNR, and mathematical analogies to evolutionary biology and natural gradient descent.

**Compliance With Llm Reviewing Policy:**

Affirmed.

**Final Justification:**

As explained in my original review, I think this paper is high-quality and timely and there should be more work like this on understanding emergent properties of commonly used RL algorithms with simple theoretical models. The authors addressed my remaining concerns and questions and I am comfortable that this is an accept-worthy paper.

**Key Questions For Authors:**

I have no key questions, but if the authors address any of my comments in “Weaknesses” above this would both make the paper even stronger and likely cause me to strengthen my score even more.

**Limitations:**

Yes.

**Strengths And Weaknesses:**

Strengths:

(Despite being a shorter list, these more than outweigh the weaknesses in my view.)

- The paper is very well-presented overall, with a concise and clear navigation of the main assumptions, methods and results. While the paper takes some mathematical literacy and effort/time investment to read, there are generally no severe points of confusion or sloppiness in the exposition of the authors’ work.
- The work comes just as more and more labs are scaling their use of GRPO (with implicitly noisy rewards) in practice, and a more principled understanding of learning dynamics for this and related RL algorithms on LLMs is sorely needed. The level of simplifying detail made in the authors’ model seems to be just enough to make understanding the dynamics tractable while drawing useful conclusions about learning behavior. I hope that this work will encourage more “pragramatic theory” papers along these lines to better understand more modern LLM RL outcomes.
- The simplified bandit lens the authors use to view GRPO through seems original and much of the theory in the paper is inspired and well-executed. There is lots of creativity seeping through this paper (unless there is very similar literature of which I am unaware!).
- The results in the paper seem sound, careful and correct, well-contextualised and limitations are carefully discussed.
- Some of the predictions in the paper (to do with intra-class diversity, convergence dependence on overall difficulty, effects of KL regularization) and mathematical analogies are genuinely interesting and worthwhile findings, and there are multiple of them!

Weaknesses:

(Many of the below aren’t weaknesses, rather suggestions for an even better presentation of the good results in the paper.)

- The main result of the paper, i.e. the three phases depending on Youden’s index, is somewhat obvious and not a surprising outcome. There is plenty of precedent for this in the RL literature and most practitioners will not learn something very new from this conclusion. However, demonstrating that the authors’ simplified model correctly produces this prediction, as well as producing various interesting side predictions, is important, and as well as enjoying the other theoretical findings from this paper, I have confidence that scaling these modelling techniques up to more complex scenarios (e.g. with continuous reward structures) could produce novel insights.
- The abstract of the paper is a little difficult to understand without having read the paper already. I would encourage the authors to phrase the abstract in slightly simpler terms so that the paper is more accessible to new readers. This comment extends to some of the mathematical detail in the introduction; I think the introduction should be at approximately the mathematical level of the current abstract, and additional detail in the introduction should be deferred to the relevant later section. This is just advice for readability outside the authors’ immediate circle of academic peers, and does not impact the quality of the paper in my view.
- I would like to see more discussion of the assumptions the authors make in their model and how significant each is/how they could be relaxed in the future, as well as what the take-aways of their results should be. For example, I think the case of LLM judges is the most practically important case of noisy rewards today: more discussion of implications for LLM judge rewards would be great, as well as future generalizations to continuous reward settings etc.
- I think there should be more upfront emphases on the “interesting side predictions” from their model, relative to the core phase transition prediction which in my view is more obvious.
- The paper introduces a lot of notation very quickly, and this notation in some cases has no obvious internal consistency or is even explicitly overloaded. For example, $p$ and $z$ both refer to various different things. This is also true for the TPR/FPR and $\delta_{FN}$ and $\delta_{FP}$ (redundancy) and for the vectors $y$, $z$ and the scalar $s$ (why these letters)? I would encourage the authors to use as few different letters as possible and retain the broad semantic association of each letter, using sub/superscripts and slightly longer expressions where necessary, so that the reader doesn’t have to store as many disparate meanings in their head at the same time. Of course, there is a balance to strike here. There is also some unnecessary confusion introduced due to off-by-one index differences (e.g. between their probability simplex dimension and the underlying vector space dimension).
- The caption of Figure 1 does not really describe what the underlying experiment is (setting, number of runs, etc).
- Some terms (e.g. mean-field approximation, replicator-style flow) could do with more upfront defining.

---

> ### Author Rebuttal · Authors · 2026-03-29
>
> We sincerely thank the reviewer for the careful reading and thoughtful suggestions.
>
> Your comments on the presentation are very helpful, and we plan to incorporate them in the revision:
>
> - **Abstract and introduction.** We agree that the abstract and early introduction can be made more accessible. In the revision, we will simplify the abstract, reduce the mathematical density of the introduction, and defer more technical detail to later sections.
>
> - **Assumptions and practical takeaways.** We agree that the modeling assumptions and their significance should be discussed more explicitly. We will expand the discussion of which assumptions are structural to the theory, which are simplifying approximations, and how they may be relaxed in future work. We will also strengthen the discussion of practical implications, especially for **LLM-judge rewards**, which we agree are among the most important noisy-reward settings in practice.
>
> - **Continuous rewards.** Thank you for raising this direction. Our analysis is **not fundamentally restricted to binary rewards**. The Bernoulli case is simply the special case where the reward channel is binary, in which the signal term becomes Youden’s index $J$. For a general continuous reward $r\in[0,1]$, the same derivation goes through after replacing binary error rates by the class-conditional reward moments:
>
> $$
> \mu_g:=\mathbb E[r\mid \mathrm{good}],\qquad
> \mu_b:=\mathbb E[r\mid \mathrm{bad}],\qquad
> \Gamma:=\mu_g-\mu_b .
> $$
>
> Then, with bad-mass $p$ and class-conditional variances $v_g,v_b$,
>
> $$
> q(p)=\mathbb E[r]=(1-p)\mu_g+p\mu_b,\qquad
> \sigma^2(p)=\mathrm{Var}(r)=(1-p)v_g+p v_b+p(1-p)\Gamma^2 .
> $$
>
> After the same $z$-score normalization $\tilde r=(r-q(p))/\sigma(p)$, the conditional normalized means become
>
> $$
> \mathbb E[\tilde r\mid \mathrm{good}]=\frac{p\Gamma}{\sigma(p)},\qquad
> \mathbb E[\tilde r\mid \mathrm{bad}]=-\frac{(1-p)\Gamma}{\sigma(p)},
> $$
>
> so the effective driving signal is still
>
> $$
> \mathbb E[\tilde r\mid \mathrm{good}]-\mathbb E[\tilde r\mid \mathrm{bad}]
> =\frac{\Gamma}{\sigma(p)} .
> $$
>
> Thus, all Appendix B-D results extend directly by replacing $J$ with the mean-gap $\Gamma=\mu_g-\mu_b$ and replacing the Bernoulli variance $q(1-q)$ with the general mixture variance above. The binary case is recovered exactly since $\mu_g=1-\delta_{\mathrm{FN}}$ and $\mu_b=\delta_{\mathrm{FP}}$, hence $\Gamma=1-\delta_{\mathrm{FN}}-\delta_{\mathrm{FP}}=J$.
>
> - **Emphasis on side predictions.** We agree that the side predictions of the model deserve more prominent emphasis. In the revision, we will better foreground results such as diversity collapse, difficulty dependence, and the effect of KL regularization, rather than focusing too heavily on the phase-boundary result alone.
>
> - **Notation and exposition.** We appreciate the detailed comments on notation. We agree that some notation is introduced too quickly and that certain symbols are overloaded. We will streamline the notation, reduce redundancy where possible, and fix the index inconsistencies to improve readability.
>
> - **Figure captions and terminology.** We agree that Figure 1 should describe the experimental setup more clearly, including the setting and number of runs. We will also define terms such as "mean-field approximation" and "replicator-style flow" more explicitly when they first appear.
>
> We are very grateful for these suggestions. They are highly actionable, and we believe they will  improve the clarity and accessibility of the final version.

---

> > ### Author Rebuttal · Reviewer_mi4W · 2026-04-03
> >
> > Thank you for addressing my remaining concerns. I have raised my confidence in my current score. This is a very nice paper that should definitely be accepted.

---

### Official Review · Reviewer_fwMM · 2026-03-23

**Soundness:** 3
**Presentation:** 3
**Significance:** 1
**Originality:** 2
**Overall Recommendation:** 4
**Confidence:** 3

**Summary:**

This paper studies the effect of noisy rewards (i.e., verifier errors with TP/FP noise) on learning dynamics in RLVR. The authors model the GRPO optimization process as a mean-field ODE system and use replicator dynamics to analyze how policy probability mass evolves between correct and incorrect modes. The key theoretical finding is that noise primarily affects learning *rate* rather than convergence *fate*, with a characterization of the phase transition between learning and anti-learning.

**Compliance With Llm Reviewing Policy:**

Affirmed.

**Final Justification:**

My previous major concern has been addressed by authors' rebuttal, so I plan to raise my score.

**Key Questions For Authors:**

1. **Distinction from noisy label learning**: The TP/FP noise analysis closely parallels classical noisy label learning. What is the essential theoretical or empirical incremental contribution of this work relative to those literatures? Clarifying this would substantially improve the originality assessment.
2. **Extension to sequence-level RLVR**: The current framework operates at the mode level and abstracts away sequential decision making. Can the ODE framework be extended to a sequence-level/trajectory-level RLVR setting, e.g. in some agentic RL tasks? A positive answer would significantly strengthen the paper's practical relevance.
3. **Generalization across tasks and scales**: Current experiments are limited to a single task and small model sizes. Does the theory hold for other RLVR tasks (e.g., math reasoning, tool use, agent environments) or larger models? Are there plans for broader validation?
4. **Theory-guided algorithm design**: The paper suggests the framework can inform training strategies but provides no concrete method. Have the authors attempted to derive a novel RLVR optimization strategy from this analysis and verified it against existing baselines? This would meaningfully close the gap between theory and practice.

**Limitations:**

Yes.

**Strengths And Weaknesses:**

**Strengths**

- **Soundness**: This work is theoretically rigorous. The ODE and replicator dynamics formulation is self-consistent and provides a clear dynamical explanation of how TP/FP noise affects learning direction and convergence speed.
- **Presentation**: The paper is generally well-structured narrative from problem setup to modeling, analysis, and experiments. The learning vs. anti-learning intuition is clearly communicated.
- **Significance**: The authors address a practically relevant issue in RLVR (reward noise in verifiable tasks like code generation) and provides a diagnostic lens for understanding RL training failures.
- **Originality**: The mapping of GRPO training to an ODE system and the "rate vs. fate" framing offer a degree of formal novelty.

**Weaknesses**

- **Limited methodological novelty**: Modeling reward noise via TPR/FPR and analyzing its effect on learning is fundamentally similar to classical noisy label learning; the ODE/replicator formulation largely restates known phenomena rather than introducing new mechanisms or insights.
- **Insufficient LLM/RLVR specificity**: The analysis oversimplifies the RLVR problem to a mode-level single-step bandit, ignoring sequence-level decision making, credit assignment, and multi-step reasoning. This  makes the conclusions closer to a general noisy learning analysis than a contribution specific to RLVR problem in LLM.
- **Narrow experimental scope**: Experiments focus on small-scale or synthetic settings with limited task diversity; validation across math reasoning, agent environments, or larger model scales is absent.

---

> ### Author Rebuttal · Authors · 2026-03-28
>
> We sincerely thank you the constructive, detailed feedback. Your comments helped us improve the paper’s clarity. We  also appreciate the care and effort reflected in your review. Due to the character limit, we have kept our responses brief.
>
> **Q1: Distinction from noisy-label learning.**
> We agree that our work is conceptually related to classical noisy-learning literature, and we do not claim that TPR/FPR-style noise models or replicator dynamics are new in isolation. Our contribution is specific to **RLVR for LLM post-training**: to the best of our knowledge, this is the first rigorous mean-field study of **noisy verifier rewards** in this setting. Moreover, Proposition 5.1 gives a **full dynamical system on the simplex of mode probabilities**, in addition to ODE for expected training accuracy, $p$. This yields RLVR-specific geometric consequences, including the sharp $J=0$ phase boundary, diversity collapse in GRPO, and the learnability / KL analyses in Section 5.3. In particular, for $J>0$, GRPO not only reduces bad mass but also drives within-good-mode competition and collapse toward a simplex vertex (Section 5.2, Fig. 2, Appendix I, J). We will clarify this distinction from prior noisy-learning work.
>
> ---
>
> **Q2: Extension to sequence-level RLVR.**
> Our goal is not to claim that full sequence-level RLVR exactly reduces to a single-step bandit, but to use a tractable per-completion abstraction to isolate verifier-noise effects. For GRPO, token-level credit assignment does not arise in the same way because GRPO does not train a separate critic and uses final sequence-level rewards to induce a uniform token-level advantage. To address the reviewer’s concern directly, we ran an additional **PPO** experiment, where token-level credit assignment is present, and observed the same qualitative sign-of-$J$ behavior on **GSM8K** with **Deepseek-llm-7B-chat**:
> **PPO experiment on math reasoning**
> Model: **Deepseek-llm-7B-chat**
> Benchmark: **GSM8K**
> | Regime | Step 0 | Step 30 (~1 epoch) | Step 58 (2 epochs) |
> |---|---:|---:|---:|
> | $J=1$ | 24\% | 63\% | 64\% |
> | $J=0.3$ | 24\% | 61\% | 63\% |
> | $J=0$ | 24\% | 21\% | 22\% |
> | $J=-0.1$ | 24\% | 3\% | 1\% |
>
> Thus, even with explicit token-level credit assignment, positive $J$ leads to learning, near-zero $J$ to stagnation, and negative $J$ to anti-learning. This supports that the phenomenon is not tied only to the GRPO abstraction. The term "bandit" is used only to distinguish the per-completion view from a token-level action space, consistent with prior RLHF/LLM literature such as [Back to Basics: Revisiting REINFORCE Style Optimization for Learning from Human Feedback in LLMs](https://arxiv.org/abs/2402.14740) and [Weight Ensembling Improves Reasoning in Language Models](https://arxiv.org/pdf/2504.10478v4).
>
> ---
>
> **Q3: Generalization across tasks and scales.**
> We agree broader validation is important. Motivated by this feedback, we added two experiments beyond the original coding setup:
> (1,2) **GRPO on math reasoning**: train on **DAPO-17K**, evaluate on **AIME 2025**, using **Qwen3-4B-Instruct** and bigger model
> (3) **PPO on math reasoning**: see above.
>
> For the new **GRPO math** experiment, we again observe the predicted phase transition:
>
> Model: **Qwen3-4B-Instruct**
>
> | Regime | Step 0 | Step 130 | Step 230 (1 epoch) |
> |---|---:|---:|---:|
> | $J=1$ | 42.9\% | 57.5\% | 60.0\% |
> | $J=0.3$ | 44.7\% | 51.1\% | 55.0\% |
> | $J=0$ | 44.2\% | 35.8\% | 33.54\% |
> | $J=-0.1$ | 42.9\% | 31.7\% | 10.8\%
>
> Model: **Qwen3-14B-Base**:
> | Regime | Step 0 | Step 130 | Step 230 (1 epoch) |
> |---|---:|---:|---:|
> | $J=1$ | 10.0\% | 28.9\% | 35.0\% |
> | $J=0.3$ | 9.6\% | 24.4\% | 27.3\% |
>
> when $J>0$, learning proceeds; near $J=0$, progress breaks down; and when $J<0$, training anti-learns.
>  Together with the PPO result above, this supports that the sign-of-$J$ phase behavior is not restricted to a single task, algorithm, or model family. We agree that covering another settings like agentic and multi-turn RL  are important future work.
>
> ---
>
> **Q4: Theory-guided algorithm design.**
> Yes, we believe the framework is not only explanatory but also constructive. In follow-up work, we asked how to preserve accuracy while preventing the diversity collapse predicted by GRPO. The mean-field analysis shows that this requires a geometric intervention in the inner good-arm dynamics. At first order,
> $$
> \Delta y = \eta J(y)B(y), \qquad J(y)=\mathrm{Diag}(y)-yy^\top.
> $$
> To impose a uniform-restoring field $v_i(y)=\lambda(1/K-y_i)$, one chooses
> $$
> B_i(y)=\bigl[\mathrm{Diag}(y)^{-1}v(y)\bigr]_i
> =\lambda\left(\frac{1}{K y_i}-1\right)\propto \frac{1}{y_i}.
> $$
> This analysis motivates the inverse-probability bonus and led to a new algorithm that prevents diversity collapse and pushes exploration toward the simplex interior, with strong experimental support. Although beyond this submission’s scope, this shows the framework can guide RLVR algorithm design rather than merely explain it post hoc.

---

> > ### Author Rebuttal · Reviewer_fwMM · 2026-04-03
> >
> > I acknowledge the authors' detailed response. Specifically, the extensions PPO algorithm and new tasks confirms the effectiveness of the proposed method using $J$. Although it's still originated from noisy-label learning, reward is kind of different than the labels in supervised learning. I appreciate the merits of detailed derivation on this noisy rewards setting and how it impacts the policy gradient. I will raise my score for a positive recommendation.

---

### Decision · Program_Chairs · 2026-04-30

**Decision:**

Accept (spotlight)

**Comment:**

This paper studies noisy binary rewards in RLVR, modeling GRPO optimization as a mean-field ODE system using replicator dynamics over reasoning "modes." The key result is that Youden's index J = TPR − FPR governs a phase transition: J > 0 allows learning (albeit slower), J < 0 causes anti-learning. Secondary predictions cover diversity collapse, difficulty dependence, reward variance asymmetry, and KL regularization effects.

All reviewers recommended acceptance after the rebuttal. The theoretical framework is praised as rigorous and well-executed, the "rate vs. fate" framing is smart, and the secondary predictions go well beyond the core phase-transition result. The main pre-rebuttal concern of limited empirical scope was convincingly addressed by demonstrating consistent phase-transition behavior across new tasks, models, and algorithms. Minor presentation concerns (dense abstract, occasional notation overload) remain but are straightforwardly addressable.

This is a theoretically rigorous and timely paper providing a principled framework for understanding an important RLVR phenomenon. The core result is clean, the secondary predictions add substantial value, and the rebuttal convincingly demonstrated robustness across settings.